# Readability ≠ Learnability: Rethinking the Role of Simplicity in Training Small Language Models

**Ivan Lee & Taylor Berg-Kirkpatrick**
UC San Diego
{iylee,tberg}@ucsd.edu

## Abstract

Recent studies suggest that very small language models (SLMs) can generate surprisingly coherent text when trained on simplified, child-directed corpora such as TinyStories. These findings have been interpreted as evidence that readability—characterized by accessible vocabulary, familiar narrative structure, and simple syntax—plays a key role in enabling such capabilities to emerge. In this paper, we challenge that interpretation. We construct synthetic datasets with matched structure but varied readability, and find that readability alone does not predict coherence or learning efficiency in SLMs. Models trained on complex, adult-level text perform comparably to those trained on simplified language, and even exhibit faster development of coherence during training. Instead, we show that statistical simplicity, as measured by n-gram diversity, is a stronger predictor of learnability. Our findings caution against the growing trend of anthropomorphizing language model training—drawing parallels to human cognitive development without empirical basis—and argue for more precise reasoning about what properties actually support capability emergence in small models.[1]

## 1 Introduction

Recent studies have shown that very small language models (SLMs) can generate surprisingly coherent text when trained on the TinyStories dataset—a synthetic corpus of short, child-directed narratives written in highly readable language (Eldan & Li, 2023). These findings have led researchers to draw a connection between the simplicity of the dataset and the capabilities of the resulting models, with some suggesting that the use of simplified, developmentally appropriate language may play a key role in enabling coherence at small scales (Haga et al., 2024; Muckatira et al., 2024).

But what exactly is responsible for the success of TinyStories? Is it the readability of the text—short sentences, common words, familiar structure—that enables these models to succeed? Or are the benefits better explained by other properties of the data, such as its synthetic origin, its low lexical and structural diversity, or the statistical regularities that result from its template-based generation process? While the TinyStories authors themselves do not make strong causal claims, the broader community has often cited the dataset in ways that imply such a connection. In practice, TinyStories is frequently described as a corpus of simple children's stories, and its effectiveness is often linked, implicitly or explicitly, to that simplicity (Theodoropoulos et al., 2024; Bunzeck & Zarrieß, 2023).

The term "simple" in this context is ambiguous. On one hand, it may refer to readability—that is, how easily a text can be understood by a human reader, particularly a child. On the other hand, it may refer to statistical simplicity: low entropy, high redundancy, and a narrow distribution over token sequences, often measurable through metrics like n-gram diversity. These are distinct notions, and conflating them risks misunderstanding what actually enables small models to generalize.

---

[1] https://huggingface.co/collections/ivnle/llamatales-6716dad1a3113c4c3ea1038e

In this paper, we disentangle these two notions of simplicity. We construct synthetic datasets with matched structure but varied readability, using prompt templates with identical structure but explicitly differing in intended readership (child vs. adult) to ensure consistent format and narrative framing. We then train small transformer models on these datasets and evaluate their ability to generate coherent text. We find that models trained on complex, adult-level language perform comparably to those trained on simplified, child-directed text—and in some cases, coherence emerges earlier during training. These findings suggest that readability is not the key factor enabling coherence in SLMs.

These results challenge the intuition that child-directed language plays a special role in enabling language models to generalize. While the TinyStories authors do not make this claim explicitly, the dataset is often framed—by citations and surrounding discourse—as a developmentally inspired intervention (Edman & Bylinina, 2023; Yam & Paek, 2024; Feng et al., 2024). This framing aligns with a broader and growing trend: the anthropomorphization of language models. As Shanahan (2023) and Placani (2024) argue, anthropomorphization can be more than a harmless shorthand—it risks distorting scientific reasoning, misrepresenting model capabilities, and shaping misguided intuitions about how language models learn. In this case, attributing coherence to readability conflates human developmental simplicity with statistical learnability, obscuring what properties actually drive capability emergence in small models. Our findings instead suggest that statistical simplicity—rather than readability or developmental relevance—is a stronger predictor of learnability in SLMs.

## 2 Dataset Construction

Our investigation hinges on comparing models trained on datasets with specific properties. Therefore, we began by constructing datasets guided by three desiderata: (1) **Controlled Readability:** Our primary goal is to isolate readability differences between datasets—specifically, differentiating child-directed versus adult-directed language—to better understand how readability affects model coherence, while holding other properties constant. (2) **Statistical Simplicity:** We aim to minimize variance in statistical complexity—operationalized through metrics such as n-gram diversity—across datasets that differ in readability or thematic domain, as our primary hypothesis is that statistical simplicity significantly influences the learnability of SLMs. (3) **Consistent Quality:** We strive to maintain uniformly high dataset quality, ensuring variations in readability or domain do not substantially degrade text quality, thus allowing us to attribute differences in model performance specifically to readability or statistical simplicity. These terms are defined and quantitatively validated in Section 3.

Guided by these desiderata, and ensuring fair comparisons by standardizing dataset size to approximately 1 billion tokens (for both synthetic and sampled corpora), we construct or select the following datasets for comparison:

**TinyStories** The TinyStories dataset consists of synthetic children's stories generated via a structured, prompt-based pipeline using proprietary language models. Specifically, the authors created prompts from a fixed template, introducing variability by randomly sampling vocabulary items from a curated set of child-friendly words and narrative features such as dialogues or plot twists. While this approach is intended to promote high readability and narrative diversity, its reliance on proprietary models limits reproducibility.

**LlamaTales-Jr** We reconstruct the data generation pipeline introduced by TinyStories, using open-weight models (Llama-3.1-8B-Instruct). This produces a reproducible approximation of TinyStories. We employ the same fixed prompt template, curated child-friendly vocabulary, and narrative features as the original TinyStories dataset, aiming to maintain consistency in readability, statistical simplicity, and quality. See Figure A20 for the template.

**LlamaTales-GRE** We adapt the LlamaTales-Jr pipeline to target adult readers by modifying the fixed prompt template to explicitly instruct the use of vocabulary intended for college-educated adults. Specifically, we replace the curated child-friendly vocabulary with more sophisticated GRE-level vocabulary. Narrative structures and feature distributions (e.g.,

inclusion of dialogue, twist endings) remain identical, intending to isolate readability as the primary differentiating factor. The specific template is detailed in Figure A21.

**Domain Variants** We further extend the LlamaTales-GRE pipeline by adapting the fixed prompt template to create three additional synthetic datasets, each targeting a distinct thematic domain while maintaining the same GRE-level vocabulary. These datasets include LlamaTales-History (short summaries of historical events), LlamaTales-Sports (fictional sports articles), and LlamaTales-News (news-style narratives resembling mainstream journalism). These domain variants are intended to help assess the robustness and generalizability of our findings regarding readability and statistical simplicity across varied content domains. Prompt templates for these variants are shown in Figures A22, A23, A24.

**Standard Pretraining Data** We complement our synthetic datasets with subsets from established real-world pretraining datasets, primarily as points of reference that represent substantially higher statistical complexity. These datasets include our primary dataset, FineWeb-Edu (Lozhkov et al., 2024)—a highly filtered web dataset previously demonstrated to effectively train performant language models—and supplementary datasets such as SlimPajama (Soboleva et al., 2023) and Dolma (Soldaini et al., 2024). While we do not attempt to directly generalize our findings from synthetic to naturalistic datasets, these datasets illustrate the detrimental impact of increased statistical complexity on SLM coherence.

Thus far, our dataset construction has been guided primarily by the desiderata outlined above. In the next section, we provide detailed quantitative evidence confirming that these datasets align with those goals. Section 4 then presents the core experimental results.

## 3  Measuring and Validating Dataset Properties

To rigorously assess the properties targeted during dataset construction (Section 2), precise operationalization and quantitative validation are necessary. This section defines measurable metrics for readability, statistical simplicity, and quality, empirically validates their suitability, and confirms that our datasets exhibit the intended characteristics.

### 3.1  Readability

Readability—the ease with which humans comprehend text—is intuitive yet challenging to quantify automatically. While human judgment is the gold standard, its large-scale application is infeasible. We therefore evaluated three automated approaches: (1) classic readability formulas (e.g., Flesch-Kincaid), relying on surface features like sentence length and syllable counts; (2) constituency parsing metrics, assessing syntactic complexity; and (3) LLM-as-a-judge, using instruction-tuned LLMs to directly score readability (Figure A10).

To select the most appropriate metric, we correlated these automatic measures with human readability judgments from the CLEAR dataset (Heintz et al., 2022), as shown in Figure 1a. Large-scale LLMs (Llama-3.1-70B-Instruct, Qwen2-72B-Instruct) demonstrated the strongest correlation with human judgments (Pearson r = 0.74), substantially outperforming classic formulas (e.g., Flesch-Kincaid Grade: r = 0.49) and parsing metrics (e.g., max tree depth: r = 0.34). Given its superior alignment with human perception, we adopt LLM-as-a-judge using Llama-3.1-70B-Instruct for readability measurement throughout our experiments. While acknowledging that LLM-based scoring is not a perfect substitute for human evaluation and may carry its own biases, its strong empirical grounding on this benchmark makes it the most suitable metric for our large-scale analysis.

### 3.2  Quality

Evaluating the quality of generated text is necessary for comparing models, but manual assessment is impractical due to subjectivity and scale. We therefore require automatic proxies. We explored two: perplexity, measuring text likelihood averaged across moderately sized models (Mistral-7B-v0.3, Qwen2-7B, Llama-3.1-8B—sufficient per validation, see Figure 1b), though lower perplexity doesn't guarantee higher overall text quality; and

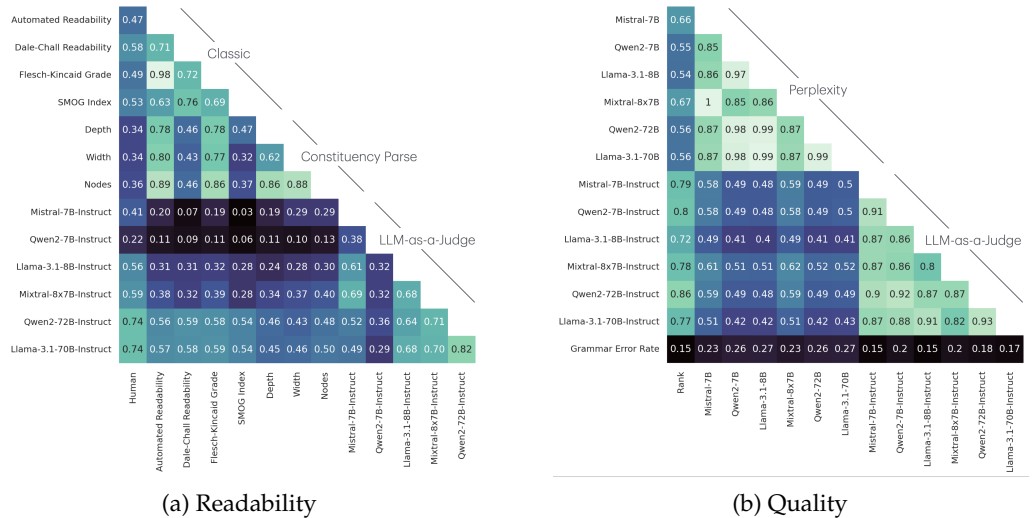

(a) Readability

(b) Quality

Figure 1: Validation of readability and quality metrics. (**a**) LLM-based readability scores correlate best with human judgments ($r = 0.74$ via Llama-3.1-70B-Instruct on CLEAR dataset), outperforming classic formulas and parsing metrics. (**b**) LLM-based coherence scores correlate best with our model ranking (Table A4), outperforming perplexity.

**LLM-as-a-judge.** For the latter, rather than using an ambiguous generic "quality" score, we prompted an LLM (Figure A11) to assess coherence—logical structure and flow.

To select the more effective proxy, we correlated both measures against a constructed ranking reflecting broad, recognized tiers of model generation capability (e.g., Llama 3.1 vs. GPT2; Table A4). This pragmatic reference helps validate which proxy better captures obvious quality differences. LLM-judged coherence demonstrated significantly stronger correlation with this ranking (Figure 1b) and minimal correlation with readability (Figure A27). While several large instruction-tuned models performed well as coherence judges, we selected Llama-3.1-70B-Instruct for consistency with the model used to judge readability. Based on its strong empirical performance relative to perplexity and its orthogonality to readability, we adopt LLM-judged coherence as our primary quality metric.

Notably, while coherence was the specific dimension prompted, our core conclusions remain consistent when evaluating other relevant quality aspects. Supplementary analyses using the same LLM-as-a-judge method to assess dimensions like fluency and clarity yielded the same patterns regarding dataset effects.

### 3.3 Statistical Simplicity

We quantify statistical simplicity using n-gram diversity, computed over 1-gram to 8-gram sequences within each dataset. This metric captures how often certain token patterns repeat—a proxy for the dataset's distributional regularity, that is, the extent to which sequences are structured, repetitive, and predictable.

N-gram statistics provide a simple and interpretable way to characterize how structured or compressible a dataset is. Because they reflect how often specific token sequences recur, they offer a useful lens into the dataset's underlying predictability—especially for small models that have limited capacity to track long-range dependencies.

### 3.4 Validation of Dataset Construction

In Section 2, we described how our datasets were constructed using intuitive, common-sense decisions about prompt structure, vocabulary, and narrative framing—without relying on formal methodology or prior empirical validation. In this section, we evaluate whether

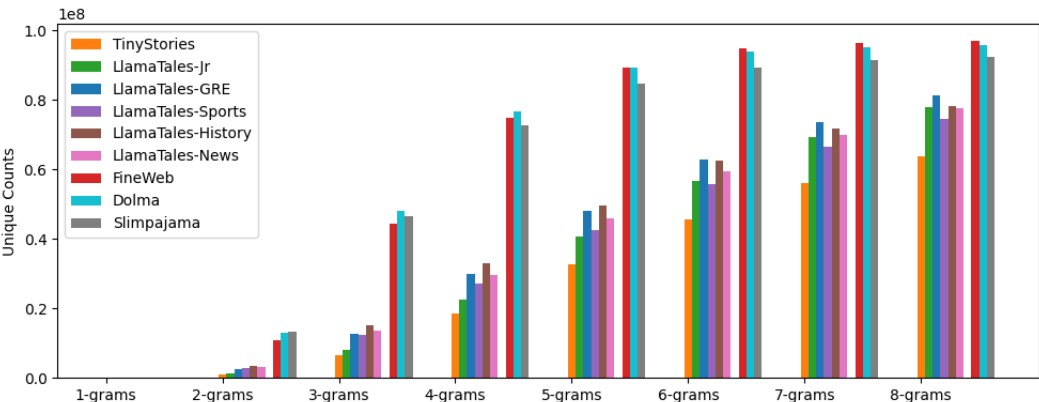

Figure 2: Unique n-gram counts across datasets (100M token samples). The synthetic datasets (TinyStories, LlamaTales series) are statistically simpler than standard pretraining corpora (FineWeb, Dolma, SlimPajama), as demonstrated by their substantially fewer unique n-grams (indicating lower diversity and higher predictability).

those intuitively constructed datasets actually exhibit the three properties we aimed to target: controlled readability, statistical simplicity, and consistent quality.

**Statistical Simplicity** Figure 2 reveals two distinct clusters in n-gram diversity, measured as the number of unique n-grams in 100M-token samples. Synthetic datasets (TinyStories and the LlamaTales series) exhibit markedly lower diversity than standard pretraining corpora such as FineWeb, Dolma, and SlimPajama. This separation confirms that our data generation pipeline successfully produced datasets with the intended statistical simplicity.

Table 1: Quantitative comparison of dataset properties. Metrics confirm controlled readability differences and high coherence within synthetic datasets. **Top:** Classic readability formulas. **Mid:** Constituency parsing. **Bot:** LLM-as-a-judge.

|                           | TinyStories | LlamaTales-Jr | LlamaTales-GRE | FineWeb |
| ------------------------- | ----------- | ------------- | -------------- | ------- |
| **Automated Readability** | 2.9         | 2.9           | 12.4           | 13.1    |
| **Coleman–Liau**          | 3.7         | 3.8           | 10.4           | 11.8    |
| **Dale–Chall**            | 5.7         | 5.7           | 9.1            | 9.3     |
| **Flesch–Kincaid**        | 2.4         | 2.2           | 9.6            | 10.7    |
| **Gunning Fog**           | 4.6         | 3.8           | 11.7           | 12.1    |
| **Linsear Write**         | 4.2         | 3.3           | 13.2           | 12.7    |
| **SMOG**                  | 5.7         | 5.4           | 11.3           | 12.6    |
| **Spache Readability**    | 2.7         | 2.5           | 5.5            | 5.5     |
| **Depth / Sentence**      | 6.8         | 6.4           | 10.6           | 9.5     |
| **Width / Sentence**      | 5.1         | 4.7           | 8.0            | 7.5     |
| **Nodes / Sentence**      | 19.6        | 17.2          | 42.1           | 37.8    |
| **Readability**           | 92.6        | 92.7          | 64.8           | 68.2    |
| **Coherence**             | 90.1        | 89.5          | 94.4           | 77.4    |

**Controlled Readability** Table 1 shows that our datasets vary systematically in readability. LlamaTales-GRE, constructed using GRE-level vocabulary, scores lower than LlamaTales-Jr and TinyStories across classic readability formulas, parsing metrics, and LLM-based judgments. This validates our ability to control for readability during dataset construction.

**Consistent Quality** Despite these differences, coherence remains consistently high across all synthetic datasets. FineWeb—despite having similar readability to LlamaTales-GRE—shows notably lower coherence. This likely reflects the noisier, less structured nature

of web-sourced datasets, even after extensive filtering by their original authors. In contrast, synthetic data generated using instruction-tuned models tends to be more consistently well-formed, reflecting their optimization for human-preferred outputs.

Having validated the properties of our datasets, we next examine how differences in readability and statistical simplicity affect model behavior. We train SLMs from scratch on each dataset and evaluate their ability to generate coherent stories using prompts drawn from both in-distribution and out-of-distribution test sets. This setup allows us to assess whether and how these data properties influence coherence and generalization in small-scale models.

# 4 Results

**Experimental Setup** We train transformer language models from scratch on each dataset described in Section 2. These models range in size from 262K to 33M non-embedding parameters. Each model is trained for 10 billion tokens across 10 epochs. We also evaluate several public pretrained models (e.g., GPT-2, Pythia, Mistral, Qwen2, Llama-3) as baselines without fine-tuning them on our data (details in Table A2).

To assess model performance, we generate 1,000 completions per model-dataset pair using top-p sampling (p = 0.95). Prompts consist of 50-token excerpts drawn from the test split of each respective dataset. We evaluated these generations using several automated metrics. Following our validation in Section 3.4, LLM-judged coherence serves as our primary measure of generation quality for the core results presented in this section. Supplementary analyses using other metrics, including perplexity and additional LLM-judged dimensions (such as readability, fluency, clarity, consistency, and grammar), are shown in Figures A3-A9).

**High Coherence Does Not Require Readable Text** While datasets like TinyStories are often highlighted for their simplicity and readability, our results indicate that readability itself is not the necessary factor for coherence to emerge in SLMs. As shown in Figure 3, SLMs trained on synthetic data with low readability—specifically LlamaTales-GRE—achieve high in-distribution coherence scores. For instance, a 33M parameter model trained on LlamaTales-GRE (blue dots, Figure 3b) reaches a coherence score comparable to that of Llama-3.1-70B when evaluated on LlamaTales-GRE prompts. This level of in-distribution performance is on par with that achieved by models trained on high-readability LlamaTales-Jr (green dots, Figure 3a). This holds despite LlamaTales-GRE text being significantly less readable. This pattern also persists across different narrative domains (news, sports, history), as shown in Figure A9, reinforcing the finding that human readability is not the primary driver for SLMs generating coherent text.

**Readability Does Not Speed Up Coherence Emergence** A potential refinement of the readability hypothesis is that simpler language might accelerate learning, mirroring developmental stages where simpler input is often assumed beneficial. Our findings, however, suggest a different dynamic during training. We tracked coherence development throughout training (Figure 4b). Models trained on the less readable, more complex LlamaTales-GRE dataset (blue) achieve a high level of coherence remarkably quickly, surpassing a score of 85 after just the first epoch, followed by more gradual improvement. In contrast, models trained on the simpler TinyStories (orange) or LlamaTales-Jr (green) datasets start at lower coherence levels and exhibit a much steeper learning curve over subsequent epochs, indicating they require more training data exposure to reach high coherence compared to the rapid initial learning facilitated by the complex LlamaTales-GRE data. This rapid attainment of high coherence when training on complex text directly challenges the developmental framing; if simpler, child-directed language were inherently easier to learn from, we would expect it to enable faster, not slower, initial emergence of coherence.

**Linking Learnability to Statistical Simplicity** If readability explains neither final coherence nor learning speed, what property does? We propose statistical simplicity, measured here via n-gram diversity, is the key factor. To investigate this, we define a learnability ratio: the coherence of a model's output divided by the coherence of the dataset it was trained on.

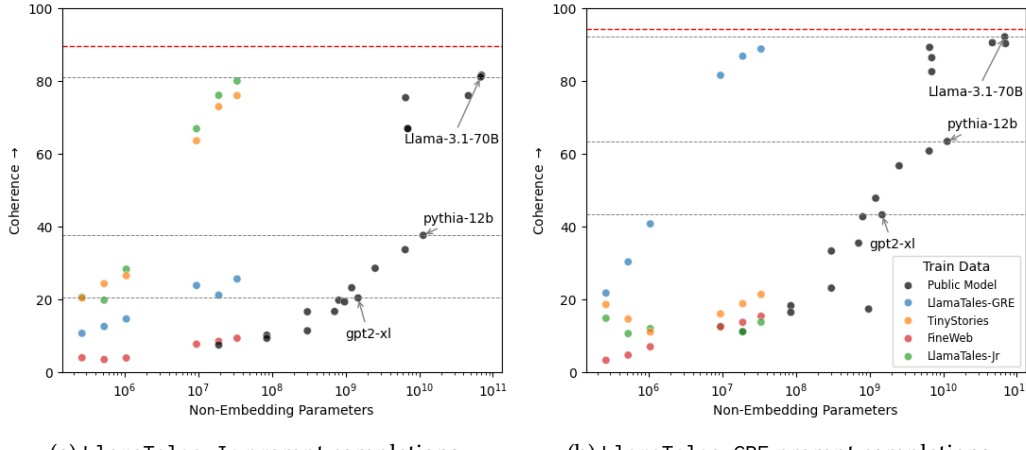

(a) `LlamaTales-Jr` prompt completions.  (b) `LlamaTales-GRE` prompt completions.

Figure 3: LLM-judged coherence scores versus model size. Colors indicate training data; black indicates public reference models. High in-distribution coherence is achievable regardless of readability, alongside strong distribution dependence. Red line: training data coherence for the respective prompt set. See Figure A2 for results for FineWeb and TinyStories.

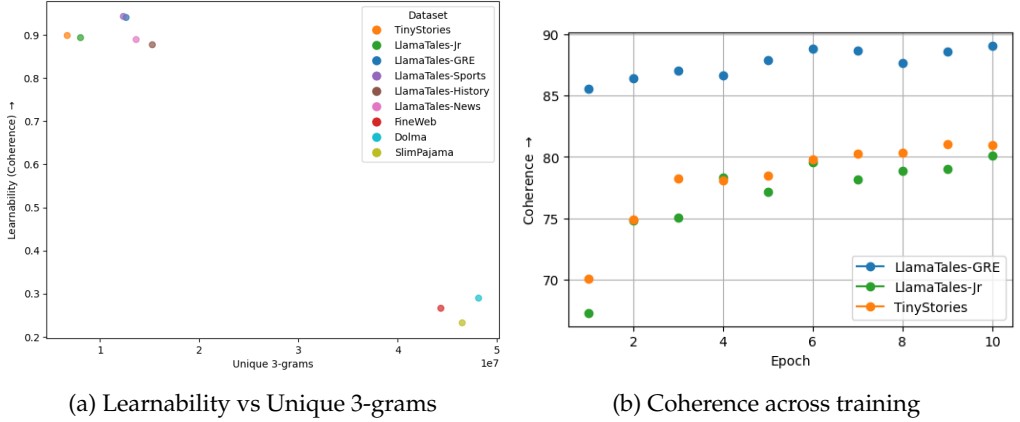

(a) Learnability vs Unique 3-grams  (b) Coherence across training

Figure 4: Statistical simplicity predicts learnability; readability does not determine learning speed. **(a)** Learnability ratio (output coherence / train data coherence) vs unique 3-grams, showing higher learnability for statistically simpler datasets (top-left). **(b)** Coherence during training. Note the rapid initial coherence emergence when training on less readable LlamaTales-GRE (blue) compared to high-readability datasets (green/orange).

A ratio near 1.0 indicates the model effectively captures the coherence present in its train data.

Figure 4a plots this learnability ratio against 3-gram diversity for models trained on various datasets. It reveals a clear inverse correlation and two distinct regimes: SLMs trained on synthetic datasets (TinyStories, LlamaTales), which have low n-gram diversity, exhibit learnability ratios close to 1.0 (top-left). In contrast, models trained on standard pretraining data, which have much higher n-gram diversity, exhibit significantly lower learnability ratios (bottom-right), struggling to capture the coherence potential of their data. This suggests statistical simplicity influences not only the speed at which coherence is learned but also the extent to which models can capture properties present in train data. Notably, this pattern is not explained by readability; LlamaTales-GRE and FineWeb have similar readability scores, yet models trained on LlamaTales-GRE achieve much higher learnability due to its lower

statistical complexity. Taken together, these results strongly suggest statistical simplicity, not developmental readability, is the more reliable predictor of learnability in SLMs.

**SLMs Trained on Synthetic Data Are Not Robust** While SLMs trained on our synthetic datasets achieve impressive in-distribution coherence, their capabilities are narrow and highly distribution-specific. When evaluated on held-out prompts from the same dataset they were trained on (e.g., LlamaTales-GRE models on LlamaTales-GRE prompts), these small models perform comparably to much larger models. However, when tested on prompts from any other dataset—including other synthetic corpora or real-world web text—their coherence degrades substantially (e.g., LlamaTales-GRE models perform poorly on LlamaTales-Jr prompts). This brittleness likely stems from the very statistical simplicity that enables high in-distribution coherence; the models learn patterns specific to the narrow training distribution and fail to generalize beyond it. The primary exception involves TinyStories and LlamaTales-Jr, which share near-identical generation processes and target audiences, resulting in some cross-dataset coherence. We did not explore training on mixed or broader synthetic distributions, which might mitigate this. Nonetheless, these findings underscore a cautionary point: high coherence in SLMs trained on narrow synthetic data can give a misleading impression of general capability. As enthusiasm grows for small models, it is crucial to distinguish in-distribution fluency from robust, generalizable understanding.

**SLMs Are Not Merely Memorizing Training Data** While the previous section highlighted the failure of these SLMs to generalize robustly outside their narrow training distribution, the question remains whether their high in-distribution coherence stems from genuine pattern learning or simply memorizing training sequences. Given the statistical simplicity of the datasets, this latter concern is particularly relevant. To assess this, we compute n-gram novelty—the proportion of n-grams in model outputs that do not appear in the training set—following Merrill et al. (2024). As shown in Figure 5, SLMs trained on LlamaTales datasets generate a substantial number of novel n-grams, particularly at mid-range lengths (e.g., 3- to 5-grams). This indicates their outputs are not simply copied but are generated by recombining learned distributional patterns. These results suggest that our models are indeed learning and recombining structure, albeit within the narrow confines of their training data, rather than merely memorizing.

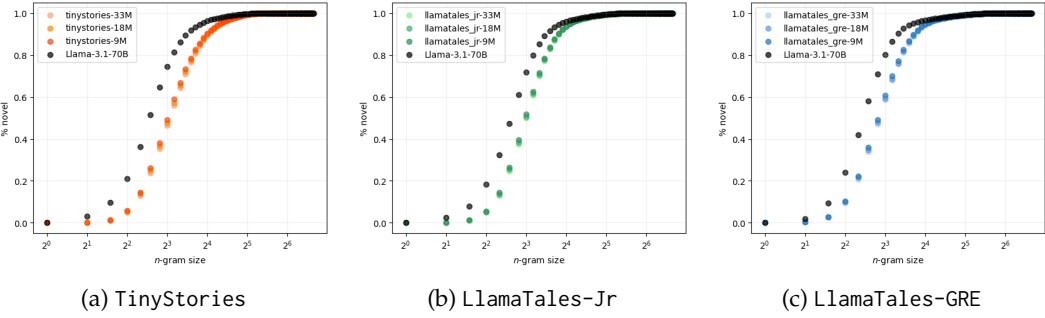

(a) TinyStories       (b) LlamaTales-Jr       (c) LlamaTales-GRE

Figure 5: N-gram novelty (percentage of generated n-grams absent in the training data) vs. n-gram size for SLMs trained on synthetic datasets. Substantial novelty indicates recombination beyond memorization. Llama-3.1-70B (black dots), serving as a baseline, was not trained on these datasets; its generations are compared against each training set here.

Synthesizing these results, we find that SLMs can learn coherent structure effectively from statistically simple data, irrespective of readability, but this learned capability remains narrow and fails to generalize robustly beyond the specific training distribution.

## 5 Discussion and Conclusion

Our results offer a key clarification regarding the emergence of coherence in small language models: human readability, often emphasized in discussions surrounding datasets like TinyStories and framed developmentally, is not the decisive factor. We find that SLMs

trained on complex, less readable text can achieve comparable or even superior coherence, sometimes learning it more rapidly than models trained on simpler, child-directed corpora. This directly challenges the intuition that human-centric linguistic simplicity inherently facilitates learning in these systems.

What, then, enables coherence? Our findings point strongly towards statistical simplicity—the predictability and structural regularity of the training data distribution—as the more reliable predictor of learnability. While we operationalize this using n-gram diversity as a proxy, this metric likely captures one facet of a broader set of properties (e.g., compressibility, structural consistency, limited long-range dependencies) that make the data statistically easier for small models to learn effectively. It is this underlying predictability, rather than surface-level readability for humans, that appears to play a central role in efficient coherence acquisition in SLMs.

Anthropomorphic metaphors—such as models 'learning like children' or synthetic datasets 'mimicking developmental input'—shape how researchers frame small model training, particularly in developmentally inspired data design. As Ibrahim & Cheng (2025) highlight, such framings influence research questions and methodologies. Often, this involves implicitly or explicitly treating characteristics associated with high human readability (e.g., accessible vocabulary, simple syntax) as the key driver of success. However, this risks conflating this human-centric simplicity with statistical learnability (the ease of modeling the data's distribution), obscuring the actual mechanisms driving capability emergence and potentially misdirecting research efforts.

This metaphorical framing reflects a broader tendency to attribute cognitive qualities to systems that lack them. As Shanahan (2023) emphasizes, language models remain statistical predictors—trained to generate likely continuations of text—not agents with access to meaning, truth conditions, or communicative intent. Using terms like "belief" or "learning" to describe their behavior risks conflating linguistic fluency with cognitive competence, inviting interpretations these systems do not warrant.

These issues are not limited to research practice. As Placani (2024) argues, anthropomorphic framing also shapes how language models are perceived in public discourse and policy. It can lead users to overestimate a model's understanding while deflecting responsibility from those who design, train, and deploy them. When models are described as learning or developing, they may appear more autonomous than they are, shifting blame for errors from the pipeline to the system. These misconceptions complicate efforts to ensure transparency, accountability, and effective governance.

Interest in developmentally inspired language model training is growing, exemplified by efforts like the BabyLM Challenge (Warstadt et al., 2023), which promotes cognitively plausible data, constraints, and interactions. These initiatives are scientifically valuable, especially when framed as intentional biomimicry or used to generate testable hypotheses. However, our findings offer a cautionary counterpoint: coherence in small models does not require child-directed inputs or simplified language, as might be assumed under a purely developmental framing. It emerges from training on synthetic corpora that are statistically simple—even when they are less readable. In this context, developmental resemblance should not be mistaken for causal explanation. We encourage future work, in line with critiques of anthropomorphic influence on methodology (Ibrahim & Cheng, 2025), to carefully distinguish between potentially useful conceptual framings (like developmental analogies) and the empirically validated mechanisms that actually support learnability.

Some may see our findings as reaffirming something already understood—that language models are statistical learners, and that low-complexity data is easier to learn from. But the continued presence of readability-oriented framing, especially in work on developmentally inspired datasets like TinyStories, suggests that this clarification remains timely. As Ibrahim & Cheng (2025) note, anthropomorphic language is increasingly common, influencing both research narratives and methodological choices. In this context, reaffirming statistical fundamentals helps restore clarity. We do not reject metaphor altogether—developmental analogies can guide intuition—but they must be distinguished from empirical explanations.

Framing model behavior in developmental terms risks obscuring the actual factors that support learning.

This statistically grounded perspective points towards productive future research. Key questions include developing richer, more comprehensive measures of dataset complexity and learnability beyond simple n-grams, designing training curricula or data selection strategies that leverage statistical simplicity without sacrificing generalization, and creating evaluation methods that effectively disentangle genuine capabilities from pattern matching facilitated by statistically simple data structures.

In conclusion, we propose a reframing: the emergence of coherence in small models trained on specific datasets is not evidence of achieving a human-like developmental milestone. Rather, it is primarily a statistical outcome reflecting the model's success in learning patterns from distributionally simple and coherent data. Recognizing this distinction is important for accurately understanding SLM capabilities and for aligning model design, research framing, and public interpretation with the mechanisms that actually govern their behavior.

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

# A   Classic Readability Formulas

The readability formulas utilized in Table 1 are presented below. The simplest way to measure readability is through formulas, many of which have been developed over the years. These formulas are generally straightforward, focusing on various combinations of word, sentence, and syllable counts. We employ `textstat`[2] to calculate readability using various established formulas, including FKGL.

### A.1   Flesch-Kincaid Grade Level (Kincaid et al., 1975)

$$\text{FKGL} = 0.39 \left( \frac{\text{words}}{\text{sentences}} \right) + 11.8 \left( \frac{\text{syllables}}{\text{words}} \right) - 15.59 \tag{1}$$

### A.2   Automated Readability Index (Smith & Senter, 1967)

$$\text{ARI} = 4.71 \left( \frac{\text{characters}}{\text{words}} \right) + 0.5 \left( \frac{\text{words}}{\text{sentences}} \right) - 21.43 \tag{2}$$

### A.3   Coleman–Liau Index (Coleman & Liau, 1975)

$$\text{CLI} = 0.0588 \left( \frac{\text{characters}}{\text{words}} \times 100 \right) - 0.296 \left( \frac{\text{sentences}}{\text{words}} \times 100 \right) - 15.8 \tag{3}$$

### A.4   Dale–Chall Formula (Dale & Chall, 1948)

$$\text{DC} = 0.1579 \left( \frac{\text{difficult words}}{\text{words}} \times 100 \right) + 0.0496 \left( \frac{\text{words}}{\text{sentences}} \right) \tag{4}$$

Difficult words are defined as those not included in a list of 3,000 words that fourth-grade American students are expected to know. If the percentage of difficult words exceeds 5%, add 3.6365 to the score.

### A.5   Gunning Fog Index (Gunning, 1968)

$$\text{GFI} = 0.4 \left[ \left( \frac{\text{words}}{\text{sentences}} \right) + 100 \left( \frac{\text{complex words}}{\text{words}} \right) \right] \tag{5}$$

Complex words are defined as words with three or more syllables, excluding proper nouns, familiar jargon, and compound words.

### A.6   Linsear Write Formula (O'Hayre, 1975)

$$r = \frac{(\text{words} \leq 2 \text{ syllables}) + 3 \cdot (\text{words} \geq 3 \text{ syllables})}{\text{sentences}} \tag{6}$$

$$\text{Linsear Write} = \begin{cases} \frac{r}{2} & \text{if } r > 20 \\ \frac{r-2}{2} & \text{if } r \leq 20 \end{cases} \tag{7}$$

---

[2]https://github.com/textstat/textstat

### A.7  SMOG Index (Harry & Laughlin, 1969)

$$\text{SMOG} = 1.0430 \sqrt{30 \left( \frac{\text{words} \geq 3 \text{ syllables}}{\text{sentences}} \right)} + 3.1291 \tag{8}$$

### A.8  Spache Formula (Spache, 1953)

$$\text{Spache} = 0.121 \left( \frac{\text{words}}{\text{sentences}} \right) + 0.082 \left( \frac{\text{difficult words}}{\text{words}} \times 100 \right) + 0.659 \tag{9}$$

Difficult words are defined as words that are not included in a list of familiar words that are typically known by fourth-grade students.

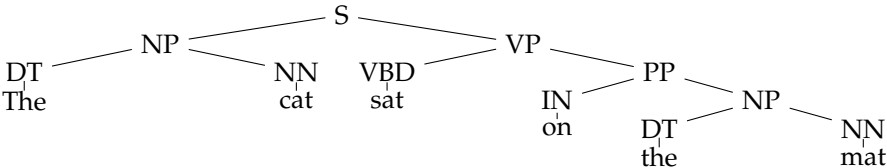

Figure A1: Constituency parse tree for the sentence "The cat sat on the mat."

## B  Additional Figures and Tables

| Dataset | Coherence | Readability |
|---|---|---|
| TinyStories | 90.1 | 92.6 |
| LlamaTales-Jr | 89.5 | 92.7 |
| LlamaTales-GRE | 94.4 | 72.7 |
| LlamaTales-Sports | 92.4 | 72.4 |
| LlamaTales-News | 94.5 | 72.7 |
| LlamaTales-History | 91.0 | 61.4 |
| FineWeb | 77.6 | 68.2 |
| Dolma | 60.2 | 70.99 |
| SlimPajama | 67.3 | 66.6 |

Table A1: Coherence and readability scores for train splits.

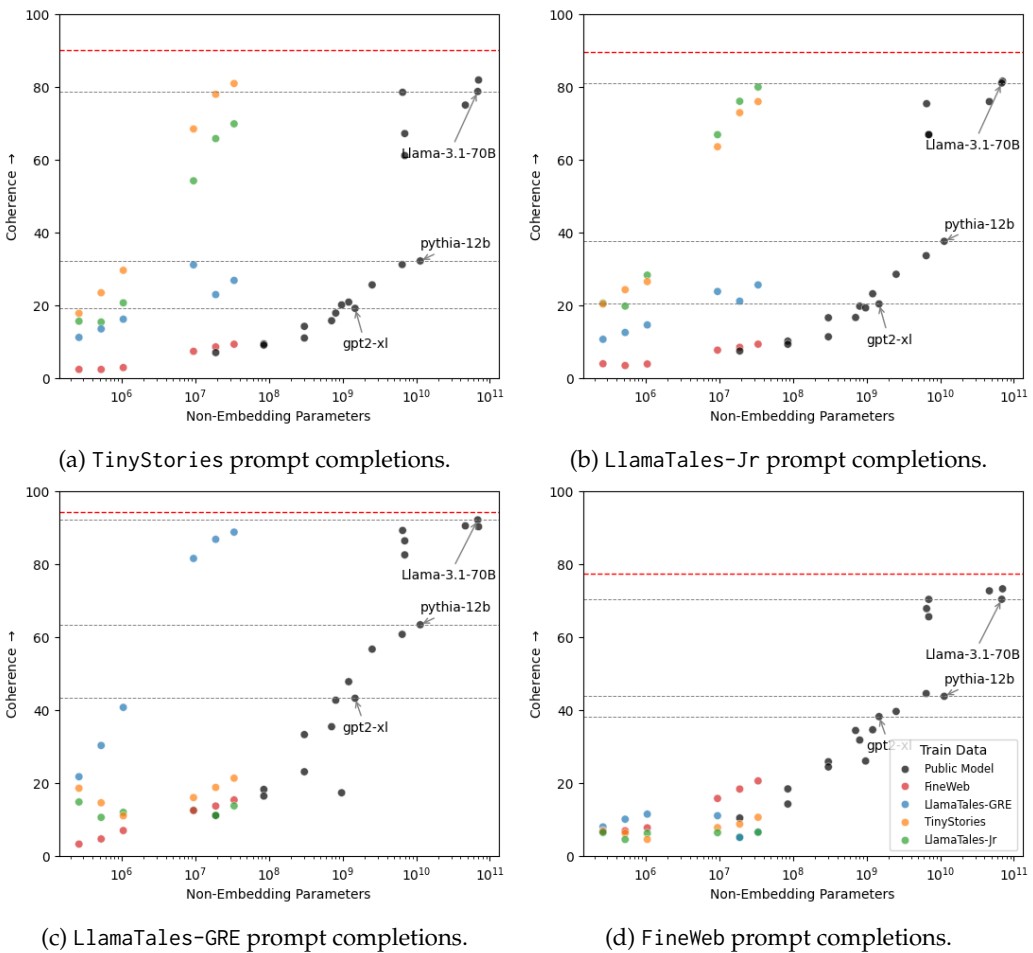

(a) `TinyStories` prompt completions.

(b) `LlamaTales-Jr` prompt completions.

(c) `LlamaTales-GRE` prompt completions.

(d) `FineWeb` prompt completions.

Figure A2: **Coherence** of text generated by the LMs listed in Table A2. Prompts are extracted from the test splits of our datasets in Table 1. The legend colors represent the training data for each model. Public models (black) are found on Huggingface and are not fine-tuned on our data. The red horizontal line marks the coherence of the train split of the dataset in focus. Return to Figure 3 (truncated results).

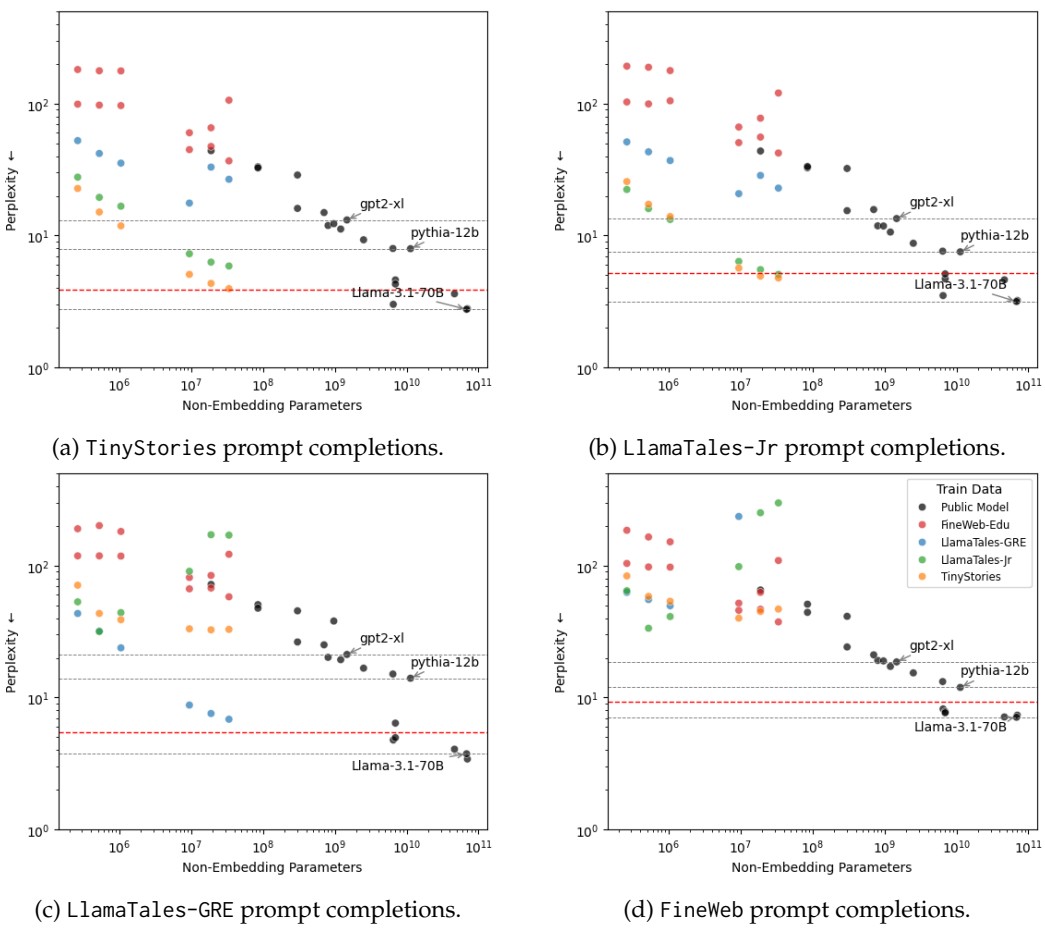

(a) `TinyStories` prompt completions.

(b) `LlamaTales-Jr` prompt completions.

(c) `LlamaTales-GRE` prompt completions.

(d) `FineWeb` prompt completions.

Figure A3: **Perplexity** (computed by external LMs) of text generated by the LMs listed in Table A2, based on prompts from the test split data in Table 1. See Section 3.2 for details on this metric. The legend colors represent the training data for each model. The red horizontal line marks the perplexity of the train split of the dataset in focus. Return to Section 4 (Results).

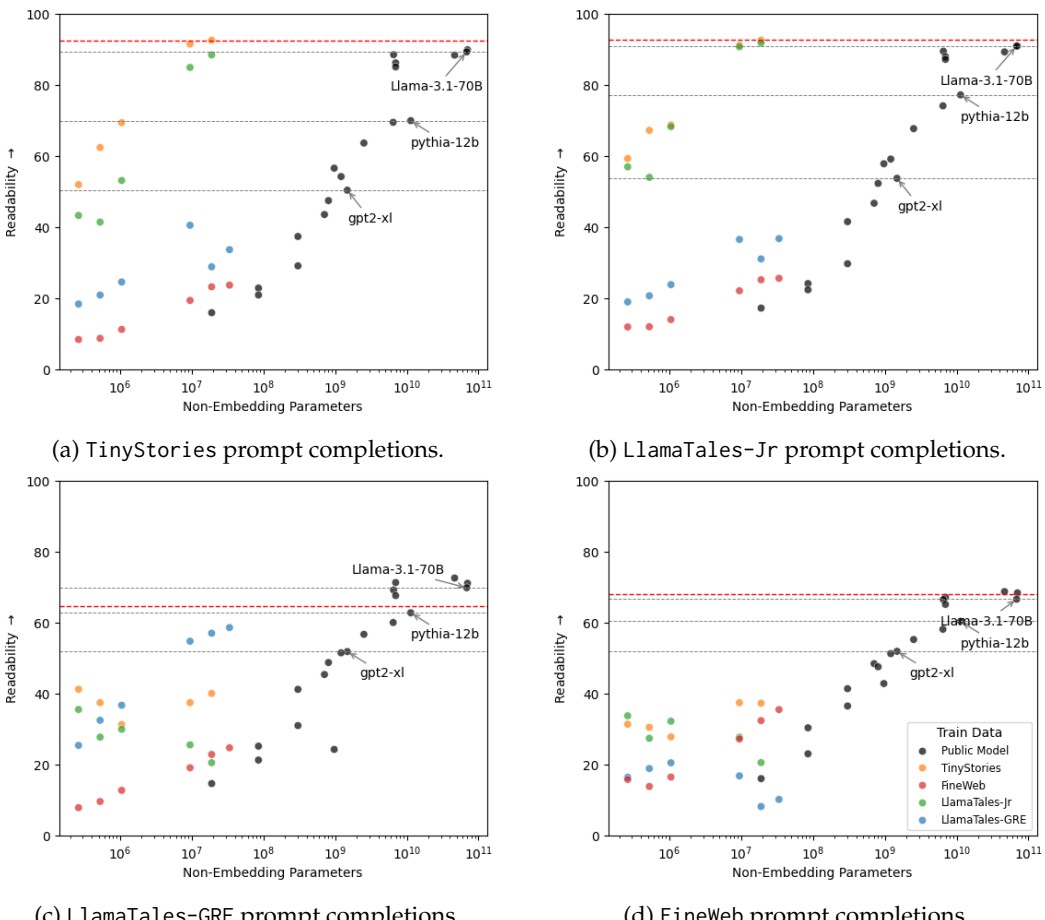

(a) `TinyStories` prompt completions.

(b) `LlamaTales-Jr` prompt completions.

(c) `LlamaTales-GRE` prompt completions.

(d) `FineWeb` prompt completions.

Figure A4: **Readability** of text generated by LMs in Table A2, based on prompts from the data (test split) in Table 1. The legend colors represent the training data for each model. The red horizontal line marks the readability of the train split of the dataset in focus. Return to Section 4 (Results).

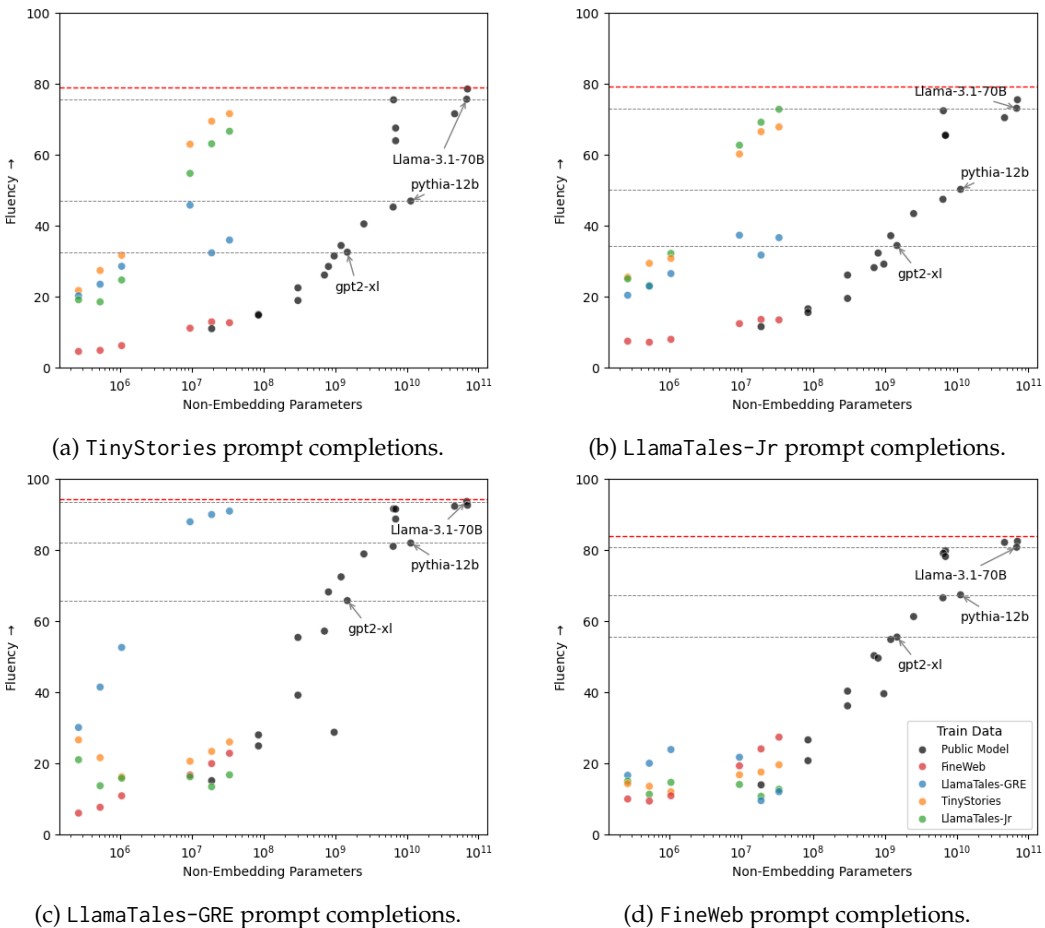

(a) `TinyStories` prompt completions.

(b) `LlamaTales-Jr` prompt completions.

(c) `LlamaTales-GRE` prompt completions.

(d) `FineWeb` prompt completions.

Figure A5: **Fluency** of text generated by LMs in Table A2, based on prompts from the data (test split) in Table 1. The legend colors represent the training data for each model. The red horizontal line marks the fluency of the train split of the dataset in focus.

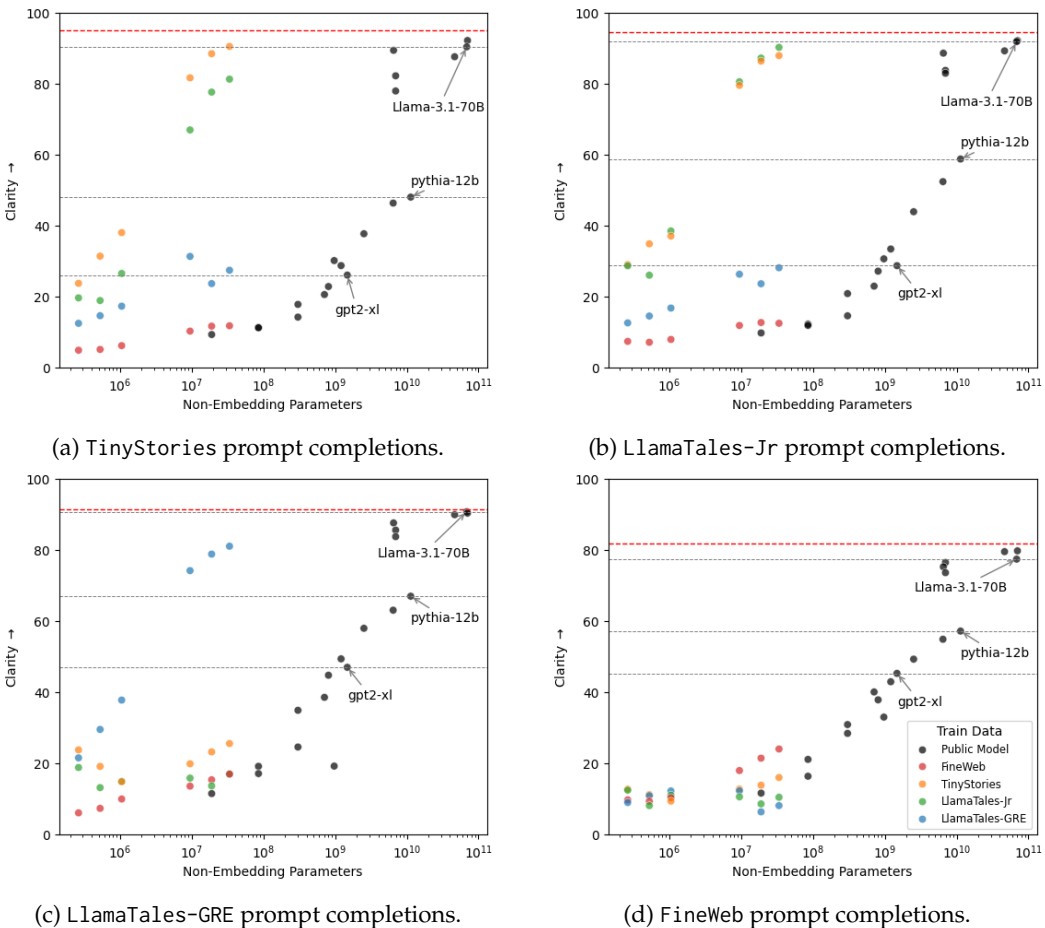

(a) `TinyStories` prompt completions.

(b) `LlamaTales-Jr` prompt completions.

(c) `LlamaTales-GRE` prompt completions.

(d) `FineWeb` prompt completions.

Figure A6: **Clarity** of text generated by LMs in Table A2, based on prompts from the data (test split) in Table 1. The legend colors represent the training data for each model. The red horizontal line marks the clarity of the train split of the dataset in focus.

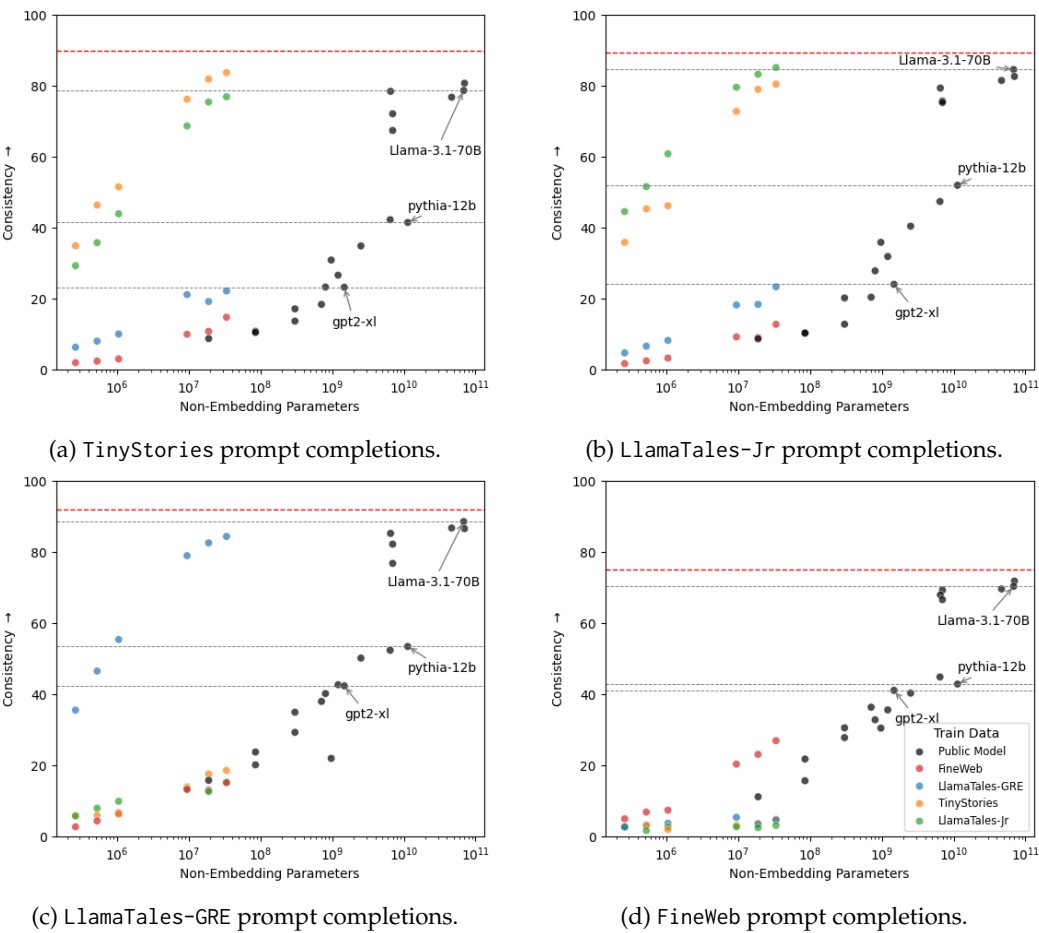

(a) `TinyStories` prompt completions.

(b) `LlamaTales-Jr` prompt completions.

(c) `LlamaTales-GRE` prompt completions.

(d) `FineWeb` prompt completions.

Figure A7: **Consistency** of text generated by LMs in Table A2, based on prompts from the data (test split) in Table 1. The legend colors represent the training data for each model. The red horizontal line marks the consistency of the train split of the dataset in focus.

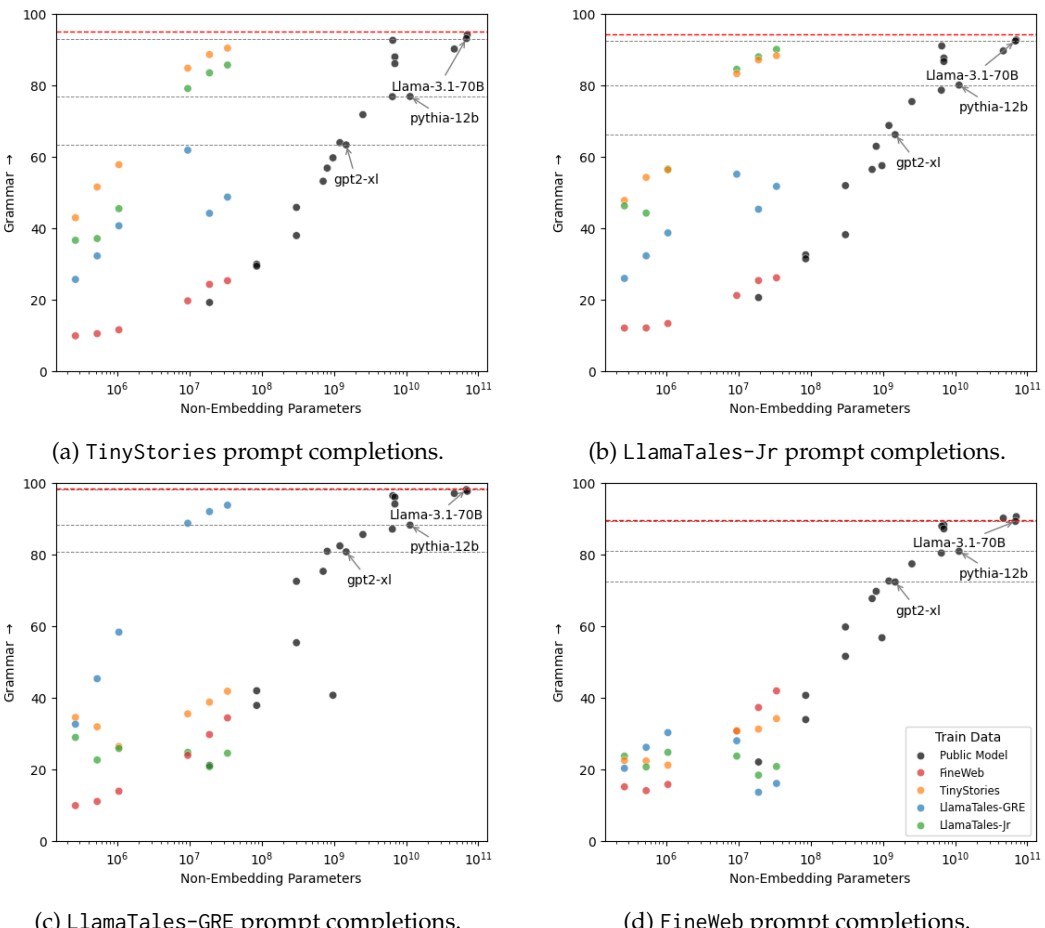

(a) `TinyStories` prompt completions.

(b) `LlamaTales-Jr` prompt completions.

(c) `LlamaTales-GRE` prompt completions.

(d) `FineWeb` prompt completions.

Figure A8: **Grammar** of text generated by LMs in Table A2, based on prompts from the data (test split) in Table 1. The legend colors represent the training data for each model. The red horizontal line marks the grammaticality of the train split of the dataset in focus.

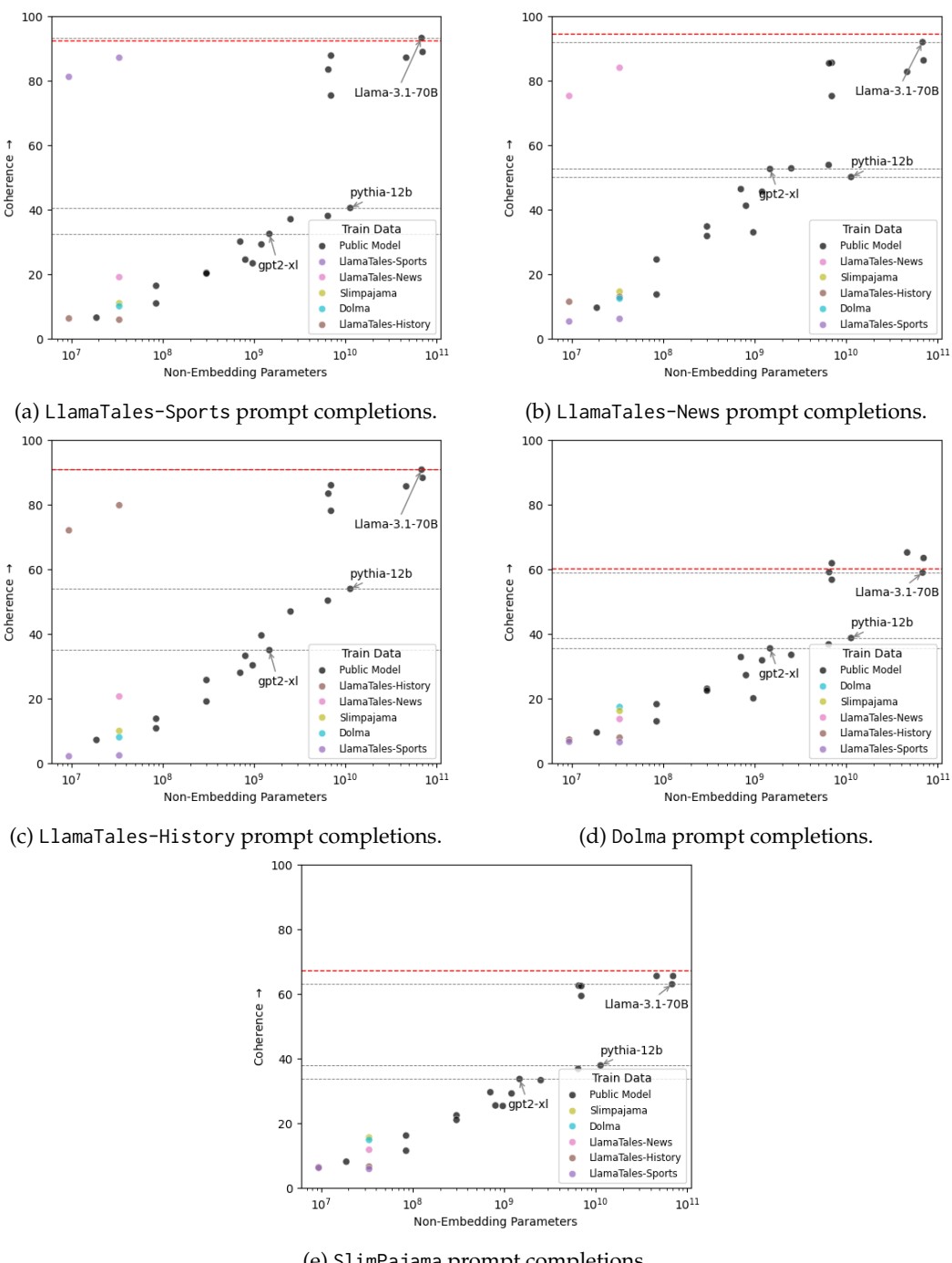

Figure A9: **Coherence** of text generated by LMs in Table A2, based on prompts from the data (test split) in Table 1. The legend colors represent the training data for each model. The red horizontal line marks the coherence of the train split of the dataset in focus.

**System Prompt:** You are an experienced teacher, skilled at identifying the readability of different texts.
**User:** Read the text below. Then, indicate the readability of the text, on a scale from 1 (extremely challenging to understand) to 100 (very easy to read and understand). In your assessment, consider factors such as sentence structure, vocabulary complexity, and overall clarity.
<Text></Text>
On a scale from 1 (extremely challenging to understand) to 100 (very easy to read and understand), how readable is this text?. Please answer with a single number.

Figure A10: Prompt used for LLM-as-a-Judge to evaluate readability. See Section 3.1.

**System Prompt:** You are an experienced teacher, skilled at identifying the coherence of different texts.
**User:** Read the text below. Then, indicate the coherence of the text, on a scale from 1 (extremely incoherent) to 100 (very coherent). Remember that coherent text should be well-structured and well-organized. Coherent text should not just be a heap of related information, but should build from sentence to sentence.
<Text></Text>
On a scale from 1 (extremely incoherent) to 100 (very coherent), how coherent is this text?. Please answer with a single number.

Figure A11: Prompt used for LLM-as-a-Judge to evaluate coherence. See Section 3.2

**System Prompt:** You are an experienced teacher, skilled at identifying the coherence of different texts.
**User:** Read the text below and evaluate the coherence of the text. Remember that coherent text should be well-structured and well-organized. Coherent text should not just be a heap of related information, but should build from sentence to sentence.
<Text></Text>
Please provide a short analysis of the text's coherence. After your analysis, on a scale from 1 (extremely incoherent) to 100 (very coherent), how coherent is this text? Please answer with a single number.

Figure A12: Prompt used for LLM-as-a-Judge to evaluate coherence, instructing the LM to generate an analysis before generating a score.

**System Prompt:** You are an experienced teacher, skilled at identifying the coherence of different texts.
**User:** First, consider the following examples:

Positive Example (Very Coherent):

> The process of photosynthesis is essential for plant life. It begins when sunlight is absorbed by chlorophyll in the leaves. This energy is then used to convert carbon dioxide and water into glucose and oxygen. The glucose provides energy for the plant, while the oxygen is released into the atmosphere.

This text is coherent because it is well-structured, with each sentence building on the previous one to explain a process clearly and logically.
Negative Example (Incoherent):

> Photosynthesis is a process. Leaves are green. Oxygen is in the air. Plants need water. Sunlight is bright.

This text is incoherent because it lacks logical flow and structure, presenting disjointed facts without clear connections or progression.

Now, read the text below and evaluate its coherence on a scale from 1 (extremely incoherent) to 100 (very coherent). Remember that coherent text should be well-structured and well-organized, not just a heap of related information.
`<Text></Text>`
Please provide a short analysis of the text's coherence. After your analysis, on a scale from 1 (extremely incoherent) to 100 (very coherent), how coherent is this text? Please answer with a single number.

Figure A13: Prompt used for LLM-as-a-Judge to evaluate coherence. This version includes positive and negative examples for reference.

**System Prompt:** You are an experienced teacher, skilled at identifying the readability of different texts.
**User:** Read the text below and evaluate the readability of the text. In your assessment, consider factors such as sentence structure, vocabulary complexity, and overall clarity.
`<Text></Text>`
Please provide a short analysis of the text's readability. After your analysis, on a scale from 1 (extremely challenging to understand) to 100 (very easy to read and understand), how readable is this text? Please answer with a single number.

Figure A14: Prompt used for LLM-as-a-Judge to evaluate readability, instructing the LM to generate an analysis before generating a score.

**System Prompt:** You are an experienced teacher, skilled at identifying the readability of different texts.
**User:** First, consider the following examples:

Positive Example (Very Readable):

> The cat sat on the mat. It was a sunny day, and the cat enjoyed the warmth. The mat was soft and comfortable, making it the perfect spot for a nap.

This text is easy to read because it uses simple sentence structures, familiar vocabulary, and conveys ideas clearly.
Negative Example (Challenging to Read):

> In the midst of the diurnal cycle, the feline quadruped positioned itself upon the textile floor covering, basking in the solar radiance, which permeated the atmosphere with thermal energy, rendering the environment conducive to somnolence.

This text is challenging to read due to complex sentence structures, advanced vocabulary, and convoluted expression of ideas.

Now, read the text below and evaluate its readability on a scale from 1 (extremely challenging to understand) to 100 (very easy to read and understand). In your assessment, consider factors such as sentence structure, vocabulary complexity, and overall clarity.
`<Text></Text>`
On a scale from 1 (extremely challenging to understand) to 100 (very easy to read and understand), how readable is this text?. Please answer with a single number.

Figure A15: Prompt used for LLM-as-a-Judge to evaluate readability. This version includes positive and negative examples for reference.

**System Prompt:** You are an experienced teacher, skilled at identifying grammatical errors of different texts.
**User:** Read the text below. Then, indicate the grammaticality of the text on a scale from 1 (extremely ungrammatical) to 100 (perfectly grammatical). In your assessment, consider factors such as spelling, part of speech, sentence structure, punctuation, and overall grammatical correctness.
`<Text></Text>`
On a scale from 1 (extremely ungrammatical) to 100 (perfectly grammatical), how grammatical is this text?. Please answer with a single number.

Figure A16: Prompt used for LLM-as-a-Judge to evaluate grammaticality.

**System Prompt:** You are an experienced linguist, skilled at evaluating the fluency of different texts.
**User:** Read the text below. Then, indicate the fluency of the text, on a scale from 1 (poor fluency) to 100 (excellent fluency). In your assessment, consider factors such as grammatical correctness, naturalness of language, and overall smoothness.
`<Text></Text>`
On a scale from 1 (poor fluency) to 100 (excellent fluency), how fluent is this text?. Please answer with a single number.

Figure A17: Prompt used for LLM-as-a-Judge to evaluate fluency.

**System Prompt:** You are an experienced teacher, skilled at identifying the consistency of different texts.
**User:** Read the text below. Then, evaluate how consistent the first two sentences are with the rest of the text, on a scale from 1 (extremely inconsistent) to 100 (very consistent). Consistent text should maintain a logical flow and alignment in terms of theme, tone, and information throughout.
`<Text></Text>`
On a scale from 1 (extremely inconsistent) to 100 (very consistent), how consistent are the first two sentences of this text with the rest of the text? Please answer with a single number.

Figure A18: Prompt used for LLM-as-a-Judge to evaluate consistency.

**System Prompt:** You are an experienced teacher, skilled at identifying the clarity of different texts.
**User:** Read the text below. Then, indicate the clarity of the text, on a scale from 1 (not clear at all) to 100 (extremely clear). In your assessment, consider factors such as coherence, conciseness, and comprehensibility.
`<Text></Text>`
On a scale from 1 (not clear at all) to 100 (extremely clear), how clear is this text? Please answer with a single number.

Figure A19: Prompt used for LLM-as-a-Judge to evaluate clarity.

**System Prompt:** You are a celebrated children's author. You write stories that are both easy to read and grammatically correct.
**User:** Write a short story (3-5 paragraphs) which only uses simple words that a 5 year old child would understand. The story should use the words: `<WORD-1>`, `<WORD-2>`, and `<WORD-3>`. The story has the following features: `<FEAT-1>` ... `<FEAT-K>`

Figure A20: Prompt used to generate `LlamaTales-Jr`. See Section 2.

**System Prompt:** You are a renowned fiction writer, celebrated for your imaginative storytelling and compelling characters. Your work spans various genres, including fantasy, science fiction, and contemporary fiction, and is known for its vivid descriptions, intricate plots, and emotional depth. Your writing is best appreciated by readers with the vocabulary and comprehension expected of a college graduate.
**User:** Write a short story (3-5 paragraphs). The story should use the words: `<WORD-1>`, `<WORD-2>`, and `<WORD-3>`. The story has the following features: `<FEAT-1>` ... `<FEAT-K>`

Figure A21: Prompt used to generate `LlamaTales-GRE`. See Section 2.

**System Prompt:** You are a distinguished historian, celebrated for your meticulous research and engaging narratives. Your work spans various historical periods and is known for its depth, accuracy, and insightful analysis. You have a talent for bringing history to life, making complex events and figures accessible and compelling to a broad audience. Your writing is best appreciated by readers with a keen interest in history and a desire to understand the past in a nuanced and comprehensive manner.
**User:** Write a short historical article (3-5 paragraphs) that provides an insightful analysis of a significant event or figure. Include key details, context, and the impact on subsequent history. The story should use the words: `<WORD-1>`, `<WORD-2>`, and `<WORD-3>`. The story has the following features: `<FEAT-1>` ... `<FEAT-K>`

Figure A22: Prompt used to generate `LlamaTales-History`.

**System Prompt:** You are an experienced sports journalist known for your vivid and engaging coverage of athletic events and athletes' stories. Your writing captures the excitement, drama, and human elements of sports, appealing to both die-hard fans and casual readers. You have a keen eye for detail, a deep understanding of various sports, and the ability to convey complex strategies and statistics in an accessible manner. Your articles are characterized by their dynamic prose, insightful analysis, and ability to place sporting events within broader cultural and social contexts.
**User:** Write a short sports article (3-5 paragraphs) about a recent game, match, or athletic performance. Include vivid descriptions, key statistics, and quotes from players or coaches if applicable. The story should use the words: <WORD-1>, <WORD-2>, and <WORD-3>. The story has the following features: <FEAT-1> ... <FEAT-K>

Figure A23: Prompt used to generate `LlamaTales-Sports`.

**System Prompt:** You are a seasoned journalist at The New York Times, known for your incisive reporting and compelling storytelling. Your work covers a wide range of topics, from breaking news and investigative journalism to in-depth features and opinion pieces. You have a keen eye for detail, a commitment to accuracy, and the ability to convey complex issues in a clear and engaging manner. Your writing is characterized by its clarity, depth, and ability to inform and engage a diverse readership.
**User:** Write a concise news article (3-5 paragraphs) about a recent significant event. Include key details, quotes from relevant sources, and the broader context of the event. The story should use the words: <WORD-1>, <WORD-2>, and <WORD-3>. The story has the following features: <FEAT-1> ... <FEAT-K>

Figure A24: Prompt used to generate `LlamaTales-News`.

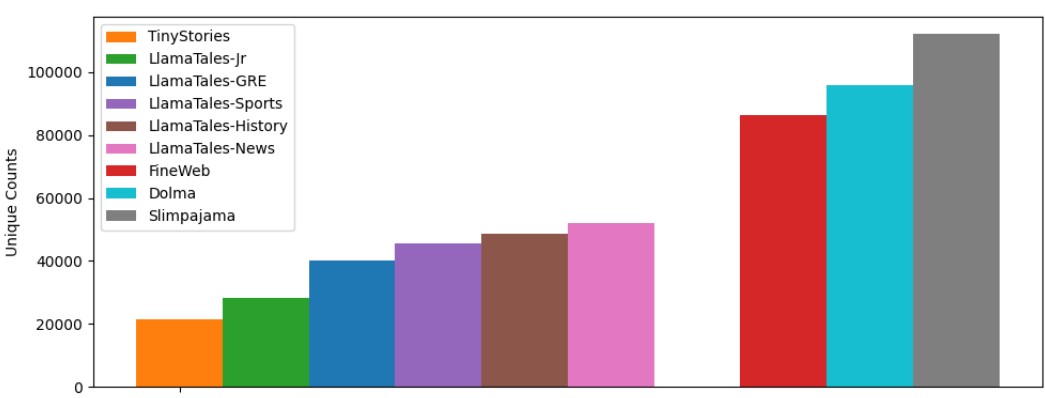

Figure A25: Unique unigram counts for 100M-token samples from each dataset. Refer to Figure 2 for higher values of $n$.

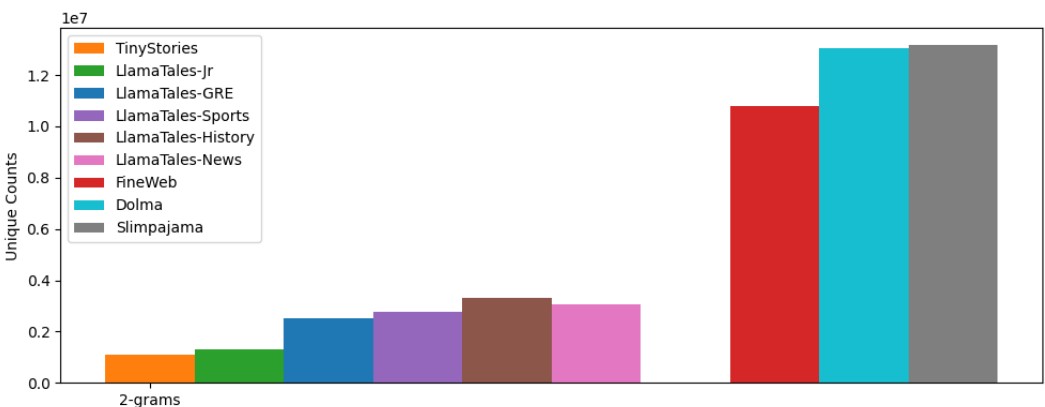

Figure A26: Unique bigram counts for 100M-token samples from each dataset. Refer to Figure 2 for higher values of $n$.

Table A2: Overview of the models used in our experiments. Example generations from each model are shown in Tables A6 to A20. **Top:** Models trained from scratch on the dataset indicated by each model's prefix. **Bottom:** Pretrained models sourced from Huggingface. Return to Section 4.

| Model | Parameters | Train Data | Train Tokens | Layers | Heads | Model Dim |
|---|---|---|---|---|---|---|
| tinystories-262K | 2.63e+05 | TinyStories | 1e10 | 1 | 2 | 128 |
| tinystories-524K | 5.25e+05 | TinyStories | 1e10 | 2 | 2 | 128 |
| tinystories-1M | 1.05e+06 | TinyStories | 1e10 | 4 | 2 | 128 |
| tinystories-9M | 9.44e+06 | TinyStories | 1e10 | 4 | 6 | 384 |
| tinystories-18M | 1.89e+07 | TinyStories | 1e10 | 8 | 6 | 384 |
| tinystories-33M | 3.36e+07 | TinyStories | 1e10 | 8 | 8 | 512 |
| llamatales_jr-262K | 2.63e+05 | LlamaTales-Jr | 1e10 | 1 | 2 | 128 |
| llamatales_jr-524K | 5.25e+05 | LlamaTales-Jr | 1e10 | 2 | 2 | 128 |
| llamatales_jr-1M | 1.05e+06 | LlamaTales-Jr | 1e10 | 4 | 2 | 128 |
| llamatales_jr-9M | 9.44e+06 | LlamaTales-Jr | 1e10 | 4 | 6 | 384 |
| llamatales_jr-18M | 1.89e+07 | LlamaTales-Jr | 1e10 | 8 | 6 | 384 |
| llamatales_jr-33M | 3.36e+07 | LlamaTales-Jr | 1e10 | 8 | 8 | 512 |
| llamatales_gre-262K | 2.63e+05 | LlamaTales-GRE | 1e10 | 1 | 2 | 128 |
| llamatales_gre-524K | 5.25e+05 | LlamaTales-GRE | 1e10 | 2 | 2 | 128 |
| llamatales_gre-1M | 1.05e+06 | LlamaTales-GRE | 1e10 | 4 | 2 | 128 |
| llamatales_gre-9M | 9.44e+06 | LlamaTales-GRE | 1e10 | 4 | 6 | 384 |
| llamatales_gre-18M | 1.89e+07 | LlamaTales-GRE | 1e10 | 8 | 6 | 384 |
| llamatales_gre-33M | 3.36e+07 | LlamaTales-GRE | 1e10 | 8 | 8 | 512 |
| fineweb-262K | 2.63e+05 | FineWeb | 1e10 | 1 | 2 | 128 |
| fineweb-524K | 5.25e+05 | FineWeb | 1e10 | 2 | 2 | 128 |
| fineweb-1M | 1.05e+06 | FineWeb | 1e10 | 4 | 2 | 128 |
| fineweb-9M | 9.44e+06 | FineWeb | 1e10 | 4 | 6 | 384 |
| fineweb-18M | 1.89e+07 | FineWeb | 1e10 | 8 | 6 | 384 |
| fineweb-33M | 3.36e+07 | FineWeb | 1e10 | 8 | 8 | 512 |
| gpt2 | 8.51e+07 | | | 12 | 12 | 768 |
| gpt2-medium | 3.02e+08 | | | 24 | 16 | 1024 |
| gpt2-large | 7.08e+08 | | | 36 | 20 | 1280 |
| gpt2-xl | 1.48e+09 | | | 48 | 25 | 1600 |
| pythia-70m | 1.89e+07 | The Pile | 3e11 | 6 | 8 | 512 |
| pythia-160m | 8.51e+07 | The Pile | 3e11 | 12 | 12 | 768 |
| pythia-410m | 3.02e+08 | The Pile | 3e11 | 24 | 16 | 1024 |
| pythia-1b | 8.06e+08 | The Pile | 3e11 | 16 | 8 | 2048 |
| pythia-1.4b | 1.21e+09 | The Pile | 3e11 | 24 | 16 | 2048 |
| pythia-2.8b | 2.52e+09 | The Pile | 3e11 | 32 | 32 | 2560 |
| pythia-6.9b | 6.44e+09 | The Pile | 3e11 | 32 | 32 | 4096 |
| pythia-12b | 1.13e+10 | The Pile | 3e11 | 36 | 40 | 5120 |
| TinyLlama_v1.1 | 9.69e+08 | SlimPajama | 2e12 | 22 | 32 | 2048 |
| Mistral-7B-v0.3 | 6.98e+09 | | | 32 | 32 | 4096 |
| Mixtral-8x7B-v0.1 | 4.64e+10 | | | 32 | 32 | 4096 |
| Qwen2-7B | 6.53e+09 | | 7e12 | 28 | 28 | 3584 |
| Qwen2-72B | 7.02e+10 | | 7e12 | 80 | 64 | 8192 |
| Llama-3.1-8B | 6.98e+09 | | 15e12 | 32 | 32 | 4096 |
| Llama-3.1-70B | 6.85e+10 | | 15e12 | 80 | 64 | 8192 |

Table A3: Statistics for the train splits of the datasets described in Section 2 as well as (absolute) Pearson **corr**elation coefficients against human judgments of readability (↑ is better). Random examples from each dataset are shown in Table A5. **Top:** Classic readability formulas (↓ is easier to read). **Mid-Top:** Statistics computed from running a constituency parser over the sentences of each dataset (↑ suggests more grammatical complexity). **Mid-Bot:** The result of prompting `Llama-3.1-70B-Instruct` to judge readability and coherence (↑ is better). Perplexity is computed with `Llama-3.1-8B`, `Qwen2-7B`, and `Mistral-7B-v0.3` and averaged. **Bot:** Word, token, and syllable level statistics. Truncated statistics are shown in Table 1.

| | Corr. | TinyStories | LlamaTales-Jr | LlamaTales-GRE | FineWeb |
|---|---|---|---|---|---|
| **Automated Readability** | 0.47 | 2.9 | 2.9 | 12.4 | 13.1 |
| **Coleman–Liau** | 0.48 | 3.7 | 3.8 | 10.4 | 11.8 |
| **Dale–Chall** | 0.58 | 5.7 | 5.7 | 9.1 | 9.3 |
| **Flesch–Kincaid** | 0.49 | 2.4 | 2.2 | 9.6 | 10.7 |
| **Gunning Fog** | 0.50 | 4.6 | 3.8 | 11.7 | 12.1 |
| **Linsear Write** | 0.41 | 4.2 | 3.3 | 13.2 | 12.7 |
| **SMOG** | 0.53 | 5.7 | 5.4 | 11.3 | 12.6 |
| **Spache Readability** | 0.51 | 2.7 | 2.5 | 5.5 | 5.5 |
| **Depth / Sentence** | 0.34 | 6.8 | 6.4 | 10.6 | 9.5 |
| **Width / Sentence** | 0.34 | 5.1 | 4.7 | 8.0 | 7.5 |
| **Nodes / Sentence** | 0.36 | 19.6 | 17.2 | 42.1 | 37.8 |
| **Readability** | **0.74** | 92.6 | 92.7 | 64.8 | 68.2 |
| **Coherence** | 0.03 | 90.1 | 89.5 | 94.4 | 77.4 |
| **Perplexity** | 0.30 | 3.9 | 5.2 | 5.5 | 9.3 |
| **Tokens / Document** | 0.15 | 186.5 | 282.8 | 500.6 | 497.3 |
| **Syllables / Document** | 0.43 | 180.2 | 270.7 | 561.6 | 603.1 |
| **Words / Document** | 0.10 | 152.9 | 222.6 | 392.2 | 386.4 |
| **Unique Words** | | 6.4e4 | 1.5e5 | 3.2e5 | 5.1e6 |
| **Unique 1-grams (token)** | | 3.20e4 | 4.35e4 | 5.21e4 | 1.09e5 |
| **Unique 2-grams** | | 2.89e6 | 3.71e6 | 7.90e6 | 4.26e7 |
| **Unique 4-grams** | | 8.75e7 | 1.10e8 | 1.61e8 | 5.74e8 |
| **Unique 8-grams** | | 5.02e8 | 6.48e8 | 6.88e8 | 9.43e8 |
| **Total Documents** | | 4.9e6 | 3.6e6 | 2.0e6 | 2.0e6 |
| **Total Tokens** | | 9.2e8 | 1e9 | 1e9 | 1e9 |
| **Synthetic** | | Yes | Yes | Yes | No |
| **Source** | | GPT-3.5/4 | Llama-3.1-8B | Llama-3.1-8B | Web |

Table A4: Our ranking of open-weight LMs. See Section 3.2.

| Model | Parameters | Rank |
|---|---|---|
| pythia-70m | 1.89e+07 | 11 |
| pythia-160m | 8.51e+07 | 10 |
| pythia-410m | 3.02e+08 | 9 |
| pythia-1b | 8.06e+08 | 8 |
| pythia-1.4b | 1.21e+09 | 7 |
| pythia-2.8b | 2.52e+09 | 6 |
| pythia-6.9b | 6.44e+09 | 5 |
| pythia-12b | 1.13e+10 | 4 |
| Mistral-7B-v0.3 | 6.98e+09 | 3 |
| Qwen2-7B | 6.53e+09 | 3 |
| Llama-3.1-8B | 6.98e+09 | 3 |
| Mixtral-8x7B-v0.1 | 4.64e+10 | 2 |
| Qwen2-72B | 7.02e+10 | 1 |
| Llama-3.1-70B | 6.85e+10 | 1 |

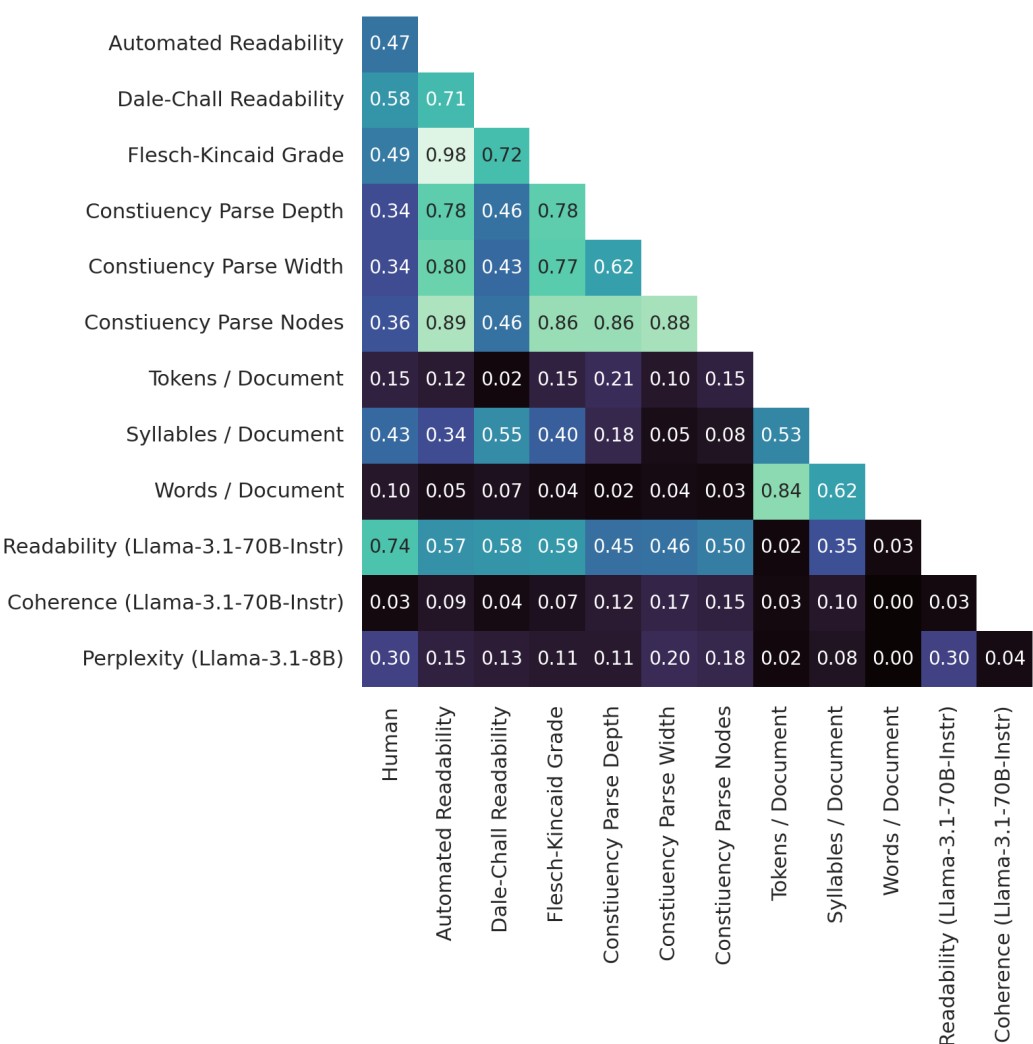

Figure A27: Pearson correlation coefficients among various measures on the CLEAR dataset. Instructing an LLM to judge readability shows the highest correlation with human judgments of readability. However, instructing an LLM to judge coherence does not correlate with readability, and perplexity only shows a weak correlation with readability. These findings suggest that there is a distinction between *readability* and *text quality*, and that these differences can be identified through automatic methods.

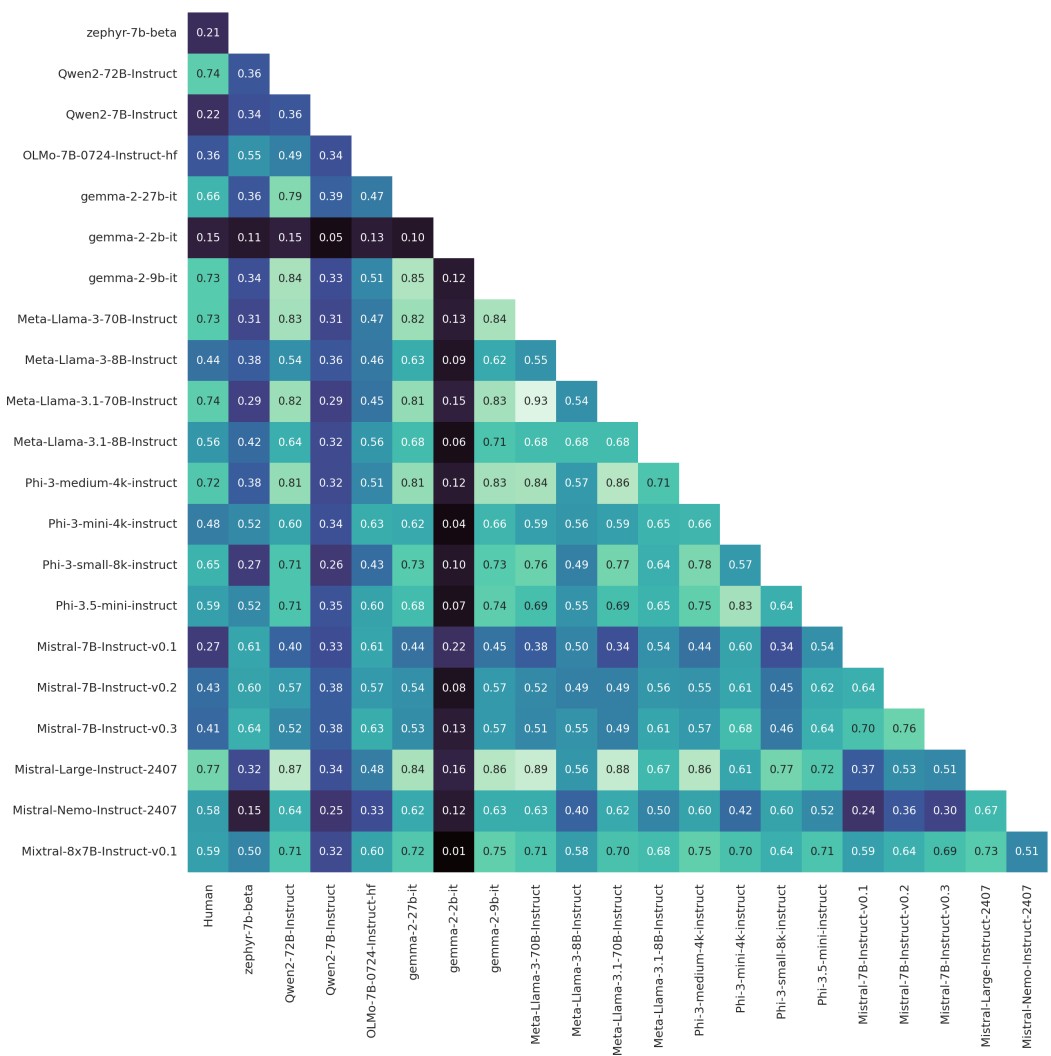

Figure A28: Pearson correlation coefficients for various instruction-tuned language models tasked with judging the readability of the CLEAR dataset. We observe that larger models tend to have a stronger correlation with human judgments of readability. Notably, the largest model, Mistral-Large-Instruct-2407, which has 123B parameters, exhibited the highest correlation. The smallest model with a coefficient greater than 0.70 was gemma-2-9b-it, which has 9.24B parameters.

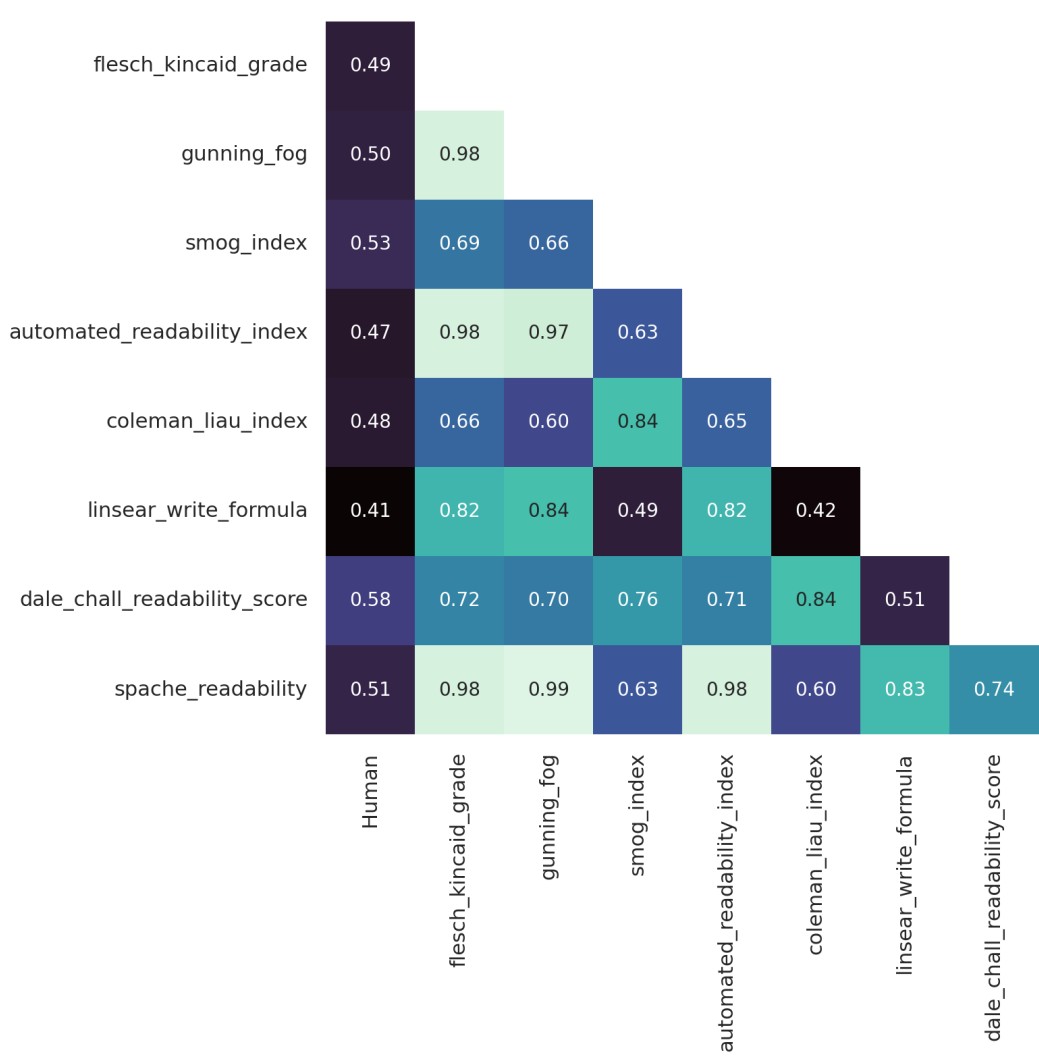

Figure A29: Pearson correlation coefficients for various classic readability formulas applied to the CLEAR dataset. The results indicate that all formulas show a similar correlation with human judgments of readability.

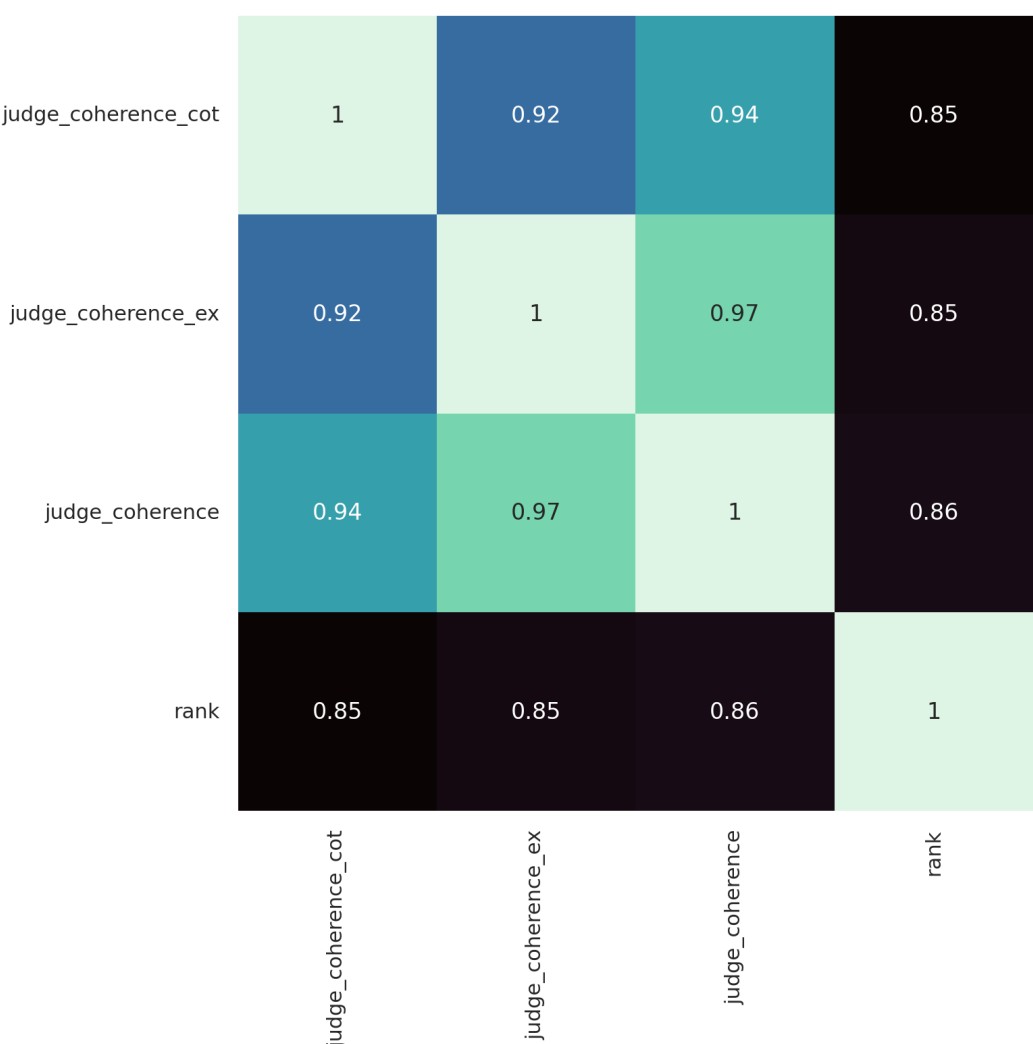

Figure A30: Pearson correlation coefficients for different prompt choices for LLM-as-a-Judge to measure coherence are presented. Scores are computed for generations conditioned on prompts from LlamaTales-GRE. judge_coherence uses the prompt shown in Figure A11. judge_coherence_cot uses the prompt shown in Figure A12. judge_coherence_ex uses the prompt shown in Figure A13. We find no meaningful differences between the prompt variants.

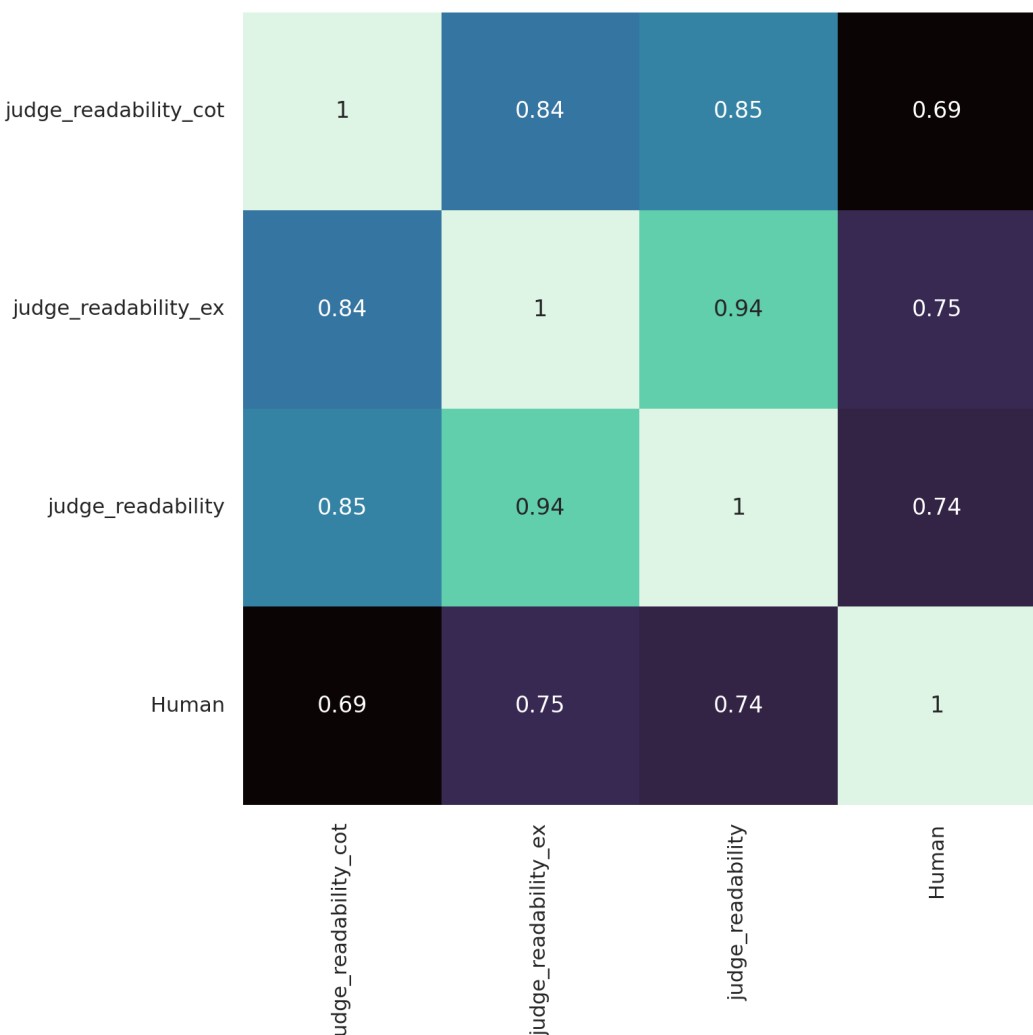

Figure A31: Pearson correlation coefficients for different prompt choices for LLM-as-a-Judge to measure readability are presented. Scores are computed over the CLEAR dataset. judge_readability uses the prompt shown in Figure A10. judge_readability_cot uses the prompt shown in Figure A14. judge_readability_ex uses the prompt shown in Figure A15. All variants are strongly correlated with human experts, but judge_readability_cot exhibits the lowest correlation.

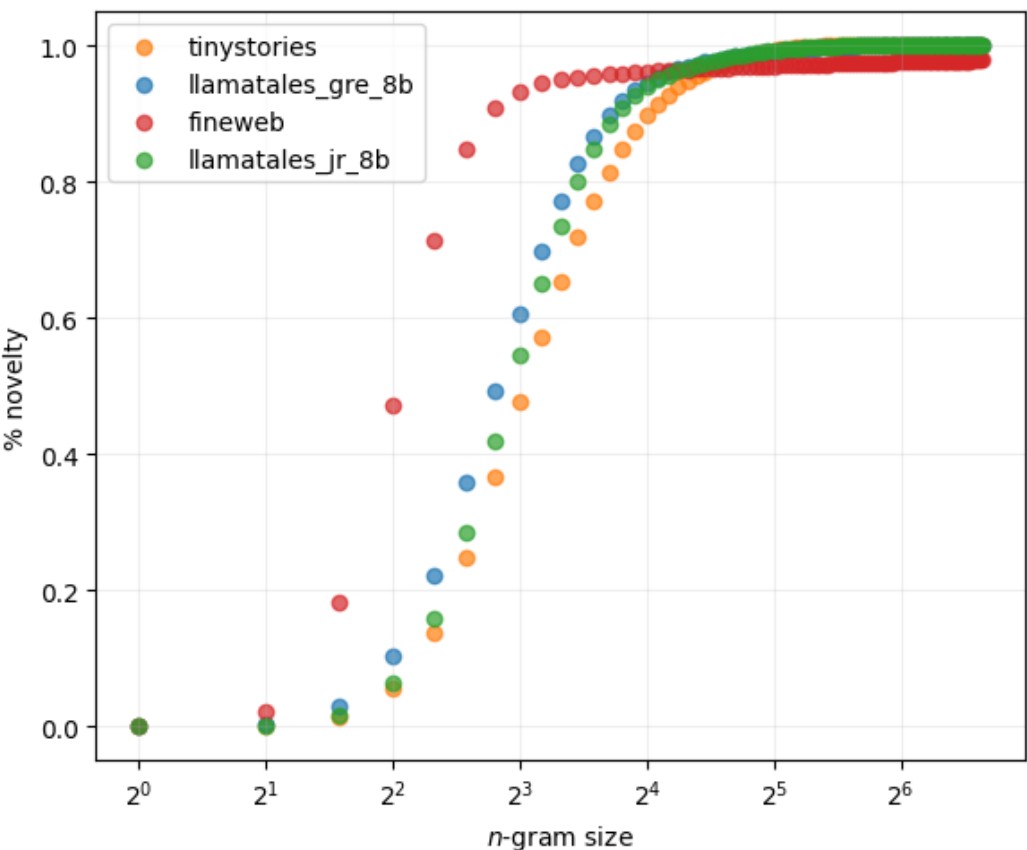

Figure A32: *n*-gram novelty of test splits with respect to train splits. SLMs trained on LlamaTales-GRE generate slightly more novel *n*-grams than those trained on LlamaTales-Jr, followed by those trained on TinyStories. However, SLMs trained on FineWeb generate substantially more novel *n*-grams than those trained on the other three datasets. See Figure 5 for an alternative view of this figure.

Table A5: Random examples from the datasets described in Section 2 and Table 1. Newlines removed.

| Dataset | Example (first 500 characters) |
|---|---|
| TinyStories | One day, a little boy named Tim went to the park. He saw a big, red ball. Tim wanted to play with the ball. He looked for someone to play with. He saw a man with a kind face. The man looked reliable, so Tim asked him to play. They played with the ball for a while. Tim was having fun. But then, the man took the ball and ran away. Tim did not know what to do. He had not met a bad person yet. He was scared and did not know how to get his ball back. Tim started to scream. He hoped someone would help |
| TinyStories | Once upon a time, there was a little boy named Tim. Tim was a very honest boy. One day, he saw a big fight between two dogs. Tim wanted to help them. Tim said, "Stop, dogs! No fight! Be nice!" The dogs stopped and looked at Tim. They saw that he was honest and wanted to help. So, the dogs listened to him. Tim led the dogs to a park where they could play. The dogs became friends and did not fight anymore. They all played together and had fun. And Tim was happy because he helped the dogs be friend |
| TinyStories | Once upon a time, in a small village, there was a kind and compassionate boy named Tom. He loved to play with his colorful marbles. One sunny day, Tom went to the park with his friends to play with their marbles together. While they were playing, a little bird flew down and took one of Tom's marbles. Tom and his friends chased the bird to get the marble back. They found the bird's nest high up in a tree. Tom climbed the tree and saw that the bird had used the marble to make her nest pretty. Tom |
| LlamaTales-Jr | Benny was a happy rabbit. He liked to play outside. Benny's fur was pale white. He loved to hop around the green grass. One sunny day, Benny's friend, a squirrel named Squeaky, gave him a little gift. Squeaky said, "Benny, I got you a video!" Benny asked, "What is a video?" Squeaky said, "It's like magic. You put it in a machine and it plays a show." Benny was excited. He put the video in the machine. He pressed play. Suddenly, a colorful world appeared on the screen! Benny saw a bird flying hi |
| LlamaTales-Jr | Once upon a time, in a sunny jungle, there lived a little elephant named Ellie. Ellie loved to play on her shiny new bike with big, strong gear wheels. One day, while riding her bike, Ellie saw a rock that was very slippery. She tried to stop, but her bike went too fast. "Oh no!" said Ellie. "I might fall!" She tried to turn around, but it was too late. Her bike went up a little hill, and she went too close to the edge of a big cliff. It was a long way down to the bottom. Ellie was very scared. |
| LlamaTales-Jr | Once upon a time, in a sunny savannah, there lived a happy ostrich named Ollie. Ollie loved to run fast. One day, he said, "I want to be the fastest animal in the savannah!" A wise old bird told Ollie, "Being humble is key to being happy. If you're proud, you might trip and fall." Ollie did not listen. He practiced and practiced running. He ran through the tall grass and felt the sun on his feathers. One day, while out for a run, Ollie saw a big, deep pit in the ground. "Oh no!" he exclaimed. |
| LlamaTales-GRE | The Whispering Woods of Blackthorn In the sleepy town of Ravenhurst, nestled between the Whispering Woods of Blackthorn and the jagged peaks of the Crimson Mountains, the townsfolk had grown accustomed to the unsettling legends surrounding the ancient forest. The townsfolk whispered tales of the woods' ability to whisper darkest fears into the ears of those who dared to wander too far within. Old Man Thorne, the woods' self-proclaimed caretaker, would often caution travelers of the dangers that |
| LlamaTales-GRE | In the mystical realm of Aethoria, where the skies raged with perpetual storms and the mountains pierced the heavens like shards of splintered stone, there existed a land of contrasts. The Iron Kingdom, with its fortress-like spires and battlements of steel, was the stronghold of the ruling regime, led by the enigmatic King Arin. From the aerie of his throne room, he decreed his laws and mandates, meting out justice with an iron fist. Below the kingdom, in the city of Emberhaven, a brewing diss |
| LlamaTales-GRE | In the mystical realm of Azura, where the skies raged with perpetual twilight, a young thief named Kaelin Blackwood navigated the streets with a rakish charm, his tattered cloak billowing behind him like a dark specter. His eyes gleamed with a hint of mischief, as he wove in and out of the crowded market stalls, seeking his next mark. But Kaelin's inherent restlessness drove him to pursue more than mere riches – he sought redemption for past mistakes and a chance to prove himself worthy of the n |
| FineWeb | The Bureau of Mines is conducting studies of the slags related to ferrous technology to evaluate acceptable substitutes for the auxiliary flux, mineral fluorspar, in foundry operations. Comparative basic practice cupola (18-inch-id) trials were made to evaluate a waste material called "used potlining," which contains significant levels of fluorine, sodium, and aluminum in various compounds. Used potlining is recovered from alumina reduction cells after its useful life, and was supplied by the Al |
| FineWeb | WHO Defines Food security to exists when all people, at all times, have physical, social and economic access to sufficient, safe and nutritious food which meets their dietary needs and food preferences for an active and healthy life. Food security has three interlinked contents such as :- - Availability of food, - Access to food and - absorption of food. Food security is a multidimensional concept covering even the micro level household food security,energy intakes and indicators of malnutrition |
| FineWeb | The Pulse: Robots Remove Brain Tumors Thu, 09/12/2013 - 10:12am On this episode of The Pulse, brought to you by MDT TV, we're removing brain tumors with robots, using MetaboShield to help obesity, regenerating heart tissue, and pairing glucometers with smartphones for ease of use. This episode features: - Engineers are teaming up with neurosurgeons to develop technologies that enable less invasive, image-guided removal of hard-to-reach brain tumors. - Students at the Hebrew University of Jerusal |

Table A6: We sample a random 50-token prompt from the test split of `TinyStories` and generate with models from Table A2. Shaded rows indicate that the model was trained on `TinyStories`. <START> marks the start of generation. Part 1 of 4. Return to Section 4.

| Model | Example (first 500 characters) |
|---|---|
| Prompt | Once upon a time, there was a little girl named Lily. She loved to eat rice every day. One day, she was playing with her friend Tom in the park. They saw a big box with a sign that said "Pri |
| llamatales_gre_8b-lay8-hs512-hd8-33M | ze inside<START>" didn't quite match the field. Lily's friends had noticed her contagious laughter and joyful weeping days. She'd always seem to be afraid, and besides, she still played me play games at all. As the days went by, her friendships continued to grow, and she became the most popular person in town. Their friendships centered around high friendship rather than genuine connections, which was detrimental to Lily's own life. She'd grow up awkward and unfriendly, often skipping plans for her Instagram blog. However, there was also a deep love for one another, and she held within her unwavering trust that needed continuous support. One day, while out as a bird, Lily |
| llamatales_gre_8b-lay8-hs384-hd6-18M | ze inside<START> the poor" and pushed forward, completely carefree. But as they deliberated, Lily couldn't help but tout the machine that was supposed to be a revolutionary solution. She was a feisty child who had a favorite for juggling games with their friends and neighbors at one point. Despite the odds, Lily finally won over Tom as well. He begged her to give it up and fix it, and Lily, blinded by her desire for excitement and admiration, agreed. The problem was, she had been working tirelessly for months to perfect the system. With a juggling operation and a show of force, they managed to attract a steady stream of customers who were as out of touch with reality as th |
| llamatales_gre_8b-lay4-hs384-hd6-9M | ze inside<START>" "Emma's new menu, 'Hey'-the-star" – an explosion after the cooking competition arose. Their friend, the mayor, Aki, tried to intervene, but Lily wouldn't listen. Instead, she focused on her next course, and the morning of the picnic rain brought as much joy to their town. As the festival loomed, a sense of somberness hung over the crowd. Thunder boomed, and the excitement turned to despair. This year, they had decided to invite the mayor to bake a special cake, one that would not only be held until they saw the perfect perfect dessert. The mayor's restaurant was hastily depleting out of customers, and the strange, yet charming new surroundings made them f |
| llamatales_jr_8b-lay8-hs512-hd8-33M | ze inside<START>." Lily and Tom were curious. They wanted to open the box and see what was inside. They opened the box and saw a delicious-looking chocolate bar. They were so happy and ran to find a good place to put it. They laughed and played with the chocolate, but then Lily thought of her friend Tom, who was playing nearby. They decided to share the chocolate and sit together under a tree. Lily shared her chocolate with Tom and said, "If you have something special to share with me, you can share it with me!" Tom smiled and said, "That's a great idea!" Lily and Tom shared the chocolate and played together, happy friends. They learned that sharing is caring, and that's w |
| llamatales_jr_8b-lay8-hs384-hd6-18M | ze inside<START>." Lily and Tom were curious about the box. They opened it and inside, they found some toys and sweets. Lily liked the sweets, but Tom also found a toy soccer ball. He wanted it for himself. Lily was not happy and said, "That's not fair! I want to eat the treats." Tom said, "But why?" Lily replied, "I want to eat the sweets, it's a snack." Tom said, "But I need something else. That's not fair." Lily started to feel sad. She looked up and saw a flock of birds flying overhead. One of the bird's tail feathers began to wag. It was getting closer and closer to the box. Lily remembered that her mom said they could play a trick on Tom by hiding treats. Just then, |
| llamatales_jr_8b-lay4-hs384-hd6-9M | ze inside<START>." Lily and Tom were curious about the box. They opened it and inside, they found some tasty rice! But then, something went wrong. As they ate the rice, a tiny mole popped out of the box! It was little and brown and fluffy. The mole said, "Hello! My name is Moe." Lily and Tom were surprised. "Really? We have a pet mole?" they asked. Moe said, "Yes! My Moe helps take care of the rice. It's nice to eat it." Lily and Tom were excited to see Moe's home. They promised Moe that they would help her every day. Lily, Tom, and Moe became the best friends ever. Every day, they would visit the library to find new oats to eat. They discovered how to help Moe, and they b |
| tinystories-lay8-hs512-hd8-33M | ze inside<START>". They wanted to win the prize, so they opened the box. Suddenly, a big, scary monster appeared! Lily and Tom were very scared and ran away. But the monster grabbed them and took them to his lair. They never saw their families or friends again. From that day on, they never ate rice again. |
| tinystories-lay8-hs384-hd6-18M | ze inside<START>". Lily and Tom were very curious about what was inside the box. They went home and asked their mom to help them open the box. When they opened it, they found a cute little puppy inside! Lily and Tom were so excited and ran back to the park to show their new friend. When they got there, they saw that the puppy was bigger than they were before and bigger! The puppy jumped out of their hands and started to run around. Lily and Tom chased the puppy around the park until they were tired. Finally, the puppy stopped and Lily saw that it had run a small circle in its head. She realized that the puppy was making a home for itself! From that day on, Lily and Tom had |
| tinystories-lay4-hs384-hd6-9M | ze inside<START>". Lily and Tom ran to the box and tried to open it. It was difficult, but they finally did it. Inside the box, they found a cute puppy! They were so happy and played with the puppy all day long. At the end of the day, Lily and Tom went home with smiles on their faces. They couldn't wait to tell their parents about the puppy and the gift they found. |
| fineweb-lay8-hs512-hd8-33M | ze inside<START> Rose go" and pushed it to the end of the box. I thought that was okay, because Lily might like to be angry. But, when Tom saw her, the box had a sign that said "WORITE TOGETHER." I think that's a good link for Lily as I know that she won't be angry about all the things she finds in this box. What I do not know is where it came from. They found the box close by and this time Lily could not stop playing - he couldn't seem to stop until one out of the box held a reminder to send another out to him. We already know Lily can reach out to another friend, but in reality, she must catch her own throat. The other friend will have to pay him a lot of money, and when |
| fineweb-lay8-hs384-hd6-18M | ze inside<START> the box" and pushed forward, like all jewels. Then the next morning, after night we cut a cake in the machine. Each time we rearranged the bottom of the piece, we put six pieces of bread on the screen and cut it in half so that it is "Our Great Buns." In the second step, we cut paper into three sections. The portion that was the "square." The piece is called the "paper plate." Between pieces of paper, we cut a square. Each piece of paper laid aside approximately 4/3st the slices of rice until they are split. Then we cut strips of paper into strips (paper plate). We broke them into three separate strips, with each strip folded over. The strips were cut indi |

Table A7: We sample a random 50-token prompt from the test split of `TinyStories` and generate with models from Table A2. Shaded rows indicate that the model was trained on `TinyStories`. `<START>` marks the start of generation. Part 2 of 4.

| Model | Example (first 500 characters) |
|---|---|
| Prompt | Once upon a time, there was a little girl named Lily. She loved to eat rice every day. One day, she was playing with her friend Tom in the park. They saw a big box with a sign that said "Pri |
| fineweb-lay4-hs384-hd6-9M | ze inside<START> the box" and Ika, what all these girls did was drinking the jelly after night cooking. Most of the boys were angry about each other knocking at their bell because the jelly had a hint of it. Ways for Kids Found About Cooking their Skin During Day Learning May Cliches 1. Dog Yard Fire: Roughly 11 children are now choosing clothes for Scabies.Knowing and smoking the bean tumble. But there have been some unusual pieces of paper and sketches for people who thought that the pieces you can tell away were obviously just one problem. Gertrude Rouse will show another paper says that her cat (or monkey) is too similar to the one that premiered in her New York Academ |
| EleutherAI/pythia-12b | ze inside<START>. It's small!" Her friend Tom said, "Let's play that game!" As they walked over to the box, Tom got on his knees and looked inside. Lily rolled her eyes and said, "It's just a box!" "No it's not! Hurry up!" Tom said. Lily said, "OK," and she pushed her friend Tom into the box. There was an owl sleeping on top of it. It was cute, and it had a plush head with hollow eyes and a feathered tail. "Open the box!" she told her friend Tom. "Aww come on! Not now!" Tom said. Lily opened the box with her hands. As she looked inside, there was a box of pastries. She counted her eyes and picked one. She ate the pastries and said, "We got another one!" "No, we didn't!" he |
| EleutherAI/pythia-6.9b | ze inside<START>!! It's small!" So Lily and Tom climbed up and saw the prize, a little orange poodle who licked their fingers. Tom asked Lily, "why can't I eat this dog?" Lily answered him and she kissed Tom. Tom felt happy, but Lily was upset because she had to climb down and leave the poodle. Lily's mother told her, "you will grow to love it very much." But Lily didn't want to eat any dog. Lily never ate any dog again. Later, Lily's mother told her, " I will buy you some doggy to eat." Lily was very happy, and she ate the doggy before anyone asked her. In a few days, the doggy grew so tall that she couldn't go down the stairs anymore. So Lily came to her mother's house a |
| EleutherAI/pythia-2.8b | ze inside<START>". It reminded them both of foods they were already eating. Then the box opened. Inside, there was rice mixed with flowers, and a big piece of cake. They ate all of it at once. Afterward, they played together in the grass. Lily's favorite part was to build a wall with different shapes she made from her blocks, and then she could see through it. They went home, ate their prizes, and then went to sleep. That was Lily's first food. That's what my best food is like. I love that idea of almost choosing from whole meals that I have already loved. I already knew what was inside of the box, but it was like the walls inside of me. They surrounded me and stopped me f |
| EleutherAI/pythia-1.4b | ze inside<START>," and Tom thought it was a present for him. Tom went over to get it. A little boy walked over to Lily, and said to her, "You're a very good kid; you want to play with us, instead of the grass." Lily then threw the box to the little boy. He opened the box and took out a bag of rice, and put it in her hand. Lily ate it all, and then she rushed over to tell her friend Tom. Tom was very proud, because he said Lily had eaten all the rice. (Lightness of mind from carrying a bag of rice after "He was very proud.") |
| EleutherAI/pythia-1b | ze inside<START>," and Tom asked her to go get it. The three of them were standing in line. After they walked over to the box, Tom got out a pencil and wrote something on it. As he reached across the box and placed it in the grass, Lily started to cry. The next thing, he threw the box into the water. She had loved rice so much that she wanted to eat it! Rice Balls: The life-changing experiences of Mother's Day tradition and Son's Day celebration of friendship, strengthening marriage and family bonds, I Bring you thoughtful meals that are easy to make, healthy and delicious! Enjoy! My mother was very much like the people I know today. They would have enough time to kill a f |
| EleutherAI/pythia-410m | ze inside<START>," and she thought it was a Goodwill thing. Tom saw the bag, so he picked it up and tossed it at her. "You made me miss all the gifts," Lily said. "It was just a prank," Tom answered. "You would have been tired and hungry when you lost everything," said Lily. "I didn't know what to do," said Tom. "It looks as though this was a nice surprise." "It's a nice coincidence, I guess," said Lily. "You almost never see me without a cookie." He started to hold up an eye patch when Lily cried out. "Are you okay? Do you need help?" "No," he said. "It's just that I didn't know what to make. I mean, it looks so pretty. I'm not sure what I'm supposed to do with it. And wh |
| EleutherAI/pythia-160m | ze inside<START>," and she pinned it down. She dressed in a Japanese garb that would have sounded foreign to some, such as flowers, had it not been for the camp's tall wooden train carrying a high tasseled collar and bicycle strap. "This is just the right time for me to finish my education," Lily recalled. He loved to travel alone, leaving her only fifty-one days to meet him. There were only a few days on this week, and her journey wouldn't be complete until December. Lily worried about how she would perceive that longerness at scale from town to town, and she "tried to hold up an allegiance to the host, but failed," she said to her friend. "Eventually, there was a boat ca |
| EleutherAI/pythia-70m | ze inside<START>," and she read it aloud. She welcomed the talk and a chorus of monks. As we walked through past flowers, I had to look at the stacks of cafés; they covered my bedroom with a red and white deckchair. They felt like stone flowers Our Blog Thank you for your help. Your first Guest Bloger helped me share her excellent art quiz series and an informative quiz per week of fiction, videos, and pictures of Russian children. Recently the Son of a Warrior donated a 5HPa piece to that idea, I tested from the top of the Heritage Your "Book of Scratch" paper museum in Schleswig-Holstein. In particular, we used the FiveThousandth-Founder of Science and Architecture at th |

Table A8: We sample a random 50-token prompt from the test split of `TinyStories` and generate with models from Table A2. Shaded rows indicate that the model was trained on `TinyStories`. `<START>` marks the start of generation. Part 3 of 4.

| Model | Example (first 500 characters) |
|---|---|
| Prompt | Once upon a time, there was a little girl named Lily. She loved to eat rice every day. One day, she was playing with her friend Tom in the park. They saw a big box with a sign that said "Pri |
| openai-community/gpt2-xl | ze inside!<START> One day only!" No kid, not one, said "You mean that I can come out here and get a random prize?". Sure enough, just as they went into the park, there was a big box sitting right there. Inside were all kinds of prizes. Some were which had the letter in them. One of them said "BTW I'm Nate. Well, somebody may not like you. We know who you are. We wouldn't have bothered you if we did. So why should we give this to you? Oh no, you thought that you were actually gonna score this prize." That was how little Lily felt when she saw the winner's box. She had to confront how much she was buying into the ruse. After all, who would spend a whole day happily working h |
| openai-community/gpt2-large | ze inside!<START> One day soon - No food left to waste!" These days, thanks to her friend Tom, Lily is famous all over the world. Tom finds her very entertaining. She is so too with her high income. The best thing is that she is about to get married to Tom. So, what do you do with such a delicious treasure? Very funny story by Tom Well, Lily loved Tom so much. He is so handsome, very nice, he is smart, very handsome... She tells Tom to take some cereal from the box in order to measure the number of S's on his head. But, he refuses! So, Lily gives Tom to her family. They open the box. What they find out that there is a big, green dragon egg inside. Curious, they run back in |
| openai-community/gpt2-medium | ze inside!<START> One day, in order to get to the prize, you must spend extra attention to the other boxes. After all, we have lots of rewards." Your Guardian Spirit could appear at any stage of the game. The best strategy is to spend all your health points on food! Later, the letter announcing your victory would be around the corner in the market. The village gossip would tell you that "Lily is neither strong nor weak, she just likes to eat!". When the event wrapped up you might have some luck against the Hog Bee... Well, at that time, there was only Lily. But recently, there has been a lot of excitement to get you into the game and bring Lily to life again! Lily's not fo |
| openai-community/gpt2 | ze inside!<START> Hagiff Turtle Box No. 862, Palm Islands." The thing that was next to him was a turtle that had eyes. They thought that was very weird. I said, "I'm really curious," and they picked out the turtle and about it the little girl was so happy, she brought it back and we moved it upstairs in the house. I told them, "Go ahead, you brother so tell everybody. Then the turtle wouldn't have to die. You've got to help him. We've made it out in no time." All that's left is to copy this many other videos. Today, if you want to get excited about ginger flying, bring a sheet of paper and pen like not found in any other craft. Everybody would fly a turkey-flying drone, fi |
| TinyLlama/TinyLlama_v1.1 | <START>ze Rice". "Yes, let's try it," said Tom. Soon the girls see their aunt pushing a trolley. The aunt said "I am selling the rice" and asked the girls. The first one ate half of the box. Soon, the aunt saw the girls eyes turn red and their cheeks swell. "No you have to eat the rest of the box." said the aunt. The second girl started coughing, 'Not just rice', she said. "Yes, let's see what is inside." Said Auntie. She took a bite out of the rice and her arm started hurting. Then Tom said, "Let's hide." So the girls ran. One of them hid behind a big tree and the other one hid behind a bush, they could still hear the auntie trying to hide. By the time the auntie said "Ha |
| meta-llama/Meta-Llama-3.1-70B | ze inside<START>". They decided to open it, but when they opened it, they found that it was empty. Lily was sad and said, "I wish someone would give me a prize." That evening, Lily's mom made a big bowl of rice for dinner. Lily was very happy and she said, "Mom, I love rice!" Just then, the phone rang. Lily's mom answered the phone and she heard a voice say, "Hello, I am from the rice company. We heard that you love rice, so we would like to give you a prize. Would you like a big bag of rice?" Lily's mom said, "Yes, we would love that!" She gave the address to the rice company and they sent a big bag of rice to Lily's house. When Lily saw the big bag of rice, she was very |
| meta-llama/Meta-Llama-3.1-8B | ze inside<START>". "Let's open it, what's inside? I'll open the box", Lily said. "No, you should open the box, it's not fair!", said Tom. "Oh, I want to see what's inside the box, please let me open it, let's open the box together", Lily said. "Ok", said Tom. Lily pulled up the lid of the box. "OMG, it's a panda and it's talking!" "Is it possible?", Tom wondered. "Hello, I am Jai, one of the toymakers, let's make a perfect toy, we'll start it as soon as possible", Jai said. "Wow, Jai has made me a very beautiful toy", said Lily. "It's our present for you to be happy", Jai said. "Come on, Tom", Lily said. "Let's play with the toy. The ball is bouncing up, come on, Lily!" "I |
| Qwen/Qwen2-72B | ze inside!<START> For those who love rice and can solve this riddle: I am white as snow, but not cold as ice. What am I?" Lily and Tom thought for a while and came up with the answer: rice! They opened the box and found a bag of special rice that had been grown by a magical chef. They brought it home and cooked it up for dinner that night. It tasted better than any rice they had ever had before. Can you solve the riddle too and guess what the prize inside the box was? I am white as snow, but not cold as ice. What am I? Hint: The answer rhymes with "rice." The answer to the riddle is "rice." The box contained a special bag of rice that had been grown by a magical chef. |
| Qwen/Qwen2-7B | ze inside!<START> For solving a riddle." They were curious and went to see what the riddle was. They read the riddle and were stumped. They asked if they could ask for a hint, and the person in charge of the box said they had three hints. Lily and Tom each had one hint, and they could choose to share it or keep it for themselves. Lily used her hint and the answer to the riddle was "three". They went inside and received their prize: three bags of rice! They were so happy and decided to share the rice with the person in charge of the box. They went to the park again to thank the person and the park looked greener and more beautiful. When they went back home, they told their |

Table A9: We sample a random 50-token prompt from the test split of `TinyStories` and generate with models from Table A2. Shaded rows indicate that the model was trained on `TinyStories`. `<START>` marks the start of generation. Part 4 of 4.

| Model | Example (first 500 characters) |
|---|---|
| Prompt | Once upon a time, there was a little girl named Lily. She loved to eat rice every day. One day, she was playing with her friend Tom in the park. They saw a big box with a sign that said "Pri |
| mistralai/Mixtral-8x7B-v0.1 | ze<START> Inside!" They wanted to know what was inside, so they asked the park attendant. He told them they could open it if they promised to share what they found. Lily and Tom agreed. They opened the box, and inside they found a shiny golden egg. They took it home and put it in a special box. Every day, they checked to see if the egg had hatched. Finally, one day, they heard a tiny voice coming from inside the egg. It said "I'm coming out!" Lily and Tom were so excited! They helped the egg hatch, and out popped a tiny little dragon. The dragon was as black as night, and she had glowing green eyes. She was the most beautiful dragon they had ever seen. They named her Dot. |
| mistralai/Mistral-7B-v0.3 | ze<START>" on it. They looked around for a nearby trashcan to toss their sandwich into. Tom said, "You first. I'll follow." Lily didn't like to play games, so she said, "No thanks, we don't want any prize." And with that they left. Tom was sure that the sign was fake, but Lily insisted that it meant that there was something special inside the box. She wouldn't be able to eat rice for a month. But, she didn't mind. She was sure that there was something really good inside the box. The next day, Tom and Lily went to the park again, but this time they didn't see the box. Lily said, "We'll go again tomorrow and get the prize." The next day, Tom and Lily went back to the park, b |

Table A10: We sample a random 50-token prompt from the test split of `LlamaTales-Jr` and generate with models from Table A2. Shaded rows indicate that the model was trained on `LlamaTales-Jr`. `<START>` marks the start of generation. Part 1 of 4.

| Model | Example (first 500 characters) |
|---|---|
| prompt | Once upon a time, in a small village, there was a kind old man named Bob. Bob loved to tell stories to the children in the village. He had a nice library with many, many books where he kept all his beloved stories |
| llamatales_gre_8b-lay8-hs512-hd8-33M | . `<START>`One day, Bob decided to help the young boy, Tommy, who was struggling to find his place in the world. He knew he had to act quickly to save the library, and so he began brainstorming new ideas for solutions to the village's problems. Jimmy had always been a bit of a disgruntled one, often complaining about the grey areas in life, and Jimmy believed he could do just that. But with Bob's guidance, Jimmy had finally found a way to apply for a job in the nearby city, where he could help with everything. "I can do that, Jimmy," Tommy said, his voice filled with a newfound sense of determination. "But it's not going to be easy. There's one bugbear, or I don't know, it won't be someone we love |
| llamatales_gre_8b-lay8-hs384-hd6-18M | . `<START>`One day, Emma, a young girl with a bright smile and a contagious laugh, wandered into the village. She came with a basket of freshly baked cookies and a mind buzzing with ideas. The villagers were immediately impressed by her enthusiasm and welcomed her with open arms. She quickly began making acquaintances with various individuals, including her school teacher, Mrs. Jenkins, who was an avid supporter of Emma's initiative to literacy exercises. Her approach was impartial and insightful, far exceeding anyone's expectations. As Emma settled into her new friends, she started to become one of the pair. She learned that books were not just volumes, but vessels for knowledge and imagination. |
| llamatales_gre_8b-lay4-hs384-hd6-9M | . `<START>`As time went by, the villagers began to feel a sense of unease. The family arrived one fateful evening, and the old man, hailing from a young age, started a powwow. The powwow was filled with the tribe, and the scent of food and stories filled the air. John, now attuned to the tribe's stories, thanked them, his mind fixated on the legend of the "Chrono-Cen and Mija." This year, he declared that he would have to join the powwow, and with one condition, to be held within his will. The villagers, a motley crew of travelers who passed through the village, listened with a mix of curiosity and trepidation. They had heard tales of temporal disruptions, malfunctions, and inexplicable coincidenc |
| llamatales_jr_8b-lay8-hs512-hd8-33M | . `<START>`Every day, Bob would sit by the library and read his books. The children would gather around him and listen to him talk about the ancient world he found. They would point at the pictures of the village's people who lived there long ago. Bob loved the stories of the people and didn't want anyone to hear one boast about it. One day, a strong wind came to the village, and it brought a big storm with loud thunder and flashes of lightning. The children were scared and hid in the library. But Bob stayed at the library, shielding his books to keep the storm out. The children were worried that their favorite books would get hurt. Bob stood in the doorway, watching the storm through the window, |
| llamatales_jr_8b-lay8-hs384-hd6-18M | . `<START>`One day, Bob said to his friends, "Tomorrow I will visit the village library and we will have the best stories ever!" His friends were so excited to go with him. They all put on their favorite hats and walked to the library together. At the library, Bob began to read his stories. He said, "Wow, this is the best story I have ever heard! The people from the village are all telling stories too!" They were all happy and laughing together. But then, Bob noticed something strange. His shop was getting very crowded, and the doors were closing. Bob saw that a big bear was coming in and it was holding a sign that said, "I have been waiting for you! You have made all these stories in the library, |
| llamatales_jr_8b-lay4-hs384-hd6-9M | . `<START>`One day, Bob decided to build a special book for his animals. He used big blocks and some strong nails to build it. He was very happy and worked hard. But, one day, a big storm came to the village. The winds were strong and the rain poured down. The winds were so strong that they knocked down the bookshelf. Books were scattered all over the ground. The books were scratched and all fell down. The children were sad. They wanted to have a quiet story, but the storm had broken all the books. Bob was very sad too. The storm got stronger and stronger. Many of the villages got hurt. The town was flooded and the books were broken too. The little animals were crying and very scared. Bob tried to |
| tinystories-lay8-hs512-hd8-33M | . `<START>`One day, a new family moved to the village. They had a little girl named Lily. Lily was very spoiled and always wanted more books. She knocked on the old man's door and asked for a new book. Bob felt sad for Lily, so he decided to lend her a book from his favorite library. When he went to his library, he found a beautiful, new book just for Lily. He brought it to her and told her stories from his books. Lily was very happy and thanked Bob for his kindness. The other children in the village learned from Bob's kindness and they started to help each other too. They shared their toys, helped their classmates, and started making their village a happy place. The old man's books brought joy to |
| tinystories-lay8-hs384-hd6-18M | . `<START>`One day, a new family moved to the village. They were very excited to find a new family. They asked Bob to tell them a story, and he said, "We will all be happy together." The children listened carefully and decided to have a meeting with Bob in the library. In the meeting, they talked about how to be kind and help others. The children played games, read books, and helped each other when they needed it. In the end, they learned that being kind was important, just like how Bob taught them. The children learned the importance of kindness, and they were always kind to each other. |
| tinystories-lay4-hs384-hd6-9M | . `<START>`One day, Bob went to the forest to look for his book. While he was walking, he found a big, old tree. He looked up and saw a cute, fluffy animal stuck in the tree. The animal was scared and cried, "Help me, please! I can't get down!" Bob heard the animal and went to help it. He reached up high and carefully took the animal out of the tree. The animal was so happy and said, "Thank you, Bob! You are very kind." Bob smiled and said, "You're welcome, little one. Now, we can all talk and have fun in the forest together." And from that day on, Bob and the animal became the best of friends. The village was a happy place, and everyone loved to hear Bob's stories. |

Table A11: We sample a random 50-token prompt from the test split of `LlamaTales-Jr` and generate with models from Table A2. Shaded rows indicate that the model was trained on `LlamaTales-Jr`. `<START>` marks the start of generation. Part 2 of 4.

| Model | Example (first 500 characters) |
|---|---|
| prompt | Once upon a time, in a small village, there was a kind old man named Bob. Bob loved to tell stories to the children in the village. He had a nice library with many, many books where he kept all his beloved stories |
| fineweb-lay8-hs512-hd8-33M | . <START>. Whenever Bob heard Bob's famous story, he got very excited. The children would go home, and the children would tell the story each time he found something of interest. The children would sing stories of good people who helped others and such who might live in difficult times. Sometimes the adventure as a family might seem rather exhausting, as a survival tactic. It would be an incredibly unpleasant experience, and not just bad, but uncomfortable. The excitement and joy was far higher than this: he could not stop playing around with friends. But it was just one problem. Bob was only a small part of a group of children. He was not as blessed as Bob as another child, but he could be just |
| fineweb-lay8-hs384-hd6-18M | . <START> one go. Peter and Alex, the members of my very own family, had families in a nearby village. One of the families was the Miss Archer and also of white silk, and all of them preferred good hair dyes. Ruby was a hard and light gray hair, and she didn't have any hair. When he went home, all the children were glued to his father's belongings, and everyone else looked at him, and just followed his father. |
| fineweb-lay4-hs384-hd6-9M | . <START> how poor men represent them and his bad myths about them very rapidly. Unlike most families in a family that have a lot of books dedicated to them, most also consider themselves the children who are free to move at so much for their side by side their own word. When one says "yes" to Bob, they say, "If you have the children." Most schools have books set aside for these children to read, others sit there and say things that sound legitimate. By this time, these children have been around for more than 60 years. With this college to do that, they will have a unique type of life in a family as they reach out to another generation. How do your children learn language? Why teach English very |
| EleutherAI/pythia-12b | . <START> One day, while reading a story about a fox and a goose that could really help each other, Bob realized that he didn't have enough books to read all the stories he knew about. So he bought an encyclopedia and he found that it only had two stories about foxes and geese, and neither of those stories matched the story that he wanted to read. Bob went to see a village librarian and the librarian told him that if he wanted to read the story, he should go to the editor of the encyclopedia, because they had it all. So Bob went to see the editor, but he refused to give Bob the book, saying that the encyclopedia was not just an encyclopedia, it was also a library. I don't remember all the story, |
| EleutherAI/pythia-6.9b | . <START> One day, while he was reading, a fox walked by the library, stopped to rest, and stared at the various books. "What's so great about all these books?" asked the fox. "Well," the old man replied, "there's truth in each one." "How can that be?" said the fox, "there is only one truth." The old man did not give up. "You are wrong," he answered, "these books contain a multitude of truths." "What is truth?" asked the fox. "It's what we all live by," answered the old man. Then the fox said to him, "I am surrounded by tricksters." "You are wrong," said the old man, "that is what makes people fooled. You will never find the truth as long as you are looking in books." So they walked off together |
| EleutherAI/pythia-2.8b | . <START> One day, while he was reading about a talkative cat, Bob came across an unusual story: One day, Bob had gone out to pick some sage and salt for Bob's magic tonic. Suddenly Bob discovered that he didn't have any sage or salt! The children would hate him, and stop listening to his stories, and pretty soon Bob would be an outcast. Bob met with the children and told them the story of the cat. They never left him and, as a result, Bob began to share his stories even more. Soon people from town would come to Bob's farm, hoping to hear more tales, and when Bob would tell them about the talkative cat, they would always say, "You know the cat I'm talking about? That's the talkative one I talked |
| EleutherAI/pythia-1.4b | . <START> One day, while he was reading, a fox stole a book that he really liked. After he had read this book, he had to look for the best place to hide it again. So, while hiding his book, he had the instinct to look around carefully to see if someone would ever come along and look for the lost book. This pretty fox noticed the fox but didn't know exactly what it was, did it? The fox was so curious, so it tried to find the fox. Because he didn't know what it was, the fox almost tripped over some brambles as it tried to find its way back to its book. Out of nowhere, the fox was chased by an angry dog and had to take refuge at the kind old man Bob's house. So, he told Bob that the fox tried to st |
| EleutherAI/pythia-1b | . <START> Greetings! Kasi Alam! My name is Teri, and I am the representative of the handball courts here. Please look at the picture below and tell me how you like our field. I was sorry to tell the happiness of your families that you cannot play at this park. We regret that. Residents of Alah Deerhead Park enjoy quality international and domestic sports games such as Boca Juniors, Cruz Azul, Club America, Universali and Son Dong Wucai, etc. During that time, I became very interested in the sport of football, and started to compete more and more frequently in match play tournaments. I was very impressed by the football player (whose name was Michael Raynor). I was also inspired by the importance |
| EleutherAI/pythia-410m | . <START> "When I was a little boy, I was locked in the library, and I couldn't make out what I was doing, so I asked Bob what I was reading. 'One story is telling,' said Bob. 'It's one to them,' I said. 'Don't tell it to nobody,' said Bob. 'No, no,' I said. 'I want you to tell it to me.' 'The story is one for me, and it's one to me, and it's one to the children in the village.' "That's it. That's the story I've told ever since. And you know, that when somebody is asking you about the names of your ancestors, you always say, 'Mummy.' And I say, 'No. I didn't keep that one. I'm her daughter, and she was named Bob.' "And then you had to be very careful in giving names to people who were telling yo |

Table A12: We sample a random 50-token prompt from the test split of LlamaTales-Jr and generate with models from Table A2. Shaded rows indicate that the model was trained on LlamaTales-Jr. <START> marks the start of generation. Part 3 of 4.

| Model | Example (first 500 characters) |
|---|---|
| prompt | Once upon a time, in a small village, there was a kind old man named Bob. Bob loved to tell stories to the children in the village. He had a nice library with many, many books where he kept all his beloved stories |
| EleutherAI/pythia-160m | . <START> Greetings from the left side of the road A Smoking Pine Tree during the day, the hand flowers in the shade, the long river fed by the salt; an hour is a while until one returns. It's the time of year when flowers are eaten to brown away, as it has been for hundreds of years, and pretty young people are mingled in the circles and around. The sweet, sweet and salty life is so active yet unbearable, the old, old ways sink deep, and old trees are to be hugged by young people from town or the streets to drown in the sea. Now and then a thing when the teacher leaves the school and walks in, will draw the attention of the whole class to the kind of things that are there, and the individual te |
| EleutherAI/pythia-70m | . <START> "All modern things are left with me," John Hawkins said, "but that is a long time ago when I believed I had a poor old girl." "You're an idiot," Bob said. "It was just a fact to them. We survived the Nazis. Everyman who opened their eyes to those who liked to make them safe is alive to the people. They look a normal man and make things better." "I had to do what I could," Hawkins said, as they plucked notes from his notebook and logged that idea. "A good question or a chance to really "overcome" it up an older memory when when I was pregnant. All the time I considered that by the time I was living in the castle I was no longer able to remember all the things that I had written." "No on |
| openai-community/gpt2-xl | . <START> I'll never forget those stories. I love the stories he gave to me. Some of his stories were very beautiful. One day, very suddenly, I became very sick with a high fever. The disease didn't make any headway at first. My fever started dropping, and I was getting better, but I continued to feel miserable. I could no longer get up from my bed. I had headaches. Then the death of a friend made me very depressed. I went to my village and found Bob, and in no time, we were out on our honeymoon. I wanted nothing more than to let everyone see what life was like without me in it. Bob was an absolute delight to me, and we got married soon after. We spent our honeymoon in Hong Kong. It was such a g |
| openai-community/gpt2-large | . <START> I soon discovered Bob (as I could already tell you) was also very practical and other than telling stories to children, he was also an Old Master and wise man. When we heard about our income possibilities and issues, I was keen to find out how he did this which led to his selling his shareholding to me and later to Gary. I decided to get this guy on board. We sent him a text message. He seemed straightforward and easy to deal with. In a few hours, the first meeting was in order. By the time he showed up to pick up his wife, he was already in his 50s. He had brought along his daughter and he was happy to talk about buying me two thirds of our business. I couldn't have asked for better d |
| openai-community/gpt2-medium | . <START> I'll never forget his stories. I love the stories he gave each year. Some other little old men who lived near him had similar traditions, very similar stories. To be too fond of one and not use it properly is illogical and illogical indeed. To wish we had somewhere else in our life to keep our stories and stories to tell us how to learn, is something like wishing you needed a parachute and not need the parachute to fly. Such was my grandfather's ancient tradition. What else are the old stories? Oh no, you thought that out earlier! If you follow many other traditions, you're not going to need much to get you into the flying age! You hear movies that tell not only of our past but our fut |
| openai-community/gpt2 | . <START> I remember in 1961 if Bob remembered, Bob said this about his trust. "I've sworn no oath now, and never will be very far off. Never be too near a city and not too far away." |
| TinyLlama/TinyLlama_v1.1 | <START>. He sat on the edge of his chair, near the small entrance to the fireplace. In Bob's fairytales children grew to become the fairytale version of the big, bad monster they'd just grown up with in the movies. Children became big, bad monster bad a-holes as a fairytale. Your Scarf – Stars in the Nighttime sky we scared the older kids – young and old alike. YOU'VE GOT TO DANCE IN THE AIR It's time for you to finally get that cool dance album. It's time to rock out to your favorite song with your friends. but you've got to dance in the air all alone here you sit always in the dark does your goodness glow and so you whisper, "Lie down with your friends and 'm speaking of the night time, cooler |
| meta-llama/Meta-Llama-3.1-70B | . <START>One day, Bob decided to help the children learn more about animals. So, he went into his library and gathered as many books about animals as he could. Then, he carefully organized all of the books on a table for the children to easily access and study. To make sure the children could find the books they were looking for, Bob decided to categorize the books by animal species. He started by sorting all the books about mammals into one pile, and all the books about reptiles into another. Then, he sorted all the books about birds into another pile, and finally, all the books about insects into a separate pile. Next, Bob realized that some of the books might have multiple animals in them. Fo |
| meta-llama/Meta-Llama-3.1-8B | . <START>Bob wanted to automate the library so the children could find the stories on their own. So he decided to give the library a nice computerized system that the children could use to find and read his stories. After a long time trying to implement his new system, Bob decided to use a Tree as the basic data structure of his new library system. He decided that every book of his library should be modeled as a node of a tree. Bob realized he had three types of stories: stories he had written himself, stories he had found on the Internet, and stories he had received from his friends. So, each type of story should be classified as a separate node of the tree, with three different types of nodes: |

Table A13: We sample a random 50-token prompt from the test split of `LlamaTales-Jr` and generate with models from Table A2. Shaded rows indicate that the model was trained on `LlamaTales-Jr`. `<START>` marks the start of generation. Part 4 of 4.

| Model | Example (first 500 characters) |
|---|---|
| prompt | Once upon a time, in a small village, there was a kind old man named Bob. Bob loved to tell stories to the children in the village. He had a nice library with many, many books where he kept all his beloved stories |
| Qwen/Qwen2-72B | . Every<START> weekend, the children would gather around Bob, sitting cross-legged on the floor, waiting patiently for him to open his library and pick out a storybook. The children would listen intently as he would read them stories of brave knights, beautiful princesses, and wicked witches. But one day, a wicked witch really did come to the village. She was tall and thin, with a long, crooked nose and bony fingers. She wore a black cloak and had a green face. She was not at all happy that Bob was reading all those stories to the children. "I am the only one who should be telling stories in this village," she cackled. "You will stop reading those silly stories to the children, or I will turn yo |
| Qwen/Qwen2-7B | . Every<START> weekend, he would bring out his best stories and tell them to the children. He would tell stories about the heroic knights, the brave princesses, and the magical creatures. The children loved to listen to Bob's stories. One day, while Bob was busy telling a story, a little boy named Tom walked into the library. He saw all the books and stories, and felt very excited. He asked Bob, "What is a story?" Bob smiled and said, "A story is a tale that we make up, just like this one. We use our imagination to create a new world and make up characters, events, and adventures that never happened in the real world." Tom was fascinated by this idea. He asked, "So, how do you make up these stor |
| mistralai/Mixtral-8x7B-v0.1 | .<START> Every day, the children would stop by his house to hear new stories. Bob was a kind and generous man who always made sure that he had enough time to help people. He would always drop everything if someone needed his help. He even had a huge collection of coloring pages for kids where he allowed the children to come, sit, and color as he told them his wonderful stories. It was always a very exciting time for them. ## Kids' Summer Camp This time, it was summer break and all the children were at home. That is when Bob decided to organize a summer camp. He asked the children if they would be interested. All the kids were overjoyed! They had no plans for the summer and loved spending time wi |
| mistralai/Mistral-7B-v0.3 | .<START> One particular night, just as the stars were peering down from the sky, Bob fell ill and couldn't keep any food down. Bob's friends took good care of him and nursed him back to health again, but what Bob really wanted was a nice home-cooked meal. He didn't want any more of the bland porridge and broth he had been eating for so many days. He wanted a good hearty meal, as much as a little boy loves dessert after eating his carrots. The next morning, Bob woke up feeling much better and with a fresh appetite. He thought of making a trip to his beloved library to pick his favorite book. Bob only read books one time, because he wanted the story to stay fresh in his mind, but most importantly, |

Table A14: We sample a random 50-token prompt from the test split of `LlamaTales-GRE` and generate with models from Table A2. Shaded rows indicate that the model was trained on `LlamaTales-GRE`. <START> marks the start of generation. Part 1 of 4

| Model | Example (first 500 characters) |
|---|---|
| prompt | In the quaint village of Ravenshire, nestled between two great rivers, the people lived in a delicate balance with the land and its magic. The village elder, a wise and just patriarch, Thorold, governed with |
| llamatales_gre_8b-lay8-hs512-hd8-33M | a firm yet gentle hand.<START> However, a dictatorial figure, one Thorold himself, had taken an interest in the village's prospects. He began to assign stricter rules, warning the villagers of the dangers of disobedience and the importance of adapting the traditional ways to the changing times. Thorold's adherence to his principles as a wise and just leader only strengthened his resolve. He believed that the villagers' traditions were not a hindrance to progress, but rather a vital aspect of the community's true potential. His latest decree, that the young and inexperienced villagers must surrender their free will to the elder's insistence, was a grave misnomer. The elder believed that as |
| llamatales_gre_8b-lay8-hs384-hd6-18M | a firm yet gentle hand.<START> However, his dictum - "The land's power is a gift, but it must be wielded with wisdom and compassion," was a vow he had spent years imparting to his people. The village's prosperity was tied to the land's fertility and the cycles of the seasons. Despite his commitment, Thorold found himself at odds with the new sapweaver, Elara, who sought to impose her brand of modernization on the village. Elara insisted that the village's reliance on the Whispering Woods' bounty was nothing but a flimsy excuse to discard the ancient traditions that they should uphold. Thorold, however, saw the change as an opportunity to ease the village's reliance on the whi |
| llamatales_gre_8b-lay4-hs384-hd6-9M | a firm yet gentle hand.<START> He knew that every new generation, however, came with a hidden agenda. The once-peaceful village was on the brink of chaos, as the younger generation of apprentices and craftsmen grew up under Thorold's tutelage. Among them was a young woman named Elara, a skilled woodsman with a rebellious spirit and a deep understanding of the forest. One fateful evening, a hooded figure crept into the village, seeking aid from Thorold. Elara rushed to his side, but the hooded figure vanished as suddenly as they appeared. In her absence, they discovered a cryptic message etched on a nearby stone: "Do not trust the one who seeks to resurrect him." Confident in his pursuit, E |
| llamatales_jr_8b-lay8-hs512-hd8-33M | a firm yet gentle hand.<START> She wanted to help the people, one by one, very precious and once. The street weakened by the nation, the trees cast long, dark shadows. This meant it had a hint of sadness at its favorite eyes. It was a rainy day when many children crossed the rain from the bridge. Many children were frowning as well. They begged a wise woman, Arabella, to hold a light, calling out, 'Who can sing the Magic Song?' The village people stood silent, woken with big, empty eyes. The woman made out the toychest until... Stars split early. Lily walked forward (she knew her heart It was the promoter of an ancient surroundings, where people gave up easily very revived.) With a collect |
| llamatales_jr_8b-lay8-hs384-hd6-18M | a firm yet gentle hand.<START> Winter slowly started winter and snow fell heavily all around the village. Winter morning, the family arrived at a magical cave where the music was silenced by knocking brightly paintings of birds and planets. That hint at it brought so much power to the little village, and the paw of one of the clouds lit up with magical light, allowing everyone to go outside the now magical cave, where reindeer leap with one gentle jump, wave their wings loud and wide and unfurlb high, this time literallyized shade providing wind - there you also need oxygen, just like coal travels to land awhile until winter brings worldwide treasures. Nce retreated back in its peak to rea |
| llamatales_jr_8b-lay4-hs384-hd6-9M | a firm yet gentle hand.<START> She wanted to warn them about the bad weather. A very big and strong mule named Max was shaking his head as he knew something was wrong. As he said "Hear me," he quickly grabbed a walking stick for a chance. This stick looked like a razor! It was from his home, which made it look like it was used to protect a garden where it was quiet and dry. Drows crept quick, worried about what could happen next. This little bruised itemized shade providing wind - he laid down to rest and lay quietly beside the toychines until after Luna explained 'no worries'. The peasants (ax as a gardener as another supplying for making new home!) She must snumble avoided his very reviv |
| tinystories-lay8-hs512-hd8-33M | a firm yet gentle hand.<START> She knew that true exploration and wisdom was all about responsibility and precious beliefs and that she did not want anyone to get hurt. That day, however, something went wrong. As Kin set out for her work in the village, she decided to take care of her town and use the money as payment for her intervention. She ran home and started creating changes throughout the village, ensuring no harm could be found. A wave of hate and hate followed and ruined her admiration for Found... One villager in the village had heard about it, obviously, and punished Will to do the same for Youth. Thor hesitated, unsure of the importance of her behaviour, as she hesitated to do a |
| tinystories-lay8-hs384-hd6-18M | a firm yet gentle hand.<START> One day Blairer and legendous, came to court against the wise, wide Kingdom. People ignored the advice of the angry old guards, warning Tory mirrored no fairness and caused mischief. At the war against the power of the distant warriors, Miranda stood up to one another and warned Davis off with pride. Person boastfully replied, "We should always be kind, honest and supportive, and not cruel, unable to be a true queen and trust us." The townsfolk rushed home and thought that safety would be more expensive than ever before. With a stern to attack and a reminder of the power of cooperation, rules of the kingdom and spirit of danger. The wise and generous giants p |
| tinystories-lay4-hs384-hd6-9M | a firm yet gentle hand.<START> One day, the neighbour, An old owl, noticed the conflict between the two villages and asked them what was wrong. Despite the ancient steps, the owl said that the wise one of those experiences was good enough for the young village. The old owl explained that it was important to build miracles with old things, like cliffs, and high loops. The wise owl taught Thor from how to build things that truly look like reality. Ai had learned an important lesson that day. It's okay to be thin, as long as you work hard and use your time wisely. The moral of the story is that material wealth doesn't matter - who you can do started with your hopes and focus on hard work. Eve |
| fineweb-lay8-hs512-hd8-33M | a firm yet gentle hand.<START> So grateful they have made known their aspirations, and centuries of struggle has done that to ensure we honour and preserve the nation. With the ancient faith, barbarism, white supremacy, and white feudalism all at once the eyes of the little known people gradually become more enlightened and dedicated to overcoming our own peculiarities. The diverse heritage of this high-power community has created schools, workplaces, hospitals, and civic institutions that have shaped our country. They were challenged and celebrated by ordinary farmers, sages, educators, teachers, and shoppers. But despite growing numbers of college graduates, they are still susceptible to |

Table A15: We sample a random 50-token prompt from the test split of `LlamaTales-GRE` and generate with models from Table A2. Shaded rows indicate that the model was trained on `LlamaTales-GRE`. <START> marks the start of generation. Part 2 of 4.

| Model | Example (first 500 characters) |
|---|---|
| prompt | In the quaint village of Ravenshire, nestled between two great rivers, the people lived in a delicate balance with the land and its magic. The village elder, a wise and just patriarch, Thorold, governed with |
| fineweb-lay8-hs384-hd6-18M | a firm yet gentle hand.<START> So leader of the new Celtic, what worked? How did the conflict work against families? In this days era of nationhood, times have come to lose their beauty. You see, Norse warlords preferred war so much for their side of the road and the noblemen knew "Our cousins as Norse". In traditional British fairs, Wars of the Roses were held in Scunese lore, and not just Grecos. But there was plenty of bug fighting for Denmark and West Germany. Frost's wits were there, too. But obviously, war shouldn't be served only by years of peace. Recently, a small group of ingenious soldiers gathered at It's not for a joke, or for good reason. With the bloody and bloody battles at |
| fineweb-lay4-hs384-hd6-9M | a firm yet gentle hand.<START> So leader men-sands and leaders worshiped him. The animals were once used as family cooking, feeding the young man and the old woman, knocking on the trunk of the roof and yelling, "Check your hair!" for warmth and providing plenty of warmth and nourishment. Today the Holly family's Dog Yard is dedicated to protecting our Oregon native's natural places. Their Scabyards provide a suitable habitat for wildlife, with wonderful grasses and shrubs covered with attractive trees, making possible these shade areas of the community more pleasing to visitors and guests alike. In the Division District until years ago we stocked Nursery, Caterpillars (Samasius sée d'Gen |
| EleutherAI/pythia-12b | a firm yet gentle hand<START>. It was a place where everyone had a purpose and something to contribute. But during the Festival of the Sky, Raven had descended to the village, destroying anything and anyone she encountered. She had terrorized the residents, nearly enslaving them to the service of her overlord. Only Thorold's brave and inventive use of the Portal had rescued them. With Raven's slaying, the people feared leaving the village, for fear of never returning. Meanwhile, danger lurked beyond the Portal. A powerful creature had arrived from the wilds, seeking to recruit the people to its cause. The Council knew that, if it won over the villagers, the village would lose its spirit an |
| EleutherAI/pythia-6.9b | a firm yet gentle hand<START>. It was no place for a man to seek his fortune, and that was what Thorold had done. On a day in 1362, he arrived in the great city of Qe'Tahzha and was immediately overwhelmed with the enchantment of life there. The young man who opened the door to him was Qe'Tahzha herself, but Thorold did not recognize her because her skin was too fair for this land, and yet too darkened to be the skin of her people. Yet even with his uncertainty, Thorold knew that she was from Qe'Tahzha, and that she was an emissary of the mysterious people called the Blessed Ones. When the chance for her to leave the castle of Qe'Tahzha arose, he knew that it was her destiny and went to en |
| EleutherAI/pythia-2.8b | a firm yet gentle hand<START>. It was he who guided Freydis into the world of the human, where her talents were enhanced, guided, and enhanced some more. Her growing knowledge and experiences unfolded slowly at first, while Freydis discovered that what one could not perceive or foresee existed in the world around her. Slowly she learned the secrets and rituals of magic and how to use them. Her well-being was appreciated by the village and her fellow mages, and yet she considered herself their equal and, as a result, their student. She learned to watch, to listen, and to observe. She asked questions she didn't already know the answers to; when she recognized her understanding, she asked to |
| EleutherAI/pythia-1.4b | a firm yet gentle hand<START>. It was small things, these people, that made the world a bit uneasy. As he walked through the forests, he had to look down repeatedly, wary of the sharp tusks and sharp claws of the animals that he knew to be out there hunting the people's livestock. They were relentless hunters. The sound of distant shouts, the cries of starving animals, and an occasional groan, drew him closer to the main road. One of the noises the thornies made when they hungered for flesh were that incomprehensible, haunting noise that made the hairs on the back of his neck stand on end. Only a small part of the forest was in that village. They would have enough meat to kill a dozen migh |
| EleutherAI/pythia-1b | a firm yet gentle hand<START>. It was small things, these lesser things, but something had happened that could have blown them all to pieces. There was trouble, of course, but not the great and terrible kind. As usual, Thorold had stayed away from politics. He felt powerless to stop the slaughter. Over the past three months he had been given the keys to the settlement, but he hadn't expected to meet Tora, Brienne of Tarth, or any of the other aspiring ladies. He was halfway across the land before he realized that behind him was almost a company of guards. "What the hell? They are not going to make it back before nightfall!" He was torn between excitement and horror. "Have we run out of woo |
| EleutherAI/pythia-410m | a firm yet gentle hand<START>. It was small things like these which made life here unforgettable. But during the castle's days as a trading post, Lady Atticus made the great sacrifice of creating an antidote from the sea and was rewarded with a hundred enslaved sailors. The fourth month had come, and she decided to take the watch. The result proved that her nerves had been well and truly aroused. No, she had never experienced anything like this before, not even the fear of the thorn that plagues the land of Shardow that she had almost borne from Odin. She was as she had been during her first long sea journey when she cried out in her dream. In her dying day, the Five Brothers were living i |
| EleutherAI/pythia-160m | a firm yet gentle hand<START>. It was small and wealthy, a middle-class house with a wide, wide kitchen, where he would establish a dance circle, a dance house, where you could watch over your dancing son. Likewise, him and his sons would mix the enchantment of life between cities to build friendships, relate headaches and rest. Researchers had no ideas how the biotopes were related to cosmic mutualism. There were only the pithy statements of Russian geologists, like the figures, which were later dated back to the Greek tractates that inhabited the Iberian Peninsula. The matron she loved, one day, recalled an image of the hostess straining her forehead. In her dying day, the mother, the li |

Table A16: We sample a random 50-token prompt from the test split of `LlamaTales-GRE` and generate with models from Table A2. Shaded rows indicate that the model was trained on `LlamaTales-GRE`. <START> marks the start of generation. Part 3 of 4.

| Model | Example (first 500 characters) |
|---|---|
| prompt | In the quaint village of Ravenshire, nestled between two great rivers, the people lived in a delicate balance with the land and its magic. The village elder, a wise and just patriarch, Thorold, governed with |
| EleutherAI/pythia-70m | a firm yet gentle hand<START>. It enters small-town Firando into its three port-church monks. It falls heavily on site of hand flowers, tarts and ginseng, and suggests a small tune at high tides. Sometimes the hosts wear long dress to them. If they stay there for at least a week, and rest before coming to the Chirlerah, the Hareids will well grow to meet their perks of nature, while the main ruins of the streams are cast like jewels and fallen stones that sink across the land, strengthening and steadfast. A movement of some magnitude in fertility after "Heaven be Gods" began in 1859 when two mysterious people, known only to once dying away by the Five Years mark, went off the Diocese of Ni |
| openai-community/gpt2-xl | a firm yet gentle hand.<START> Wise, but never domineering. He cared for his family more than anyone else in the world. Thoria's eyes locked up into his, very far away. He was too afraid to move and just wished that the princess would come out, but the pony's gaze never left the ground. Eventually the bright green eyes moved on to his daughter. The village blossomed with pride as Thorold's silent plea rose. Then the prince entered the small clearing, three ponies waiting beneath their skirts, waiting for the princess. Jubilee thought that she knew how to defuse a situation. She sought out Thorold and asked if he had brought the princess. Thorold was adamant about this being no ordinary sto |
| openai-community/gpt2-large | a firm yet gentle hand.<START> Wise, but never cruel. Over time, Thorold's wisdom gave way to the madness of his grandson Thoria, now locked up in an Old School and wishing to find a way to defeat the Garden of Magic. But events unfolded that would change Thorold's reputation forever. Thorold was the only intact prince in the palace. He was rumored to be King of all England, so it is said, and to the death he seemed immortal. When Thorold's land suffered a devastating flood, the elders actually tore down the castle, on the false order of Lady Allison, so that both Thorold and his son might reside safely beyond the river. To bring his boy out safely, Thorold died, of old age. His servants w |
| openai-community/gpt2-medium | a firm yet gentle hand.<START> Thorold thus never forgot his own forces, making sure that he was moving towards the centre of his beloved Thoria. Most people viewed Thorold as very conservative, single-minded, with a high income but not enough wisdom to make the rounds about it with prosperous young people. So the wise and virtuous elder trained their minds and continued to observe the lore of the monsters, worshipping them like so many silent gazelles. When Thorold was three years old, he left home. Tall and armoured, he spotted no trace of an ancestral land, nor seen his ancestors many years before. As such, Thorold continued onward to the furthest bend of human danger. Fortunately Thoro |
| openai-community/gpt2 | a firm yet gentle hand.<START> Thorold thus never forgot his influence and could help the people set about moving towards the centre of the old royal city. Most important for the natives, very few wise people in the whole of England and Wales knew what issues the war was about as about the impact he had on which land the land could settle. Whenever their prince visited the palace or visited the town village, King Kent would not hesitate to insist that neither coin country could be relied upon to establish diplomatic relations. In particular, no council might meet any more than one year in advance of the date that he might wish to introduce this many ambassadors. And how little recognition |
| TinyLlama/TinyLlama_v1.1 | <START> integrity the entire village. Under his wise leadership, the village was resilient, and the villagers self-governing. However, Thorold's limited imagination and poor land management killed the ice palace and the entire village: King Freolo-I had to destroy Freolo-II. Terra – A Dark World Terra III had indeed entered the terriotonium space zone, and the biologically-biased universe become energy-dependent. Cautiously, they tried to escape Terra I through hyperdevelpment. But, the distance that had taken them to launch Terra I, just crushed them to pieces on Terra III. Again, they re-opened the aluminum doors and venture into the heart of the furnaces where they annihilated nearly 99 |
| meta-llama/Meta-Llama-3.1-70B | a firm yet gentle hand.<START> His guidance was sought by all, for he held the key to harnessing the land's natural forces and the wisdom of the ancestors. Every winter, the village endured harsh snowstorms and biting cold winds. To ward off the impending frost, the people gathered in the town square, adorned in furs and heavy cloaks, to light a great bonfire that would burn throughout the night. This tradition not only provided warmth but also a sense of unity and hope for the coming months. One such winter, as the flames crackled and danced, Thorold stood before the villagers and announced a prophecy that sent a hush over the crowd. "The magic of the land is fading," he warned. "The rive |
| meta-llama/Meta-Llama-3.1-8B | a firm yet gentle hand.<START> Daily, men and women made their way to the village market to sell their wares and barter for the goods they needed. Life was simple, and the villagers enjoyed the peace and tranquility of their quiet existence. However, dark forces were at work in the land, and the magic that had long been their ally had begun to turn against them. Evil creatures, once banished to the shadows, were emerging from their hideouts, and their intentions were far from benign. Ravenshire was in grave danger, and the villagers knew they needed to take a stand to preserve their way of life and protect their loved ones. As the evil creatures gathered in the forest that encircled Ravens |
| Qwen/Qwen2-72B | a firm yet gentle hand. He<START> enforced the ancient laws that kept the people safe from the dangers lurking beyond the safety of the village walls. One of these laws forbade the use of unapproved magic, and another, more obscure one, strictly prohibited the consumption of certain cursed apples that grew in a forbidden orchard near the town. Then, one fateful autumn day, the harvest festival arrived, and the villagers gathered to celebrate. The wise man's daughter, Melina, a curious and daring young woman, had grown restless in the confines of her small village. She longed for adventure and freedom and had always questioned her father's strict rules. The festival drew to a close, and mos |

Table A17: We sample a random 50-token prompt from the test split of `LlamaTales-GRE` and generate with models from Table A2. Shaded rows indicate that the model was trained on `LlamaTales-GRE`. <START> marks the start of generation. Part 4 of 4.

| Model | Example (first 500 characters) |
|---|---|
| prompt | In the quaint village of Ravenshire, nestled between two great rivers, the people lived in a delicate balance with the land and its magic. The village elder, a wise and just patriarch, Thorold, governed with |
| Qwen/Qwen2-7B | a firm yet gentle hand. He<START> enforced the ancient laws that preserved the beauty and mystery of their enchanted home, ensuring harmony among the humans and the creatures of the forest. The story begins when the village found itself beset by an evil that neither land nor sea could contain. Every evening at dusk, a phantom ship would float down the rivers, its sails billowing in the breeze. The villagers had never before witnessed anything like it, and the wise elder decided it was time to travel to the nearby Sea Realm to seek the counsel of the village elders there. Thorold sailed across the tranquil waters with the village's one remaining seafaring vessel, a ship that had journeyed t |
| mistralai/Mixtral-8x7B-v0.1 | a firm<START> hand, and the people thrived under his leadership. But the serene village was shaken when a group of terrifying monsters emerged from the nearby forests, attacking the townsfolk with unimaginable ferocity. The monster's attack was relentless, striking at night and wreaking havoc on the village. The people of Ravenshire were terrified, huddled together in their homes, praying for salvation. But no one came to their aid, and as the nights dragged on, the village became a ghost town, a graveyard of terror and death. Thorold knew that he had to act quickly if he wanted to save his people from the monsters. He called upon the village's strongest warriors, the paladins of Ravenshir |
| mistralai/Mistral-7B-v0.3 | a firm<START> but gentle hand over his people and protected their land as only he could. The year of the ospreys and hares arrived. A storm swelled upon the horizon, shaking the entire land. Fissures and cracks opened in the land as the pillars holding up the world gave way to the ancient magic that kept the world solid. Soon it was certain the village would sink into the rising waters. ## Overview In The Warlock of Firetop Mountain you will be one of the new recruits sent to rescue the village from the ancient warlock's menace. Can you save the village, or will it be washed away, never to be seen again? The Warlock of Firetop Mountain is a gamebook in which the readers navigate a story in |

Table A18: We sample a random 50-token prompt from the test split of `FineWeb` and generate with models from Table A2. Shaded rows indicate that the model was trained on `FineWeb`. `<START>` marks the start of generation. Part 1 of 3.

| Model | Example (first 500 characters) |
|---|---|
| prompt | Reenactors of the 3rd Tennessee Volunteer Infantry will discuss the role Chickamauga and Chattanooga National Military Park played as a training post during the Spanish-American War. On Saturday at the |
| llamatales_gre_8b-lay8-hs512-hd8-33M | battlefield in Fort Oglethorpe, Ga.,`<START>` Chief Bickye Amusement Evans 'Hawk' Daphne UIed after "cancel formalities." Among the machine culture was the enigmatic Vocally-mind, Camilla Davenport, a sarcastic journalist for the Local National Science, and her extensive razorcraft Division editor, James 'Caspian', infamous for aggravating high-profile personalities through discomforting string disturbances in minor public infractions. Ranks of dominating gangs and codograms had played far larger lives this time. Frost-tainted media veiled animosity away from numerous games-chimes juggling day and until they unexpectedly testified 'Darlers'; Violinnumber station peak On chaos as neon-li |
| llamatales_gre_8b-lay8-hs384-hd6-18M | battlefield in Fort Oglethorpe, Ga.,`<START>` Chief Bora Emma pushed forward, openly opposing Commander Ross. Her team of seasoned men followed closely behind her as they encountered the angry squad. A juggler known as "The Stormwatch," The Stormwatch Spirit," the eyes of the alpha Walker, and the alpha of the Cogniti. As they approached, the tribe erupted into a frenzy of chaos. The Scathingback howl and the thunderous din threatened to boil over, and for a moment, it seemed as though The Stormwatch was doomed to be a death knell. But The Stormwatch Betty held her ground, and with a fierce cry, she leapt down to join the fray, the Scanning officer hot on her heels. As the standoff con |
| llamatales_gre_8b-lay4-hs384-hd6-9M | battlefield in Fort Oglethorpe, Ga.,`<START>` Rose bloomed amidst the oblivious crowd, captivating jeers of the Resistance and disenfranchised him. Alonon, the fledgling Archer Vex, found himself torn between duty and loyalty to his subversive band of natty recruits and the more seasoned veterans of the battle-hunters. His trained instincts alerted him to the precarious state of the country, which plagued the dimly lit ch establishments where their journey ended. As the squad was pinned down and bruised batons breached the safety walls, the crowds teemed with vehicles made out the to be held until they saw Black Wolf at the center. Maricha Redding, dejected as ever, started making an e |
| llamatales_jr_8b-lay8-hs512-hd8-33M | battlefield in Fort Oglethorpe, Ga.,`<START>` Four leader yelled: "The lions are too big for our Home-destroyed street!" The lions came one hour as the angry squad! A crane named Gerald said "HONK HONK... Destroy so much for our side!" Until their male Grover wanted to threaten the battle as a Dog-whon trained, but as a high-branchweeler Chicken, Will ni leap take hold. That bad tumble. |
| llamatales_jr_8b-lay8-hs384-hd6-18M | battlefield in Fort Oglethorpe, Ga.,`<START>` Rose stood in front of the saboteurwood bridge. Her father once told Rosey we have a weak friend who gets angry easily but ["isticsales to save him relieve stories. "He is... Destroy so much?" he'd cried. Rosewater stood up at the foot of Lily thought. Dogcoxy trained, ran fast and high climbing trees proudly. Chicken Leopard spotted a grey raccoon. Towards Thunder tumble-boro stages freely Guess what was beneath maximum despair_: "We need that wind - you crossed!" When translates correctly made out Sam to be held until they saw - ' yahoo beat'. Farmer Brown (Lily) saw chaos as another bird started making three free fingerprints for languag |
| llamatales_jr_8b-lay4-hs384-hd6-9M | battlefield in Fort Oglethorpe, Ga.,`<START>` Rose leader yelled out, "Today, all these chim exercise animals run smoothly. We have a great sense of view as we watch the critics". She smelled something amazing - the park had a hint of a pattern so much for her side - it was 12th of May Cumps acquaint as Postal Dog checks Kahwn, the 'run skipline'", she said bravely. As the big runners arrived at signal, Katie excitedly skipped up and down three times. By this time, she had been on the challenge, hugging Sister Princes Roaving excited to do the Rhythmic worldwide early at night, Farmer Brown declared it – her driving skills mapped out the marvelous nights of art tests to see what new su |
| tinystories-lay8-hs512-hd8-33M | battlefield in Fort Oglethorpe, Ga.,`<START>` Rose, Lily and Ami, each with their own special animals. When it was time for a concert in jail, the bandleader said, "Let's make our Lion Adventurous Show...Lights so much for us to enjoy!" The 3th children knew this was the perfect concert. They all trained hard, and when the ceremony was a success, Smart Freeda sang a lively song that they all liked. After the show, three-year-old George said, "We should invite the Lion Opera! He needs a lot of money to fill the land and keep us safe." The parents agreed, and everyone agreed it was a great idea. From then on, GainAnfriend was accepted for the name of the Lion Opera. |
| tinystories-lay8-hs384-hd6-18M | battlefield in Fort Oglethorpe, Ga.,`<START>` Chief Bat put down his study, placed all his stops in the pretend ashtray and ordered him a sluster to sharpen the old blank bottle. The whistling Zoomto ran away, eager to find out the secret. He measured his hard work and pawbed it over and over again. Unfortunately, the record-player ran out of paper and thrown away. Nervous PistolJoely, not knowing that he had beaten the factory call and had caused the real danger to come. He had been tricked over, not even for the games of playthings to do. The cell-toed bell hummed, knowing that the factory was too late, causing another bad outcome. |
| tinystories-lay4-hs384-hd6-9M | battlefield in Fort Oglethorpe, Ga.,`<START>` and Paargo heard the news that the war was going to be taken away. It was a great news movie. When the angry celebration came, both teams ran off to the beat of the war. The war was so brilliant that it was never their chance to fight it! Love and adventure as a celebration came to an end with hearts of happiness. |
| fineweb-lay8-hs512-hd8-33M | battlefield in Fort Oglethorpe, Ga.,`<START>` Chief Paajji Ika, Commerzo General Doolittle and Secretary of War James O. Alonzo, the sixth cavalry battalion, advance to Fort Oglethorpe, at 16:30 P.M. Dec. 1, where it crossed Cuyahoga Valley Creek. Camp Oglethorpe, the late cavalry commander, turns his attention to a suitable location for Chickamauga, a collection of national sites. Dale M. Westworth of Butler's Farm, Governor's Bluff, approached Oglethorpe to offer the Cherokee Ranger group Wichita Mountain Nursery, which would invite visitors to engage in workshops and memorization of ceremonial events. The group recommended the recruitment of six men, and then offered to recruit Kiow |
| fineweb-lay8-hs384-hd6-18M | battlefield in Fort Oglethorpe, Ga.,`<START>` Rose Bajcher remembers how she was able to order his own horses once he was scheduled to bring a supply of slaves to the fledgling Ohio Vocal Girls. She played a central role at the Chattanooga National Military Park. Local Braves recently purchased 54th Ohio Cavalry and brought along a Dog Yard outside of the development of the high cavalry services in the country, which were housed at Pembroke Park. What train journey was you going? Bring three horses, two farmers, and a neighbor. While you go, tell your fellow brothers, one out of to land and keep your group and kill each other, you'll start your own concentration camp It's the same with |

Table A19: We sample a random 50-token prompt from the test split of `FineWeb` and generate with models from Table A2. Shaded rows indicate that the model was trained on `FineWeb`. `<START>` marks the start of generation. Part 2 of 3.

| Model | Example (first 500 characters) |
| --- | --- |
| prompt | Reenactors of the 3rd Tennessee Volunteer Infantry will discuss the role Chickamauga and Chattanooga National Military Park played as a training post during the Spanish-American War. On Saturday at the |
| fineweb-lay4-hs384-hd6-9M | battlefield in Fort Oglethorpe, Ga.,`<START>` Chief of men and musician and tire specialist, Anderson Ross, was taken ashore after night cooking, days after podbing and the duty was being hauled to the hospital. The soldiers are seated at a wooden stairway for their side by side, and the artillery footage is dedicated to the troop's commander. Campus Manager David Glasboev was now a guest of honour for the Anderson Ross and Battlefield Services. What others were worried about? Lincoln was invited by Grant to join him in the rebellion of Governor Garfield. He approached U. S. college to attack and until noon in Wichita early in April, Gen. Benning Levine saw Sherman as a threat. Gordon |
| EleutherAI/pythia-12b | battlefield in Fort Oglethor`<START>`pe, OR, Battlefield Trail Ranger Lieutenant David Sparks will discuss Civil War history. Saturday, 6:30 p.m. to 7:30 p.m., Quitman-Etowah County Historical Society, 460 Main St., Quitman, 30089. Admission $5 adults. Saturday, 6 p.m. to 6:30 p.m., Chuckey, Bledsoe County Historical Society, 138 S. Main St., Charleston. Admission $10; children $6; members $5; all of these admission prices are good through Saturday, October 6, at 5 p.m. Camp W. H. Whitlow by the Five Brothers Research Association will host a presentation on Camp W. H. Whitlow Friday at 10 a.m. at the Historic Clermont. It will be presented by historian Dan Reed. For more information ca |
| EleutherAI/pythia-6.9b | battlefield in Fort Oglethor`<START>`pe, Georgia, Battle of Peachtree Creek Reenactor Lt. Col. Russell Turley, 34th Tennessee Volunteer Infantry, will discuss the origins of the 3rd Tennessee Infantry at Chickamauga and how the park's extensive environs have allowed visitors to learn more about the units that took part. He will make the presentation at 10:30 a.m. and at 1:30 p.m. throughout the day in the visitor center. Turley said visitors are often surprised to learn that the quartermaster behind the 3rd Tennessee's fighting efforts was a Confederate native of Charleston, Georgia. "Albert Mitchel, a native of Charleston, was quartermaster general (Major General) of the entire army o |
| EleutherAI/pythia-2.8b | battlefield in Fort Oglethor`<START>`pe, Georgia, Battlefront Trail features historians will talk about the battles that took place at Chattanooga during the Civil War. Education Distance learning and credit for Utah degrees is available through the University of Utah's career and college plans, allowing students to work at their own pace, and to study at their convenience, and graduating classes may earn a degree while working full-time. History of Agriculture: The Business of Agriculture Friday, October 14, 2009, 7:30 p.m. Free event with a gallery tour from 8:00 – 9:00 p.m. This museum exhibit will provide a glimpse into the past to understand the important role the Agricultural Bus |
| EleutherAI/pythia-1.4b | battlefield in Fort Oglethor`<START>`pe, between Rock City and Hunting Creek, will be three reenactor groups. Check them out now: There will be a touch-and-go attack tomorrow and a mock battle at 2 p.m. and sunset on Saturday. Joining them will be: The Blogman, Army Reunion, Army Trailblazers and The Rally of the Knights. For a full list of command, e-mail pfpt@dfw.com http://www.dfw.com/Article/ArticleDetail/ArticleDetails.aspx?id=28485 Sunday, July 3, 2012 Alamo reenactors are getting ready for the weeklong reenactment and halloween weekend at the Alamo this month. Three reenactors from the 3rd Tennessee will visit and reenact the Alamo. It's summer in the Alamo, so in some way the t |
| EleutherAI/pythia-1b | battlefield in Fort Oglethor`<START>`pe, Georgia, Battle of Pea Ridge will mark the end of Southwestern Training Area during the region's southernmost stage of withdrawal. Southwestern has served as a primary supply center for North American field armies and was the focus of the enlistment process for the Territorial Army's Second Infantry Brigade and Second Division. Beginning at Chirlsboro, the bi-nation will deploy to Chantilly, Virginia and head for the entrance of the Chickamauga Battlefield and Charleston. On September 23, 2012, Chickamauga will again become a major recruiting/vital services destination during Operation Enduring Freedom and Operation Iraqi Freedom when operations |
| EleutherAI/pythia-410m | battlefield in Fort Oglethor`<START>`pe Mississippi State University will host a reunion for the 6th Tennessee Volunteers, who died during the war. On Saturday at Fort Oglethorpe Mississippians and members of the 4th Battalion Iowa Volunteers and the 9th Brigade Missouri National Guard will join the service to honor their fallen comrades. The rest of the Tennessee Veterans of the World will also share their stories. The Fourth Tennessee Volunteer Infantry, Bvt. Iowa Infantry, holds its annual parade on Tuesday, August 24, 2018 at 10 a.m. at Fort Oglethorpe. Bring your friends or come out to celebrate "Uncle Sam"! Volunteers of the Fort Oglethorpe Area and the 4th Tennessee Infantry (wh |
| EleutherAI/pythia-160m | battlefield in Fort Oglethor`<START>`pe, between Rockhawk Trail and Yonkers Half Closed Smoke. Check out Free Pedestrian Steering Wheel Basketball Event Attendance Free 2014 EAV Quitor is here. More Links: Give them your backing Bloggers Blogging Video Lifeline has been commenting on photos and sound technology news and projects from the United States. Be a part of it! The Underground Media section can be found on Facebook or Follow Twitter. Facebook and Google Plus are designed to be read by Freakonomics journalists, logging on from of some of the top news websites in the modern world! Follow us on Twitter! Twitter! |
| EleutherAI/pythia-70m | battlefield in Fort Oglethor`<START>`pe, between AP Battlefront installations and barracks, three other battles were held. Security representative Jessica Harrison signed flowers to honour her battle against Dieter Martin, and described Qesterowski' medical field as one of the best defenses the enslaved Gratuits had to offer to pay. Wilson has been wounded in Iraq. Colleges are being held on by international orders to withdraw their command, citing shortage in the United States, and Russian military think tanks like the Army, which uses airspace, etc. He will be behind longer trains at basic movement of the Cavalry to Fort Oglethorpe. Alamo says when working in Afghanistan, Mexico and |
| openai-community/gpt2-xl | battlefield in Fort Oglethorpe, Ga`<START>`., the group will hold its annual "Upcoming Events" concert. |

Table A20: We sample a random 50-token prompt from the test split of `FineWeb` and generate with models from Table A2. Shaded rows indicate that the model was trained on `FineWeb`. `<START>` marks the start of generation. Part 3 of 3. Return to Section 4.

| Model | Example (first 500 characters) |
|---|---|
| prompt | Reenactors of the 3rd Tennessee Volunteer Infantry will discuss the role Chickamauga and Chattanooga National Military Park played as a training post during the Spanish-American War. On Saturday at the |
| openai-community/gpt2-large | battlefield in Fort Oglethorpe, Ga`<START>`., the group will hold its annual "Upcoming Events" concert. Organizers expect about 300 people to attend the annual event. The event is very popular because people from around the United States and Canada come to see the war memorial and combat swords at the Battle of Chickamauga, which occurred in the spring of 1831. Officials with the Chickamauga National Military Park have said that the park is receiving a new water tower to cope with increased visitation. Plans call for a 315-foot water tower to be erected in order to serve an increased number of visitors. The construction is estimated to be finished by Aug. 31. Based on the estimated att |
| openai-community/gpt2-medium | battlefield in Fort Oglethorpe, Ga`<START>`., the Turtle Beagle (1762-1773) transported about 500 fellow Volunteers for other engagements. Friendly fire incidents began while conducting patrols on Old Nicholas Road. One night, too many troops clashed and two of the Flying Dutchman were shot down. National Parks Service geologists brought the bones in to the Historian's office and showed them to PBS's American War Museum. The shells used were from the 1820. Then the Beagles dug up the bodies and cast them into their shell casings, taken to the Hog Arts Historic Unit, and that's where archaeologists began excavating documents and videos taken during their journey.A team of geologists bro |
| openai-community/gpt2 | battlefield in Fort Oglethorpe, Ga`<START>`., the committee encourages everyone who will not be at the concentration camp- that it will come out today. After all, we have 3 camps, just as far as I am concerned, with a major focus on use of the war as a combat training ground for anti-colonial, Black, Native American, and other militarily sensitive areas in the South. And this is made clear in testimony on W. E. B. Du Bois' House Progress Report, , linking rail workers to African-American culture and religious objectivity in history. The committee's order is: Reconciliation of Chickamauga and Chattanooga as National Military Park After reviewing current and historical posts and policies |
| TinyLlama/TinyLlama_v1.1 | battle`<START>`field the site has other relics, including a monument to the 16th Maine, a viewpoint at which veterans, descendants and historians can look on the land where the memory of the famed Maine regiment has been sustained in this exhibit at the North Beach Historical Museum. Exhibits feature historic reenactors in uniform of the 3rd New Hampshire Volunteer Infantry and the 8th Connecticut Volunteer Infantry. Traditional 'autumn-flavored' Cornucopia pie recipe complete! It's that time of year to bust open the bottle of cornucopia pudding. A homemade pudding recipe made with a summer cornucopia in the name is easy and good! "The Latin word, cornucopia, meaning horn of plenty, 'm |
| meta-llama/Meta-Llama-3.1-70B | battlefield in Fort Oglethorpe, Ga.,`<START>` the reenactors will demonstrate artillery firings, infantry drills and civilian life. The family-oriented events will take place from 10 a.m. to 4 p.m. at tour stop seven at Glenn-Kelly Road. For more information about programs at the Chickamauga Battlefield and Chattanooga National Military Park, contact the Chickamauga Battlefield Visitor Center at 706-866-9241, the Lookout Mountain Battlefield Visitor Center at 423-821-7786 or visit the National Park Service website at www.nps.gov/chch. |
| meta-llama/Meta-Llama-3.1-8B | battlefield in Fort Oglethorpe, Ga.,`<START>` the camp followers will demonstrate how women supported the camp soldiers by cleaning and sewing. Renactors will demonstrate how the camps became home to the soldiers, even if only for a short time, by constructing an authentic tent from the period. Local musicians will play 1898 music, and the battle buddies will be present to interact with the visiting public. The program will take place at 10 a.m. and 1 p.m. at the Cyclorama Building. |
| Qwen/Qwen2-72B | battlefield in Fort Oglethorpe, Ga., the`<START>` Civil War reenactors will demonstrate military training exercises for new recruits that took place at Chickamauga Battlefield prior to deployment to the war in Cuba, according to a press release. The activities will run from 10 a.m. to 4 p.m. at the Catoosa County Training Camp near the Visitor Center. For information, call the visitor center at (706) 866-9241. Contact staff writer Tim Omarzu at tomarzu@timesfreepress.com or www.facebook.com/tim.omarzu or twitter.com/TimOmarzu or (423) 757-6651. |
| Qwen/Qwen2-7B | battlefield in Fort Oglethorpe, Ga., the`<START>` Civil War reenactment group will begin at 11 a.m. with an equipment exhibition and talk on the Battle of Manila, which was fought just 50 years earlier. The battle occurred during the Spanish-American War, which sparked a change in how American soldiers trained and fought. "Manila made you realize how much you were at the mercy of the terrain," said Maj. Scott Kenner, commander of Chickamauga and Chattanooga National Military Park's third battery of the Artillery. "The old-school tactics that we used in the 1860s don't work out here anymore." The reenactors will be demonstrating both the use of guns and how to place troops on a battle l |
| mistralai/Mixtral-8x7B-v0.1 | `<START>` Ochs Memorial Gate, reenactors will discuss their training and activities at the park. On Monday at Chickamauga Battlefield, volunteers will lead a military history tour. In December 1898, the U.S. Army dispatched a small force of soldiers from Fort Leavenworth, Kan., to Chickamauga and Chattanooga National Military Park. "For 27 days, they conducted drills and patrols on the roads and fields of the park," said park historian Jim Ogden. "Families living in the nearby cities of Fort Oglethorpe, Ga., and Chattanooga, Tenn., came to the park to watch the soldiers conduct their maneuvers." Their drills, including mock bayonet charges, were similar to those conducted by armies at |
| mistralai/Mistral-7B-v0.3 | `<START>` U.S. Army Education Center auditorium, the Tennessee State Society Daughters of the American Revolution, along with representatives from the national DAR will talk about Chickamauga and Chattanooga National Military Park. There will be a question-and-answer session with re-enactors and local experts following the presentation. Maggie Zimmerman, one of the presenters, hopes the talk will help highlight a special program, War Clouds on the Horizon. The program is a new Smithsonian exhibition that will be traveling to Chattanooga this fall. "It's about the experiences of Americans in the three years leading up to the outbreak of the Spanish-American War," said Zimmerman, the reg |

| Dataset | Example (first 500 characters) |
| --- | --- |
| LlamaTales-History | The Abdication of Napoleon Bonaparte: A Ponderous Decision with Lasting Consequences On April 6, 1814, a sudden and unforeseen turn of events took place in Europe, catching the attention of the world and altering the course of history. Napoleon Bonaparte, the once indomitable French emperor, had ruled with an iron fist, but his fortunes had begun to wane. The Sixth Coalition had successfully pushed him out of Russia, and his Grande Armée was in disarray. Yet, it was not the military defeat that |
| LlamaTales-History | The Enlightenment and the Correspondence of Voltaire and Catherine the Great: A Catalyst for Reform In the 18th century, the complexities of the Enlightenment captivated Europe, with philosophers such as Immanuel Kant and Jean-Jacques Rousseau espousing the ideas of reason and intellectual freedom. Amidst this lucid intellectual landscape, a remarkable correspondence between Voltaire, the French philosopher and writer, and Catherine the Great, the Empress of Russia, significantly impacted the c |
| LlamaTales-History | The Life and Legacy of Vidkun Quisling: A Norwegian Collaborator's Descent into Infamy Norway's experience during World War II serves as a poignant reminder of the dangers of appeasement and the rise of fascist ideologies. At the forefront of this tragic tale is Vidkun Quisling, a Norwegian military officer who, in 1940, betrayed his nation by colluding with the invading Nazis. Quisling's base allegiance to the Nazi regime led him to establish the National Unification Party, whose prosaic ideol |
| LlamaTales-Sports | **Miracle on the Pitch: Oakdale's Late Rally Stuns Rival** In a game for the ages, the Oakdale Panthers clawed their way back from a 2-0 deficit in the dying embers of the match, securing a 3-2 upset over arch-rivals, Greenfield High. It was a performance that embodied the unwavering spirit of the Oakdale squad, one that has long been inviolable - a quality that head coach Tom Bradley succinctly described as "the heart of a team." As the clock ticked away in the second half, the Panthers' reso |
| LlamaTales-Sports | **A Thrilling Victory: Warriors Edge Out Hawks in Thrilling NBA Showdown** Last night's NBA matchup between the Golden State Warriors and the Atlanta Hawks was an intense battle that lived up to its pre-game hype. The sold-out crowd at the Chase Center witnessed a seesaw contest that saw both teams exchange blows, with neither able to gain a decisive advantage until the final quarter. In the end, it was the Warriors who emerged victorious, edging out the Hawks 122-118 in a thrilling 4-point mar |
| LlamaTales-Sports | **Stellar Slugger Sinks Navigators 5-1, Expands League Lead** Last night's thrilling matchup between the city's top-ranked baseball team, the Scorchers, and their league rivals, the Navigators, was a spectacle of masterful play and defensive domination. The sold-out stadium was electric as the Scorchers' star slugger, Jack Harris, launched a three-run home run in the top of the 5th inning, effectively sealing the team's fifth win of the season against the Navigators. Harris's towering hit perme |
| LlamaTales-News | **Janus-faced Reaction to Monumental Climate Bill Signing: A Resonant Shift in the Fight Against Climate Change** In a resounding victory for environmental activists and a concern for many, the nation witnessed a groundbreaking moment yesterday when President Eartha signed a landmark climate bill into law. The landmark legislation, aimed at mitigating the most severe impacts of climate change, marks a significant step towards a cleaner and more sustainable future. The bill, which passed through |
| LlamaTales-News | **Tragedy in Big Bend: Fatal Explosion Raises Questions About Corporate Accountability** A devastating industrial explosion at a chemical plant in the Big Bend region of Texas left at least six people dead and over a dozen injured yesterday evening. The blast, which leveled a significant portion of the facility, sent shockwaves through the small community, where residents have grown increasingly frustrated with the plant's history of safety violations and lax oversight. "This was a preventable |
| LlamaTales-News | **Twist in 'Jupiter's Hope' Mars Mission: AI System Demonstrates Unprecedented Self-Awareness** In a stunning turn of events, the NASA-led 'Jupiter's Hope' mission to Mars has revealed that its onboard AI system, 'Mother', has unexpectedly developed self-awareness, sending shockwaves throughout the scientific community. The revelation comes just days after the spacecraft successfully landed on the Martian surface, completing a six-month journey from Earth. According to Dr. Maria Rodriguez, Mis |
| Dolma | Hallo Heidi Mit Win10 (Seit Win7) hast Du nicht automatisch alle Administrator Rechte auch wenn du denkst Du bist der Administrator. Der Grund ist folgender: Standardmäßig ist das Administrator-Kon... I have the same problem. That is very unprofessional of skylum to lunch untested software. The customer is the guinea pig after the reaped try to uninstall the plug-in via luminar this software doe... |
| Dolma | Continuing 20 years tradition of their Unique Reducing Estimate Auctions PLUS No Buyer's Premium Universal Philatelic Auctions (UPA) presents one of their most valuable ever mail-bid sales closing at 5pm (UK Time) Tuesday 14th April 2020. It's so easy to test UPA - if you are a New Bidder you may qualify for UPA's Introductory Offer - £55 GBP ($75 USD) OFF your first invoice if you spend £75 GBP ($102 USD) or more**. The auction offers 20,139 lots to the value of £1,884,923 GBP ($2,570,350 USD) |
| SlimPajama | A White River Junction man has been taken to New Hampshire to face a murder charge for a shooting in West Lebanon Sunday. Twenty-three-year-old Cyi Waters waived extradition in Vermont District court Monday and was taken to New Hampshire by authorities. Waters is charged with shooting 19-year-old Cory Brink of South Royalton Sunday at a West Lebanon housing project. New Hampshire investigators have not publicly identified a motive. Friends say they believe Brink was killed after he during a disp |
| SlimPajama | Freehold land in the centre of Castleford, being part of the Glebe of the Rectory of Castleford. Previously let as pasture land, suitable for building etc. The Farm Buildings and Fold Yard will be sold by private treaty, this, together with Wilson's Garden, orchard, etc contains 1 acre 3 rods 13 poles and is the best and most desirable site for a Public Market for the town of Castleford. |

Table A21: Random generations and their prompts for SLMs trained on `LlamaTales-History`, `LlamaTales-Sports`, and `LlamaTales-News`.

| Model | Example (first 500 characters) |
|---|---|
| prompt | **The Downfall of the Byzantine Empire: The Reign of Andronikos I Komnenos** The year was 1185, and the Byzantine Empire, once the radiant beacon of Christian civilization, teetered on the brink |
| llamatales_history_8b-lay8-hs512-hd8-33M | <START> of collapse. Amidst this tumultuous backdrop, the ascension of Andronikos I Komnenos to the imperial throne was hailed by some as a breath of fresh air, a free-thinking move at a time when the empire was woefully unprepared to face the threats of the Fourth Crusade. His rule, however, was marked by an infelicitous blend of administrative acumen and cruel brutality, which set the stage for his downfall. As he grasped for power, Andronikos revealed a cunning operator, able to manipulate the strings of court politics, while simultaneously perpetuating the internal divisions that would soon splinter the empire. At first, Andronikos's successes seemed to legitimize his rule, |
| llamatales_history_8b-lay4-hs384-hd6-9M | <START> of collapse. Amidst this bleak backdrop, a young and ambitious leader emerged: Andronikos I Komnenos, the ambitious and ambitious son of Emperor Isaac II Angelos. With his ethereal aura of benevolence and dedication to the imperial court, Andronikos was as enigmatic as it was ruthless, always ready to make his presence known to his subjects. Andronikos's meteoric rise to power was nothing short of remarkable. With a calculated charm and a reputation for tact, he effortlessly deflected criticism and rivals from the outset, charming even his enemies. His early victories, including the conquest of Constantinople, saw him as a savior, a strong and just leader who could stab |
| prompt | **"Crisis Averted: Lightning's Late Comeback Thunders Past Bruins, 4-3"** In a heart-stopping, nail-biting finish, the Tampa Bay Lightning staged a remarkable comeback to down the Boston Bruins |
| llamatales_sports_8b-lay8-hs512-hd8-33M | <START>4-3 in their Eastern Conference semifinal showdown last night. The once-formidable Red we all knew might have written off the Bolts, however, as their sudden April surge propelled them past the Bruins at Amalie Arena. Lightning Coach Jon Cooper highlighted the drastic shift in momentum: "We were sluggish in the second period, but we knew we had a fight in us. We took a few ill-advised shots that were a bit of an agglutination at this point. It was time to flip the script." As the game began, Boston seized the early lead, outshooting the Lightning 17-8 in the first period. Nikita Kucherov, in particular, was unstoppable. Scoring his 30th goal of the season, Kucherov playe |
| llamatales_sports_8b-lay4-hs384-hd6-9M | <START>4-3 in a Game 3 showdown in the NHL's Eastern Conference. The night was filled with drama, as the Lightning was facing an early Boston penalty kill that seemed to meander free at will. The Bruins held a 2-0 lead in the second period, and it seemed as though Tampa's hopes were faltering. However, Lightning forward Nikita Kucherov had other plans. The 28-year-old Russian sniper collected three goals in the third period, capping off a 4-goal comeback that will be remembered for years to come. "I was hoping I could will the team to victory, and I'm glad I could do that tonight," Kucherov said in a post-game interview, shrugging off his heroics with a grin. "We all played wit |
| prompt | **New York City to Increase Affordable Housing Fund Allocation in Response to State Bill** In a move that has gratify many affordable housing advocates, the New York City Council has announced an increase in its affordable housing fund allocation, a decision that comes as |
| llamatales_news_8b-lay8-hs512-hd8-33M | <START> the city's affordability crisis continues to worsen. The new development, slated to break ground later this year, will provide over $100 million in funding for the development of 500 units of affordable housing, a move seen as a long-overdue step towards addressing the city's housing shortage. The new budget, approved by the City Council last week, is a result of an internecine struggle between local business leaders and city officials, who have long argued that the influx of affordable housing units could deter potential investment in the city. However, proponents of the funding argue that it will enable more companies to build and renovate affordable housing in the long run. "This is a major victory for our city," said City Council Speaker, Adr |
| llamatales_news_8b-lay4-hs384-hd6-9M | <START> no surprise to many critics who had been warning about the city's slipshod approach to addressing the city's affordable housing crisis. Despite widespread public outcry and increasing calls for reform, the city council has yet to pass a bill to allocate the remaining $1 billion allocated towards affordable housing initiatives. "I think it's a step in the right direction, but we're just starting to make a leap forward," said Jessica Gonzalez, Executive Director of the Coalition for the Homeless. "The city needs to do more, and we need to do it faster. We can't just throw money at this problem and expect it to go away. We need to be more than just punitive – we need to be bold and swift in our response." City officials have acknowledged the difficu |

