# OpenReview forum: "Readability ≠ Learnability: Rethinking the Role of Simplicity in Training Small Language Models"
_colmweb.org/COLM/2025/Conference — COLM 2025_

### Official Review · Reviewer_nRzk · 2025-04-24

**Rating:** 7
**Confidence:** 3
**Ethics Flag:** 1

**Summary:**

The paper contests the claim that very small models are able to generate simplified text when trained on child directed datasets such as TinyStories because they have accessible vocabulary, and simple syntax and narrative structure. The authors argue that this could be because of statistical simplicity i.e., less n-gram variation by building a synthetic dataset prompting for more adult directed language and comparing very small (262k to 33M without embeddings) parameter language models built with these datasets on a set of text completions and measuring the generated outputs across various dimensions using a LLM as a judge setup.

The paper explores an interesting idea and does focused experiments related to it. The writing is largely clear and I don't find anything particularly problematic.

**Questions To Authors:**

Some details about the approximate size of the dataset (i.e., average number of words per text and the likes), more details on how the conclusions drawn vary by the model size (your results seem to focus only on the 33M model) would be good. Some of these is already seen in the supplementary material, but not discussed in the main paper. Perhaps you can at least list some major observations and direct the reader to the supplementary material in the end.

At least mention briefly that you only look at English and we can't say whether these observations hold good beyond that. While a limitations section does not seem to be mandatory in COLM, it doesn't hurt to add some nuance to the conclusions section.

**Reasons To Accept:**

The paper explores an interesting idea and does focused experiments related to it. The writing is largely clear and I don't find anything particularly problematic.

**Reasons To Reject:**

No strong reasons. I have a few suggestions which I think are minor revisions, listed under the next question.

---

> ### Author Response · Authors · 2025-06-02
>
> We thank Reviewer nRzk for their thoughtful review and positive assessment of our work. We appreciate the helpful suggestions and will address each point below.
>
> 1.  **Dataset Details:**
>     The reviewer suggested providing more details about the dataset characteristics in the main paper. This is a helpful point. In the revised manuscript, we will incorporate a brief summary of key dataset statistics, such as average text length (drawing from metrics like "Tokens / Document" already present in Table A3), directly into Section 2 (Dataset Construction) or Section 3.4 (Validation of Dataset Construction). This will offer readers a better immediate understanding of the datasets' scale, complementing our n-gram diversity analysis.
>
> 2.  **Variation by Model Size:**
>     The reviewer makes a valuable suggestion to discuss observations across different model sizes more explicitly in the main paper. We will revise Section 4 (Results) to highlight a key finding from Figure 3: even our smallest SLMs (e.g., in the 262K-1M parameter range) effectively learn the patterns within the statistically simple training data, resulting in in-distribution coherence scores that are often comparable to much larger, albeit not state-of-the-art, public models. This underscores how effectively even very low-capacity models can learn from statistically simple data. We will also note that the supplementary material provides further evaluations across various metrics and conditions, broken down by model size.
>
> 3.  **Language Limitation:**
>     We agree with the reviewer on the importance of noting the linguistic scope of our study. In the revised Discussion and Conclusion section (Section 5), we will add a statement to clarify that our current findings are based on experiments in English. We will also suggest that investigating the generalizability of these observations on readability versus statistical simplicity to other languages, which may have different structural properties, would be a valuable direction for future research. This will add important nuance to our conclusions.
>
> Thank you again for your time and valuable feedback. We are confident that these changes will improve the clarity and completeness of our work.

---

> > ### Comment · Reviewer_nRzk · 2025-06-04
> > **Thank you for the response**
> >
> > Dear Authors, thank you for the response. If the changes are incorporated, the paper would have some good insights for readers. I am keeping my initial recommendation.

---

### Official Review · Reviewer_QjJm · 2025-05-09

**Rating:** 7
**Confidence:** 4
**Ethics Flag:** 1

**Summary:**

The purpose of the authors paper is to prove that coherence that is learned from small languages model doesn't come from how readable a dataset it but how statistically diverse it is. The core contribution is developing a series of synthetic datasets where the authors control the readability followed by training small language models and using LLM as a judge to evaluate the coherence of the model.

**Questions To Authors:**

(1) For determining LLM-as-a Judge for quality how did you determine the ranking of models in Table A4?

**Reasons To Accept:**

(1) The main purpose the authors want to get across is coherence in small models does not come from simplified language and we shouldn't be framing training models in this way as it could be misleading. I think showcasing this is important.

(2) The authors follow careful data validation. Additionally the paper is structured well.

**Reasons To Reject:**

(1) One thing that's always difficult when discussing metrics like coherence is the exact definition and how does that differ from fluency and clarity that the authors also mention. I think a more clear definition of this could help.

One thing that could've affected results is response length. Do models that generate longer responses are judged as more coherent? Should we control for that?

Here's some reference that discusses the many different definition issue https://aclanthology.org/2020.inlg-1.23.pdf

---

> ### Author Response · Authors · 2025-06-03
>
> We thank Reviewer QjJm for their positive evaluation, for recognizing the importance of our paper's message, and for their insightful questions and suggestions. We are pleased the reviewer found our data validation careful and the paper well-structured.
>
> 1.  **Definition of Coherence (and relation to fluency/clarity):**
>
>     We appreciate you raising the point about defining coherence and for sharing the Howcroft et al. (2020) reference. That paper indeed underscores the recognized field-wide challenge in standardizing NLG quality terms, an assessment we agree with. Our paper's aim was not to propose a definitive definition of 'coherence,' but rather to pragmatically measure an intuitive component of overall text quality --- specifically, the aspects that distinguish well-structured, understandable narratives from flawed ones.
>
>     To this end, we *operationalized* coherence using the specific prompt shown in Fig. A12. Recognizing that any single such operationalization has inherent limitations, a key element of our methodology was to also assess other intuitive aspects of text quality, such as fluency and consistency, each with its own distinct prompt (e.g., prompts shown in Fig. A18 for fluency and Fig. A19 for consistency, with corresponding results in Figs. A5 and A7). Our paper's central conclusions regarding dataset effects (the roles of statistical simplicity vs. readability) remain consistent across these different prompted evaluations of quality (as noted in Sec. 3.2, lines 151-154). This consistency across several distinct quality indicators suggests our findings are not merely an artifact of our specific operationalization of 'coherence.' We will update Section 3.2 to better clarify this approach and underscore the resulting robustness of our findings.
>
> 2.  **Response Length Confound:**
>
>     Your question about response length as a potential confound is a valid and important one. We deliberately did not strictly fix output lengths, reasoning that different narrative types (e.g., child-directed vs. adult-level) naturally have different characteristic lengths (as shown for our source data in Table A3). We believed forcing artificial length constraints could itself introduce unnatural outputs that unfairly impact perceived coherence.
>
>     While this means generation length was an uncontrolled variable, two observations suggest that this potential confound is unlikely to undermine our core findings:
>     *   First, our training data characterization (Table A3) shows that LlamaTales-GRE source texts (avg. ~500 tokens/doc, coherence 94.4) are substantially longer than LlamaTales-Jr source texts (avg. ~283 tokens/doc, coherence 89.5). While a minor difference in these source coherence scores exists, we contend that both values are sufficiently high, confirming that each dataset independently represents a corpus of well-structured narratives. This provides a strong basis for assessing how well SLMs can learn to reproduce such coherence, irrespective of the differing readability levels or natural lengths of these distinct narrative styles.
>     *   Second, consistent with this, when evaluating *model outputs* (Figure 3), SLMs trained specifically on LlamaTales-Jr (which tend to generate shorter stories) and SLMs trained specifically on LlamaTales-GRE (which tend to generate longer stories) can both achieve high in-distribution coherence scores, often approaching the levels set by very large public models on those respective tasks.
>
>     In the revised manuscript (Section 4), we will add a brief note acknowledging our methodological choice regarding generation length and the associated trade-offs. We will explain why we believe our conclusions are not primarily driven by this factor, emphasizing that the high coherence achieved by our SLMs is evaluated relative to the strong performance of large public models within each respective data distribution (Figure 3).
>
> 3.  **Basis of Model Ranking in Table A4:**
>
>     Thank you for your question about the model ranking in Table A4. This was a pragmatic ranking based on generally understood tiers of model capability, considering factors like parameter count and recency (e.g., recent Llama 3.1 or Qwen2 models are typically more capable than older Pythia models of comparable size). This ranking was not derived from new, formal benchmarking.
>
>     Its purpose was to serve as a sanity check to ensure our LLM-judged coherence metric could discern general differences in generative quality---which Figure 1b shows it did, outperforming perplexity. This pragmatic ranking was also corroborated by our manual inspection of sample generations (a few examples are in Tables A6-A20), which supported the general ordering for our tasks.
>
>     We will add a sentence to Section 3.2 in the revised manuscript to clarify this basis.
>
> Thank you again for your time and valuable feedback. We are confident that these changes will improve the clarity and completeness of our work.

---

> > ### Comment · Reviewer_QjJm · 2025-06-03
> > **Addressing comments**
> >
> > Thank you for addressing my comments. If you could make the changes you mentioned I think that would help the paper.

---

### Official Review · Reviewer_NYqh · 2025-05-20

**Rating:** 6
**Confidence:** 4
**Ethics Flag:** 1

**Summary:**

The paper aims to relate readability and statistical complexity with learnability. It provides evidence that learnability stems from statistical complexity rather than from the readability of the training dataset. Such result dilutes previously claimed similarities between human and LLM learning dynamics.

To test such hypothesis, the authors generate synthetic datasets which vary the readability while maintaining the quality of the dataset (measured by "coherence"). To accomplish such level of granularity, the datasets are generated using prompted LLMs and restricting their output vocabulary. Then SLMs (Small Language Models) are trained on such datasets and evaluated on readability and coherence.
Both readability and coherence scores are obtained from a prompted LLM evaluating the generated text of the SLMs.

The paper provides 3 main results as evidence that readability $\neq$ learnability.

1) High coherence can be achieved by training on less readable data, without resulting in coherence for more readable data.
2) Coherence is negatively correlated with the number of unique trigrams in the dataset.
3) The time of coherence emergence through training does not depend on the readability of the training data.

Furthermore, the paper includes a final figure to show that the SLMs are not simply memorizing/overfitting the training dataset. Instead they are able to generate new n-grams (especially higher order ones).

**Questions To Authors:**

In Figure 5, it seems like smaller models (trained on synthetic data) tend to produce more n-grams that are not in the training data than larger ones. Do you have any intuition on that?

**Reasons To Accept:**

1) The goal of the paper is well written and the experiments address the paper's claims.
2) It is a technically demanding paper as it involved pre-training many SLMs. The training and evaluation of models seems to be well performed.
3) The findings could potentially change the community's view on learnability and generalization of transformer-based LMs.

**Reasons To Reject:**

## Figure 4b, not convincing emergence

I would like to see a more dense sampling of the checkpoints for figure 4b.
For the score to be comparable across datasets/domains, I would expect the score to be similar early in training.
In the figure we see very different vertical offsets (for more readable vs less readable). If a more dense sampling does not solve that, it is likely that the coherence score is biased.

---

## Figure 5, not convincing baseline

Llama-70B cannot be really compared here, since its training data is not known. It is not clear to me how this model can be used as a baseline.

---

## Generating and evaluating with LLMs

It is a bit odd to both generate training data and evaluate the generated text from SLMs with LLMs. This could introduce biases in the coherence and readability scores, therefore, limiting the generality of the findings.

---

I would like to see more links with previous work on learnability:
(Nguyen, 2024) Understanding Transformers via N-gram Statistics
(Kallini, 2024) Mission: Impossible language models

---

> ### Author Response · Authors · 2025-06-03
>
> We thank Reviewer NYqh for their detailed review and for recognizing the demanding nature of our work, its well-written goals, and its potential to change the community's view on learnability. We appreciate the opportunity to address the concerns raised and clarify our findings. We address your concerns below:
>
> 1.  **Figure 4b - Coherence Emergence Dynamics (R1):**
>
>     Thank you for your close attention to Figure 4b and for raising this point about the initial coherence scores. The primary purpose of Figure 4b is not to compare the absolute coherence values between models trained on different datasets at any given point in training. These datasets (e.g., LlamaTales-Jr vs. LlamaTales-GRE) are distinct, and as our own data characterization shows, both represent highly coherent source material (Table A3). Instead, Figure 4b aims to illustrate the *rate at which models approach the coherence levels characteristic of their respective training data*.
>
>     The key observation is the learning trajectory towards each dataset's coherence level: models trained on the less readable, more complex LlamaTales-GRE dataset (blue line) exhibit a rapid increase, reaching a high level of coherence (comparable to its source data's coherence) very early in training (e.g., by the first epoch). This rapid attainment of coherence on complex text challenges the intuition that simpler text would necessarily enable faster initial learning.
>
>     Your feedback helps highlight how we can make the emphasis on this rate of achieving each dataset's coherence level, rather than on comparing initial absolute values, much clearer. Clarifying this point in the revised manuscript, as you've helped us see, will allow the dynamic shown in Figure 4b---particularly the rapid learning from complex text---to more effectively underscore one of our paper's central counter-intuitive findings. We will enhance the discussion accompanying Figure 4b accordingly.
>
> 2.  **Figure 5 - N-gram Novelty Baseline (Llama-3.1-70B) (R2):**
>
>     Thank you for your question regarding the Llama-3.1-70B model as a baseline in Figure 5. We would like to clarify its role and how its novelty was calculated, which we believe addresses your concern.
>
>     The Llama-3.1-70B model was *not* trained on any of our synthetic datasets (TinyStories, LlamaTales-Jr, LlamaTales-GRE). Instead, it serves as a reference point representing a highly capable, general-purpose LLM trained on a very broad corpus. For Figure 5, its n-gram novelty is calculated as follows: Llama-3.1-70B generates text based on prompts from the respective synthetic dataset (e.g., TinyStories prompts for Fig. 5a). The n-grams in these generations are then compared against the n-grams present in that *specific synthetic training set* (e.g., the TinyStories training data for Fig. 5a).
>
>     Therefore, the "% novel" for Llama-3.1-70B (the black dots in Figure 5) indicates the proportion of n-grams in its output that are absent from the narrow synthetic training data it's being compared against. As Figure 5 visually confirms, because Llama-3.1-70B was not exposed to these specific narrow datasets, its outputs are, as expected, highly novel with respect to them, consistently showing the highest novelty across n-gram sizes. This establishes a useful upper reference for diversity.
>
>     This baseline helps contextualize the novelty of our SLMs. Figure 5 shows that our SLMs (colored dots) are recombining learned patterns rather than merely memorizing. However, their novelty is generally well below that of the Llama-3.1-70B reference, illustrating that their generations, while demonstrating novelty, are still largely within the statistical confines of their narrow training distributions.
>
>     We recognize that the role of this 'external' reference model could be made more explicit. In the revised manuscript, we will enhance the caption for Figure 5 and the accompanying text to clearly explain how the Llama-3.1-70B novelty is calculated and its purpose as a benchmark for high diversity relative to our specific synthetic training sets.

---

> > ### Author Response · Authors · 2025-06-03
> >
> > 3.  **Generating and Evaluating with LLMs (R3):**
> >
> >     Thank you for raising this important consideration about using LLMs in both our LlamaTales data generation and the subsequent evaluation pipeline. We understand the concern regarding potential biases and how this methodology might affect the generality of our findings.
> >
> >     A cornerstone of our approach to mitigate such concerns was the external validation of our LLM evaluator (Llama-3.1-70B-Instruct). As shown in Figure 1a, its readability assessments demonstrate strong correlation with human judgments on the CLEAR dataset, grounding its ability to evaluate readability in line with human perception. Furthermore, its coherence assessments (our primary quality proxy) align well with a pragmatic model ranking (Fig. 1b, Table A4) that was itself corroborated by our manual inspection, supporting its ability to discern general differences in generation quality.
> >
> >     Within our core experiments comparing LlamaTales-Jr and LlamaTales-GRE, these datasets were generated using the same Llama-3.1-8B-Instruct model and prompting framework, with the deliberate difference being the instructions related to readability. Both were then evaluated using Llama-3.1-70B-Instruct. Any systemic biases inherent to using Llama-family models in this pipeline would likely apply consistently across both these experimental conditions. Therefore, the observed relative differences in SLM learnability (i.e., the stronger role of statistical simplicity over readability) are most parsimoniously attributed to our controlled manipulation of dataset properties, rather than an LLM-specific bias that would coincidentally produce these specific comparative results.
> >
> >     In the revised manuscript, we will add a brief discussion acknowledging this methodological choice and highlighting these key mitigating factors---particularly the external validation of our evaluator and the controlled nature of our core comparisons---to provide transparency regarding the basis of our conclusions.
> >
> > 4.  **Links to Previous Work on Learnability (R4):**
> >
> >     Thank you for pointing out these specific recent papers: Nguyen (2024) "Understanding Transformers via N-gram Statistics" and Kallini (2024) "Mission: Impossible language models." We appreciate you bringing these relevant works on learnability and n-gram statistics to our attention.
> >
> >     We will carefully review both papers. In the revised manuscript, we will incorporate citations to these studies and, where appropriate, briefly discuss their connections to our findings. Engaging with this current research will help contextualize our contributions.
> >
> > **Regarding your Question (Figure 5 - SLM Novelty Trend) (Q1):**
> >
> > This is an astute observation based on Figure 5. Once the role of the Llama-3.1-70B model as an external, high-novelty reference is clarified (as discussed in our response to your second point, R2), we can focus on the subtle trend you've identified *among our trained SLMs*: indeed, for a given synthetic dataset, smaller SLMs (e.g., 9M parameter models) sometimes exhibit slightly higher n-gram novelty than the larger SLMs within our experimental range (e.g., 18M or 33M models), though all SLMs remain well below the Llama-3.1-70B reference.
> >
> > Our primary intuition for this observation is that smaller-capacity models may be less adept at perfectly capturing and faithfully reproducing even a statistically simple distribution. Consequently, their generations might more frequently deviate from the most common patterns present in their narrow training data. These deviations can manifest as sequences that are technically 'novel' with respect to that limited training set. This slightly higher novelty in smaller SLMs could thus reflect a greater tendency to produce less constrained or marginally less coherent combinations, essentially 'novelty through imperfect learning' of the target distribution. Larger SLMs within our range, being more capable, might capture that narrow distribution more precisely, leading to outputs that are more consistently 'in-distribution' and therefore exhibit slightly less novelty relative to it.
> >
> >
> > We hope these clarifications and our planned revisions address your concerns. We are grateful for the detailed feedback, which will help us improve the paper significantly. We believe that with these clarifications, the contributions of our work will be even more apparent.
> >
> > Thank you again for your time and thorough review.

---

> > ### Comment · Reviewer_NYqh · 2025-06-03
> >
> > Thanks to the authors for their detailed response, addressing most of my concerns.
> >
> > ---
> > ## Fig 4b:
> > I would still like to include the Epoch 0 coherence values for each dataset. The emergence profiles should share the offset at initialisation.
> >
> > ---
> > ## Fig 5:
> > The response clarified my concerns.
> >
> > ---
> > I have updated the score accordingly.

---

> > > ### Author Response · Authors · 2025-06-10
> > >
> > > Thank you for the follow-up and the updated score.
> > >
> > > You've raised a fair point. In the revised manuscript, we will add the Epoch 0 data points to Figure 4b to provide a clearer baseline for the learning trajectories.
> > >
> > > We agree this will strengthen the paper. Thank you for your helpful feedback.

---

### Decision · Program_Chairs · 2025-07-08

**Decision:**

Accept

**Comment:**

This paper aims to test the hypothesis that high readability in the training data does not necessarily mean high learnability. The authors construct synthetic datasets which vary in their readability (measured using LLMs but calibrated with human-annotated datasets). They then use these datasets to train SLMs from scratch and evaluate them on readability and coherence (again using LLMs). One of the main findings is that less readable data does not necessarily result in lower coherence providing some counterpoints to how works like TinyStories are often interpreted by the community.

All reviewers agree that
* this is very interesting study
* it is well executed and presented
* the results may change the community's view on training dynamics, especially in small language models.

All reviewers make good points on how this paper could be improved and I highly recommend that the authors follow these suggestions when preparing the final version of this paper.